# Estimates of ikaite export from sea ice to the underlying seawater in a sea ice-seawater mesocosm

Geilfus N.-X.[1,2], Galley R. J.[1], Else B. G. T.[3], Campbell K.[1], Papakyriakou T.[1], Crabeck O.[1], Lemes M.[1], Delille B.[4], Rysgaard S.[1,2,5]

[1] Centre for Earth Observation Science, Department of Environment and Geography, University of Manitoba, Winnipeg, Canada

[2] Arctic Research Centre, Aarhus University, Aarhus, Denmark

[3] Department of Geography, University of Calgary, Calgary, Canada

[4] Unité d'Océanographie Chimique, Université de Liège, Liège, Belgium

[5] Greenland Climate Research Centre, Greenland Institute of Natural Resources, Nuuk, Greenland

## 1. Abstract

The precipitation of ikaite and its fate within sea ice is still poorly understood. We quantify temporal inorganic carbon dynamics in sea ice from initial formation to its melt in a sea ice-seawater mesocosm pool from 11 to 29 January 2013. Based on measurements of total alkalinity (TA) and total dissolved inorganic carbon ($T$CO$_2$), the main processes affecting inorganic carbon dynamics within sea ice were ikaite precipitation and CO$_2$ exchange with the atmosphere. In the underlying seawater, the dissolution of ikaite was the main process affecting inorganic carbon dynamics. Sea ice acted as an active layer, releasing CO$_2$ to the atmosphere during the growth phase, taking up CO$_2$ as it melted and exporting both ikaite and $T$CO$_2$ into the underlying seawater during the whole experiment. Ikaite precipitation of up to 167 µmol kg$^{-1}$ within sea ice was estimated while its export and dissolution into the underlying seawater was responsible for a TA increase of 64 to 66 µmol kg$^{-1}$ in the water column. The export of $T$CO$_2$ from sea ice to the water column increased the underlying seawater $T$CO$_2$ by 43.5 µmol kg$^{-1}$, suggesting that almost all of the $T$CO$_2$ that left the sea ice was exported to the underlying seawater. The export of ikaite from the ice to the underlying seawater was associated with brine rejection during sea ice growth, increased vertical connectivity in sea ice due to the upward percolation of seawater, and meltwater flushing during sea ice melt. Based on the change in TA in the water column around the onset of sea ice melt, more than half of the total ikaite precipitated in the ice during sea ice growth was still contained in the ice when the sea ice began to melt. Ikaite crystal dissolution in the water column kept the seawater $p$CO$_2$ undersaturated with respect to the atmosphere in spite of increased salinity, TA, and $T$CO$_2$ associated with sea ice growth. Results indicate that ikaite export from sea ice and its dissolution in the

underlying seawater can potentially hamper the effect of oceanic acidification on the aragonite saturation
state ($\Omega_{aragonite}$) in fall and winter in ice-covered areas, at the time when $\Omega_{aragonite}$ is smallest.
**2.  Introduction**
Currently, each year, 7 Pg of anthropogenic carbon are released to the atmosphere, 29% of which is
estimated to be taken up by the Oceans through physical, chemical and biological processes (Sabine et al.,
2004). The Arctic Ocean are taking up -66 to -199 Tg C year$^{-1}$, (where a negative value indicates an
uptake of atmospheric $CO_2$) contributing 5-14% to the global ocean $CO_2$ uptake (Bates and Mathis,
2009), primarily through primary production and surface cooling (MacGilchrist et al., 2014). However,
polar ocean $CO_2$ uptake estimates consider sea ice as an impermeable barrier, ignoring the potential role
of ice-covered areas on gas exchange between the ocean and atmosphere. Recent studies have shown that
sea ice covered areas participate in the variable sequestration of atmospheric $CO_2$ into the mixed layer
below the ice (e.g. Papakyriakou and Miller 2011; Geilfus et al., 2012; Nomura et al., 2013; Delille et al.,
2014; Geilfus et al., 2014; 2015). Studies are required to elucidate the processes responsible as well as
their magnitudes both temporally and spatially.
The carbonate chemistry in sea ice and brine is spatially and temporally variable, which leads to
complex $CO_2$ dynamics with the potential to affect the air-sea $CO_2$ flux (Parmentier et al., 2013). Release
of $CO_2$ from sea ice to the atmosphere has been reported during sea ice formation from open water
(Geilfus et al., 2013a) and in winter (Miller et al., 2011; Fransson et al., 2013) while uptake of $CO_2$ by sea
ice from the atmosphere has been reported after sea ice melt onset (e.g. Semiletov et al., 2004; Nomura et
al., 2010; Geilfus et al., 2012; Nomura et al., 2013; Fransson et al., 2013; Geilfus et al., 2014; 2015). In
combination, these works suggest that the temporal cycle of sea ice formation and melt affects
atmospheric $CO_2$ uptake by the ocean in variable ways. Sea ice may also act as an important control on
the partial pressure of $CO_2$ ($p$CO$_2$) in the sea surface through a sea ice pump (Rysgaard et al., 2007).
During the earliest stages of sea ice formation, a small fraction of $CO_2$-supersaturated brine is expelled
upward onto the ice surface promoting a release of $CO_2$ to the atmosphere (Geilfus et al., 2013a). As sea
ice forms and grows thicker, salts are partly rejected from the sea ice to the underlying seawater and
partly trapped within the sea ice structure, concentrated in brine pockets, tubes and channels. As a result,
the concentration of dissolved salts, including inorganic carbon, increase within the brine and promote the
precipitation of calcium carbonate crystals such as ikaite ($CaCO_3 \bullet 6H_2O$) (Marion 2001). These crystals
have been reported in both natural (Dieckmann et al., 2008; Nomura et al., 2013, Søgaard et al., 2013)
and experimental sea ice (Geilfus et al., 2013b; Rysgaard et al., 2014) and have been suggested to be a
key component of the carbonate system (Rysgaard et al., 2007; Fransson et al., 2013; Delille et al., 2014).
During ikaite precipitation within sea ice, TA in brine is reduced by 2 moles due to the reduction of
bicarbonate ($HCO_3^-$) while $TCO_2$ in brine is only reduced by 1 mole (equation 1 to 3).
$$Ca^{2+} + 2HCO_3^- + 5H_2O \rightleftharpoons CaCO_3 \cdot 6H_2O + CO_2 \qquad (1)$$
$$TCO_2 = [HCO_3^-] + [CO_3^{2-}] + [CO_2] \qquad (2)$$
$$TA = [HCO_3^-] + 2[CO_3^{2-}] + [B(OH)_4^-] + [OH^-] - [H^+] \qquad (3)$$
The specific conditions leading to ikaite precipitation as well as the fate of these precipitates in sea ice are
still not fully understood. Ikaite crystals may remain within the ice structure while the $CO_2$ formed during
their precipitation is likely rejected with dense brine to the underlying seawater and sequestered below the
mixed layer. During sea ice melt, the dissolution of these crystals triggered by increased ice temperatures
and decreased bulk ice salinity will consume $CO_2$ and drive a $CO_2$ uptake from the atmosphere to the ice.
Such mechanism could be an effective sea ice pump of atmospheric $CO_2$ (Delille et al., 2014). In
addition, ikaite stored in the ice matrix could become a source of TA to the near-surface ocean upon its
subsequent dissolution during sea ice melt (Rysgaard et al., 2007; 2009).
The main air-sea fluxes of $CO_2$ and $TCO_2$ are driven by brine rejection to the underlying seawater and
its contribution to intermediate and deep-water formation (Semiletov et al., 2004; Rysgaard et al., 2007,
2009; Fransson et al., 2013) or below sea ice in ice tank studies (e.g. Killawee et al., 1998 and
Papadimitriou et al., 2004). As sea ice thickens, reduced near-surface ice temperatures result in reduced
brine volume content, increased brine salinity and increased solute concentration in the brine. In the
spring-summer, as the ice temperature increases, sea ice brine volume increases and sea ice becomes
vertically permeable to liquid (Golden et al., 2007), enhancing the potential $CO_2$ exchange between the
atmosphere, sea ice and ocean. Eventually internal ice melt promotes brine dilution, which decreases
brine salinity, TA, $TCO_2$, and leads to lower $pCO_2$ in the brine. In addition, the dissolution of ikaite
decreases brine $pCO_2$ (Eq. 1) (Geilfus et al., 2012; 2015). These conditions all favour sea ice as a sink for
atmospheric $CO_2$ (Nomura et al., 2010; Geilfus et al., 2012; Nomura et al., 2013; Geilfus et al., 2015).
Melting sea ice stratifies surface seawater leading to decreased TA, $TCO_2$ and $pCO_2$, in the sea surface,
enhancing air-sea $CO_2$ fluxes (Rysgaard et al., 2007; 2009).
Although we now have a basic understanding of the key mechanisms of carbon cycling in sea ice,
significant unknowns remain. One of the major unknowns is the fate of ikaite, $T\mathrm{CO_2}$ and $\mathrm{CO_2}$ released
from sea ice during winter. It is unclear what proportion of precipitated ikaite crystals in sea ice remain in
the matrix to be released upon melt or what proportion are expelled with brine drainage during ice
formation and growth. Examining the chemical signatures of the water column beneath sea ice may
provide an indication of the importance of the different processes. However, the signal of carbon
components released from 1-2 meters of sea ice growth is difficult to detect in a water column several
hundred meters deep.
In this study, we followed the evolution of the inorganic carbon dynamics within experimental sea ice
from sea ice formation to melt in a sea ice-seawater mesocosm pool ($\sim$435 m$^3$). The benefits of this type
of environment are multiple. An artificial pool equipped with a movable bridge makes it possible to
collect undisturbed samples from thin growing sea ice. We gain the ability to carefully track carbonate
parameters in the ice, in the atmosphere, and in the underlying seawater, while growing sea ice in a large
volume of seawater, so that conditions closely mimic the natural system. During this experiment, we
examined physical and chemical processes, in the absence of biology, responsible for changes in the
inorganic carbon system of sea ice and the underlying seawater and quantify fluxes of inorganic carbon
between the atmosphere, sea ice and the water column. We also discuss that dissolution of ikaite crystals
exported from sea ice in the underlying seawater can potentially hamper the effect of oceanic
acidification on $\Omega_{\mathrm{aragonite}}$.
**3.   Site description, sampling and analysis**
The Sea-ice Environmental Research Facility (SERF) is an in-ground outdoor concrete pool of 18.3 m
by 9.1 m in surface area and 2.6 m deep exposed to ambient temperatures, winds and solar radiation (by
retracting its roof, Fig. 1). The weather conditions in the region are conducive to sea ice growth for
several months every winter. Prior to the experiment, the pool is filled with artificial seawater (ASW)
made by dissolving large quantities of various rock salts into local groundwater to mimic the major
composition of natural seawater (see Rysgaard et al., (2014) for exact composition of the ASW). Sea ice
is melted in the pool by circulating heated ethylene glycol through a closed-loop hose located at the
bottom of the pool, allowing successive ice growth/melt experiments to be carried out during one winter.
The experimental sea ice and brine exhibit similar physical and chemical properties to those observed in
natural Arctic sea ice (Geilfus et al., 2013; Hare et al., 2013). The experiment described herein was
initiated from open water conditions on 11 January 2013 when the heater was turned off. Sea ice grew
until 26 January when the heat was turned back on. The experiment ended on 30 January when the pool
was 20% ice-free.
Four 375 W pumps were installed on the bottom of the pool near each of the corners to induce a
consistent current. The pumps were configured to draw water from their base and then propel it outward
parallel to the bottom of the pool. The pumps were oriented successively at right angles to one another,
which created a counterclockwise circulation of 2-3 cm s$^{-1}$ (Else et al., 2015).
Bulk ice and seawater temperatures were recorded by an automated type-T thermocouple array fixed
vertically in the pool. Seawater salinity was measured continuously using Aanderaa CT sensors (model
4319) located at 30, 100, 175 and 245 cm depth. The in situ seawater $p$CO$_2$ was measured every 5 sec
using a Contros HydroC (resolution < 1 µatm, accuracy ± 1% of the upper range value) located at 1.3 m
depth.
Air temperature and relative humidity were measured using a Vaisala HMP45C probe at a
meteorological station located 2 m above the sea ice surface. Solar irradiance was continuously recorded
by an Eppley Precision Spectral Pyranometer (range=0.285–2.8 µm) mounted 10 m above the sea ice
surface. In addition, estimated photosynthetically active radiation (PAR) values at the ice bottom were
recorded with Alec mkv-L PAR sensors throughout the study and ranged from 0 to 892 µmol photons m$^{-2}$
s$^{-1}$.
Sea ice and seawater samples were obtained from a confined area located on the North side of the
pool to minimize effects on other experiments (e.g. Else et al., 2015). Ice samples were collected using
ceramic knives or a Kovacs Mark II coring system depending on the ice thickness. Sampling was
performed from a movable bridge to avoid walking on the ice surface and to ensure only undisturbed sites
were sampled. Ice cores were collected from one end of the pool (half meter away from the edge of the
pool) and at least 20 cm away from previous cored sites. Ice cores were packed in clean plastic bags and
kept frozen during the 20 minutes transport to a cold laboratory and processed within a few hours.
Seawater was sampled for total alkalinity (TA) and total dissolved inorganic carbon ($T$CO$_2$) with a
peristaltic pump (Cole Palmer, Masterflex-Environment sampler, equipped with PTFE tubing) through an
ice core hole the ice-water interface, at 1.25 m, and 2.5 m depth. Samples were stored in 12 ml gas-tight
vials (Exetainer, Labco High Wycombe, UK) and poisoned with 12 µl of saturated $HgCl_2$ solution and
stored in the dark at 4°C until analysed.
Air-ice $CO_2$ fluxes were measured using a Li-Cor 8100-103 chamber associated with a LI-8100A soil
$CO_2$ flux systems. The chamber was connected in a closed loop to the IRGA with an air pump rate of 3 L
$min^{-1}$. The measurement of $pCO_2$ in the chamber was recorded every sec over a 15 minute period. The
flux was computed from the slope of the linear regression of $pCO_2$ against time ($r^2 > 0.99$) according to
Frankignoulle (1988), taking into account the volume of ice or snow enclosed within the chamber. The
uncertainty of the flux computation due to the standard error on the regression slope was on average ±3%.
In the cold laboratory, sea ice cores were cut into 2 cm sections using a pre-cleaned stainless steel
band saw. Each section was placed in a gas-tight laminated (Nylon, ethylene vinyl alcohol and
polyethylene) plastic bag (Hansen et al., 2000) fitted with a gastight Tygon tube and a valve for sampling.
The plastic bag was sealed immediately and excess air was gently removed through the valve using a
vacuum pump. The bagged sea ice samples were then melted in the dark at 4°C to minimize the
dissolution of calcium carbonate precipitates (meltwater temperature never rose significantly above 0°C).
Once melted, the meltwater mixture and bubbles were transferred to gas-tight vials (12 ml Exetainer,
Labco High Wycombe, UK), poisoned with 12 µl solution of saturated $HgCl_2$ and stored in the dark at
4°C until analysed.
Bulk ice and seawater salinities were measured using a Thermo Orion 3-star with an Orion
013610MD conductivity cell and values were converted to bulk salinity (Grasshoff et al., 1983). TA was
determined by potentiometric titration (Haraldsson et al., 1997) while $TCO_2$ was measured on a
coulometer (Johnson et al., 1987). Routine analysis of Certified Reference Materials provided by A. G.
Dickson, Scripps Institution of Oceanography, verified that TA and $TCO_2$ were analyzed within ±3 and
±2 µmol $kg^{-1}$, respectively. Brine volume was estimated from measurements of bulk salinity, temperature
and density according to Cox and Weeks (1983) for temperatures below -2°C and according to
Leppäranta and Manninen (1988) for ice temperatures within the range -2 to 0°C.
Bulk ice samples for biological measurements were collected between 14 and 21 January. Filtered
(0.2 µm) SERF seawater (FSW) was added at a ratio of 3 parts FSW to 1 part ice and the samples were
left to melt in the dark. Chlorophyll *a* was determined on three occasions by filtering two aliquots of the
melted ice sample onto GF/F filters (Whatmann brand) and extracting pigments in 10 ml of 90% acetone
for 24 h. Fluorescence was measured before and after the addition of 5% HCl (Turner Designs
Fluorometer) and Chl *a* concentration was calculated following Parsons et al. (1984). Measurements of
bacterial production were done four times during the biological sampling period by incubating 6-10 ml
subsamples of the ice-FSW solution with $^3$H-leucine (final concentration of 10 nM) for 3h at 0°C in
darkness (Kirchmann, 2001). Half of the samples were spiked with trichloroacetic acid (TCA, final
concentration 5%) as controls prior to the incubation, while the remaining active subsamples were fixed
with TCA (final concentration 5%) after incubation. Following the incubation, vials were placed in 80°C
water for 15 minutes (Garneau et al., 2006) before filtration through 0.2 μm cellulose acetate membranes
(Whatmann brand) and rinsing with 5% TCA and 95% ethanol. Filters were dried and dissolved in
scintillation vials by adding 1 ml ethyl acetate, and radioactivity was measured on a liquid scintillation
counter after an extraction period of 24 h. Bacterial production was calculated using the equations of
Kirchman (1993) and a conversion factor of 1.5 kg C mol$^{-1}$ (Ducklow et al., 2003).
## 4. Results
### 4.1. Sea ice and seawater physical conditions
Sea ice was grown in the pool from open water on 13 January 2013 and reached a maximum thickness
of 24 cm on 26 January at which point the heat at the base of the pool was turned on. On 30 January the
experiment ended with the pool 20% ice-free. Three main snowfall events occurred during the
experiment. The first, from 14 to 15 January, covered the sea ice surface with 1 cm of snow. The second,
from 18 to 23 January, deposited 6-9 cm of snow over the entire pool. On the morning of 23 January, the
snow was manually cleared off the ice surface to investigate the insulating effect of the snow on the ice
temperature and ikaite precipitation (Rysgaard et al., 2014). Finally, from noon on 24 January to 27
January, 8 cm of snow covered the entire pool until the end of the experiment on 30 January.
The air temperature at the beginning of the experiment ranged from -2°C to -26°C, which initiated
rapid sea ice growth to 15 cm until 18 January (Fig. 2). During this initial sea ice growth, the sea ice was
attached to the side of the pool resulting in the development of a hydrostatic pressure head that caused
percolation of seawater at the freezing point upwards through the sea ice volume as the sea ice grew
downwards. This resulted in repeated events of increased sea ice temperature from the bottom to the
surface observed between 15 and 18 January (Fig. 2). Subsequently, the ice was cut using an ice saw
around the perimeter, allowing the ice to float and a pressure release valve was installed to prevent such
events (Rysgaard et al., 2014). During this period, the ice temperature oscillated between relatively warm
(~ -3°C) and cold (~ -7°C) phases. Brine volume content (0.047) was low in the middle part of the ice
cover, close to the permeability threshold of 0.05 as suggested by Golden et al., (2007). The bulk ice
salinity profiles were typically C-shaped with values ranging from 6 to 23 (Fig. 2). The underlying
seawater salinity increased rapidly due to sea ice growth. From 18 to 23 January, the 9 cm snow cover
insulated the ice cover from the cold atmosphere (Rysgaard et al., 2014), resulting in a fairly constant ice
thickness, nearly no change in ice temperature and salinity, a brine volume content above the
permeability threshold and a small increase in the underlying seawater salinity. Once the ice surface was
cleared of snow on the morning of 23 January, the ice temperature decreased throughout the entire ice
thickness and the ice surface salinity increased. The sea ice volume cooled from the top downwards, and
the brine volume content decreased below the permeability threshold on 23 January and rapid sea ice
growth rapidly increased the seawater salinity. Shortly after the snow clearing, the last snowfall event
covered the ice surface with 8 cm of snow, reducing the effect of the cold atmosphere on the ice cover.
On 26 January, the heater was activated to initiate sea ice melt. Sea ice temperatures increased and
became isothermal around -2°C while the bulk ice salinity decreased and the brine volume content
increased up to 0.13. The sea ice melt decreased the seawater salinity. The pool was well mixed during
the whole growth phase with similar salinity and temperature observed at the four depths. However, once
the heat was turned on, the pool become stratified with respect to salinity changes, as the salinity at 30 cm
depth started to diverge from the deeper depths (Fig. 2).
**4.2. Carbonate system**
TA and $TCO_2$ in seawater, noted as $TA_{(sw)}$ and $TCO_{2(sw)}$, were sampled at the sea ice-seawater
interface, 1.25 and 2.5 m depth. An ANOVA test over the 3 depths revealed that the means are not
statistically different (p<0.01) so we consider the average concentration of the three depths in the
following analysis. During sea ice growth, $TA_{(sw)}$ increased from 2449 to 2644 µmol kg$^{-1}$ (black line, Fig.
3a) while $TCO_{2(sw)}$ increased from 2347 to 2516 µmol kg$^{-1}$ (black line, Fig. 3b). Once the ice started to
melt, $TA_{(sw)}$ decreased to 2607 µmol kg$^{-1}$ and $TCO_{2(sw)}$ decreased to 2461 µmol kg$^{-1}$. As the experiment
stopped before the ice was completely melted in the tank, both the seawater salinity and $TA_{(sw)}$ do not
reach their initial values at the end of the experiment (Table 1, Fig 2 and 3). To discard the effect of
salinity changes, we normalized $TA_{(sw)}$ and $TCO_{2(sw)}$ to a salinity of 33 (noted as $nTA_{(sw)}$ and $nTCO_{2(sw)}$)
according to the equations 4 and 5:
$$nTA_{(sample)\,t} = \frac{TA_{(sample)\,t}}{S_{(sample)\,t}} \times 33 \qquad (4)$$

$$nTCO_{2\,(sample)\,t} = \frac{TCO_{2\,(sample)\,t}}{S_{(sample)\,t}} \times 33 \qquad (5)$$

where $t$ is the time of the sampling and S the salinity of the sample (seawater or sea ice). During ice
growth, $nTA_{(sw)}$ and $nTCO_{2(sw)}$ increased slightly to 2446 and 2328 µmol kg$^{-1}$, respectively (Fig. 3c).
However, once the ice started to melt, $nTA_{(sw)}$ increased to 2546 µmol kg$^{-1}$ and $nTCO_{2(sw)}$ increased to
2404 µmol kg$^{-1}$.
The in situ $pCO_2$ of the underlying seawater ($pCO_{2(sw)}$) decreased from 377 to 360 µatm as the
seawater temperature in the pool decreased to the freezing point. The $pCO_{2(sw)}$ then oscillated from 360 to
365 µatm during sea ice growth. One day after the heater was turned on, the $pCO_{2(sw)}$ increased to a
similar concentration as at the beginning of the experiment before decreasing to 373 µatm by the end of
the experiment (Fig. 3d).
Within bulk sea ice, TA$_{(ice)}$ ranged from 300 to 1907 µmol kg$^{-1}$ while $TCO_{2(ice)}$ ranged from 237 to
1685 µmol kg$^{-1}$. Both TA$_{(ice)}$ and $TCO_{2(ice)}$ exhibited C-shaped profiles with higher concentrations at the
surface and bottom layers of the ice cover (Fig. 4). The concentration of TA$_{(ice)}$ (average = 476 µmol kg$^{-1}$)
and $TCO_{2(ice)}$ (average = 408 µmol kg$^{-1}$) did not show significant variability during our survey, except at
the surface of the ice. A first maximum was observed on 17 January with concentration of 1907 µmol kg$^{-1}$
for TA and 1685 µmol kg$^{-1}$ for $TCO_2$. A second maximum was observed on 23 January with
concentration of 1433 µmol kg$^{-1}$ for TA and 861 µmol kg$^{-1}$ for $TCO_2$. These maxima matched the high
bulk ice salinity (Fig. 2), so we also normalized TA$_{(ice)}$ and $TCO_{2(ice)}$ (noted as $nTA_{(ice)}$ and $nTCO_{2(ice)}$, Fig.
4) to a salinity of 33 (according to the equations 4 and 5) to discard the effect of salinity changes and
facilitate comparison with the underlying seawater. During initial sea ice formation (up to 17 January),
the concentration of both $nTA_{(ice)}$ (from 1083 to 2741, average = 1939 µmol kg$^{-1}$) and $nTCO_{2(ice)}$ (from
853 to 2440, average = 1596 µmol kg$^{-1}$) were at their minima in the experimental time series. From 17 to
21 January, both $nTA_{(ice)}$ and $nTCO_{2(ice)}$ increased throughout the ice column (average $nTA_{(ice)}$ = 2375
µmol kg$^{-1}$ and $nTCO_{2(ice)}$ = 2117 µmol kg$^{-1}$). However, from 21 January until the initial sea ice melt,
$nTA_{(ice)}$ and $nTCO_{2(ice)}$ decreased in the top 5 cm of the ice cover (average $nTA_{(ice)}$ = 2125 µmol kg$^{-1}$ and
$nTCO_{2(ice)}$ = 1635 µmol kg$^{-1}$).
**4.3. Air-ice CO$_2$ fluxes**
The $CO_2$ fluxes measured at the variably snow-covered sea ice surface (Fig. 2b), ranged from 0.29 to
4.43 mmol $m^{-2}$ $d^{-1}$ show that growing sea ice released $CO_2$ to the atmosphere (Fig. 5). However, as soon
as the ice started to warm up and then melt, the sea ice switched from source to sink for atmospheric $CO_2$
with downward fluxes from -1.3 to -2.8 mmol $m^{-2}$ $d^{-1}$. These ranges of air-ice $CO_2$ exchanges are of the
same order of magnitude as fluxes reported on natural sea ice using the same chamber technique in the
Arctic during the initial sea ice growth (from 4.2 to 9.9 mmol $m^{-2}$ $d^{-1}$ in Geilfus et al., 2013) and during
the spring-summer transition (from -1.4 to -5.4 mmol $m^{-2}$ $d^{-1}$ in Geilfus et al., 2015). In Antarctica air-ice
$CO_2$ fluxes were reported during the spring-summer transition from 1.9 to -5.2 mmol $m^{-2}$ $d^{-1}$ by Delille et
al (2014), from 0.3 to -2.9 mmol $m^{-2}$ $d^{-1}$ (Geilfus et al., 2014) and from 0.5 to -4 mmol $m^{-2}$ $d^{-1}$ (Nomura et
al., 2013).
**5. Discussion**
**5.1. Key processes affecting the carbonate system**
The dynamics of inorganic carbon in the ocean and sea ice are mainly affected by temperature and
salinity changes, precipitation and dissolution of calcium carbonate, and biological activities (Zeebe and
Wolf-Gladrow, 2001). During this experiment, neither organic matter nor biota were purposely
introduced into the pool; the observed range of bulk ice microbial activity (5.7 x $10^{-9}$ on 14 January to 7.5
x $10^{-7}$ g C $L^{-1}$ $h^{-1}$ on 21 January) and algal Chl $a$ (0.008 on 14 January to 0.002 µg $L^{-1}$ on 21 January) were
too low to support any biological activity (Rysgaard et al., 2014). Therefore biological activity is unlikely
to have played a role. During the same 2013 time series at SERF, Rysgaard et al. (2014) discussed the
precipitation of ikaite within the ice cover in detail, reporting high concentrations of ikaite (> 2000 µmol
$kg^{-1}$) at the surface of the ice in brine skim and frost flowers and ikaite precipitation up to 350 µmol $kg^{-1}$
within bulk sea ice. Within sea ice, ikaite precipitation is associated with low ice temperatures, high bulk
salinity and high $TA_{(ice)}$ and $TCO_{2(ice)}$ concentrations (Fig. 2 and 3).
The main processes affecting the carbonate system can be described by changes in TA and $TCO_2$
(Zeebe and Wolf-Gladrow, 2001). An exchange of $CO_{2(gas)}$ affects $TCO_2$ while TA remains constant and
the precipitation-dissolution of calcium carbonate affects both TA and $TCO_2$ in a ratio of 2:1 (see
equation 1 to 3, Fig. 6). To calculate the theoretical changes in TA and $TCO_2$ during the course of the
experiment, we used seawater samples from 11 January prior to sea ice formation (t=0, Table 1) as the
origin point (blue circle on Fig. 6). Sea ice data are located between the theoretical calcium carbonate
precipitation line and the $CO_2$ release line (Fig. 6a) while seawater data mainly fall on the calcium
carbonate dissolution line (Fig. 6b), suggesting that the carbonate system within sea ice is affected by
both the precipitation of ikaite and a release of $CO_{2(gas)}$ while the underlying seawater is mainly affected
by the dissolution of calcium carbonate.

### 5.2. Estimation of the precipitation-dissolution of ikaite

During the experiment, Rysgaard et al., (2014) observed ikaite within sea ice using direct microscopic
observations. The precipitation-dissolution of ikaite and gas exchange are the only two processes taking
place during the experiment. As illustrated in Fig. 6, an exchange of $CO_2$ does not affect TA while the
precipitation-dissolution of ikaite affects TA and $TCO_2$ in a ratio 2:1. Therefore, we use TA to estimate
how much ikaite is precipitated or dissolved within the ice cover and the underlying seawater.
Assuming no biological effect, ikaite precipitation/dissolution and gas exchange, TA and $TCO_2$ are
considered conservative with salinity. Therefore, we can calculate the expected TA and $TCO_2$ (noted as
$TA_{(ice)}{}^*$ and $TCO_{2(ice)}{}^*$ in the ice cover and $TA_{(sw)}{}^*$, $TCO_{2(sw)}{}^*$ for the water column) based on the initial
seawater conditions ($TA_{(sw)}$, $TCO_{2(sw)}$ and $S_{(sw)}$ at t=0, Table 1) and the sample salinity (bulk sea ice or
seawater) measured during the experiment:
$$TA^*_{(sample)\,t} = \frac{TA_{(sw)\,t=0}}{S_{(sw)\,t=0}} \times S_{(sample)\,t} \qquad\qquad (6)$$

$$TCO^*_{2\,(sample)\,t} = \frac{TCO_{2\,(sw)\,t=0}}{S_{(sw)\,t=0}} \times S_{(sample)\,t} \qquad\qquad (7)$$

where t is the time of the sampling. Within the ice cover, $TA_{(ice)}$, $TCO_{2(ice)}$, and the bulk ice salinity are
averaged throughout the ice column at each sampling day (Fig. 7a, b, black line) while for the underlying
seawater, we used the averaged $TA_{(sw)}$, $TCO_{2(sw)}$ and salinity for all the measured depths (Fig. 2a, b, black
line). The difference between $TA_{(sample)}{}^*$ and the observed TA is only due to the precipitation or
dissolution of ikaite crystals. In case of ikaite precipitation (*i.e.* $TA_{(sample)}{}^* > TA_{(sample)}$), half of this
positive difference corresponds to the amount of ikaite precipitated within the ice. This ikaite may either
remain or may be exported out of the ice. A negative difference (*i.e.* $TA_{(sample)}{}^* < TA_{(sample)}$), indicates
ikaite dissolution.

### 5.2.1. Sea ice

Greater $TA_{(ice)}{}^*$ and $TCO_{2(ice)}{}^*$ compared to the averaged observed $TA_{(ice)}$ and $TCO_{2(ice)}$ (Fig. 7a, b) are
expected as ikaite is precipitated and $CO_2$ released from the ice to the atmosphere (Fig. 5, 6). Half the
difference between $TA_{(ice)}{}^*$ and $TA_{(ice)}$ is a result of ikaite precipitation (Fig. 7c, black diamonds). Highly
variable ikaite precipitation was observed (Fig. 7c). Ikaite precipitation was up to 167 µmol kg$^{-1}$ (e.g. first
days of the experiment) and as low as 1 µmol kg$^{-1}$ (e.g. 19 January). A negative difference between
TA$_{(ice)}$$^{*}$ and TA$_{(ice)}$ (*i.e.* ikaite dissolution) occurred on three occasions: 14, 20 and after the 26 January
(beginning of the sea ice melt). On these occasions, the ice cover was relatively warm due to warmer
atmospheric temperatures (14 January), thicker snow cover insulating the ice cover from the cold
atmosphere (20 January) or when heat was turned back on (after 26 January, Fig. 2). Relatively high sea
ice temperatures likely promote ikaite dissolution in agreement with Rysgaard et al., (2014) who linked
ikaite precipitation/dissolution to ice temperature. The upward percolation of seawater observed from 15
to 18 January might complicate the effect of sea ice temperature on ikaite formation because it was in part
responsible for increased ice temperatures (Fig. 2b) and therefore increased the sea ice brine volumes
(Fig. 2c). Increased vertical connectivity (permeability) of the network of liquid inclusions throughout the
sea ice (Golden et al., 2007; Galley et al., 2015) would have allowed the export of ikaite crystals from the
ice cover to the underlying seawater. However, while we calculated a negative difference between TA$_{(ice)}$$^{*}$
and TA$_{(ice)}$, ikaite crystals were observed by Rysgaard et al., (2014). We compared the direct microscopy
observations by averaging the amount of ikaite precipitated throughout the ice thickness for each
sampling day from Rysgaard et al., (2014) (Fig. 7c, white dots) with our estimation of the amount of
ikaite based on the difference between TA$_{(ice)}$$^{*}$ and TA$_{(ice)}$ (Fig. 7c, black diamonds). Both ikaite
measurements are of the same order of magnitude however the average (22 µmol kg$^{-1}$) and maximum
(100 µmol kg$^{-1}$) of direct observations presented by Rysgaard et al. (2014) were lower than our estimated
average (40 µmol kg$^{-1}$) and maximum of up to 167 µmol kg$^{-1}$ over this whole experiment. Deviations are
likely due to methodological differences. Here, sea ice samples were melted to subsample for TA and
$T$CO$_2$, Ikaite crystals may have dissolved during melting, leading to an underestimation of the total
amount of ikaite precipitated in the ice. However, the difference between TA$_{(ice)}$$^{*}$ and TA$_{(ice)}$ provides an
estimation of how much ikaite is precipitated in the ice cover, including those crystals potentially already
exported to the underlying seawater. The method used by Rysgaard et al., (2014) avoid the bias of ikaite
dissolution during sea ice melt with the caveat that crystals need to be large enough to be optically
detected. If no crystals were observed, Rysgaard et al., (2014) assumed that no crystals were precipitated
in the ice, though ikaite crystals could have been formed and then exported into the underlying seawater
prior to microscopic observation of the sample, which may explain the difference observed between both
methods during initial sea ice formation (15-18 January) when the ice was still very thin. In addition, the
succession of upward percolation events could have facilitated the ikaite export from the ice cover to the
underlying seawater. Estimations from both methods show similar concentrations when the ice (i)
warmed due to snowfall (18-23 January) and (ii) cooled once the snow was removed (on 23 January).
Once the ice started to melt (26 January), Rysgaard et al., (2014) reported a decrease in the ikaite
precipitation while in this study we reported a negative difference between $TA_{(ice)}^{*}$ and $TA_{(ice)}$, possibly
indicating that ikaite dissolved in the ice.

### 5.2.2. Water column

The main process affecting the carbonate system in the underlying seawater in this study is the export
of ikaite from the ice and its dissolution in the water column (Fig. 6). While a few studies of ikaite
precipitation within sea ice carried out over open ocean hypothesized that ikaite remained trapped within
the sea ice matrix (Rysgaard et al., 2007; 2013; Delille et al., 2014), the observed increase of $nTA_{(sw)}$ (Fig.
3) suggests that ikaite precipitated within the ice cover was exported to the underlying seawater where the
crystals were dissolved as suggested by Fransson et al., (2013). Lower $TA_{(sw)}^{*}$ and $TCO_{2(sw)}^{*}$ compared to
$TA_{(sw)}$ and $TCO_{2(sw)}$ (Fig. 3) confirm the dissolution of ikaite in the underlying seawater as the dissolution
of ikaite crystals will decrease both TA and $TCO_2$ (equations 1 to 3). Therefore, half the difference
between $TA_{(sw)}^{*}$ and $TA_{(sw)}$ corresponds to the concentration of ikaite exported from the ice and dissolved
in the underlying seawater (Fig. 8a). This concentration increased over time to a maximum of 66 µmol
$kg^{-1}$.
During this experiment, $nTA_{(sw)}$ increased by 128 µmol $kg^{-1}$ while $nTCO_{2(sw)}$ increased by 82 µmol $kg^{-1}$
(Fig. 3c). This suggests that 64 µmol $kg^{-1}$ of ikaite are dissolved compared to the 66 µmol $kg^{-1}$ estimated
from the difference between $TA_{(sw)}^{*}$ and $TA_{(sw)}$. As a result of the effect of ikaite dissolution on the 2:1
ratio of $TA:TCO_2$, the dissolution of ikaite accounts for the entire increase of $nTA_{(sw)}$ but only accounts
for 64-66 µmol $kg^{-1}$ of the 82 µmol $kg^{-1}$ increase in $nTCO_{2(sw)}$. So, 16-18 µmol $kg^{-1}$ (about 25%) of the
increase of $nTCO_{2(sw)}$ cannot be explained by the dissolution of ikaite. The increase of both $nTA_{(sw)}$ and
$nTCO_{2(sw)}$ is more significant once the ice starts to melt (26 January). During sea ice melt, increased
vertical permeability resulting in increased liquid communication through the sea ice volume from below
likely in part dissolved ikaite crystals still residing in the ice at that time, and also will have created a
downward crystal export mechanism. As the ice melt advanced, patches of open water occurred at the
surface of the pool. Therefore, uptake of atmospheric $CO_2$ by the undersaturated seawater likely occurred,
increasing the $TCO_{2(sw)}$.

The dissolution of ikaite crystals could also have a strong impact on the $pCO_{2(sw)}$. The water column

was undersaturated compared to the atmosphere during the whole experiment (Fig. 3d). A release of $CO_2$,

from the ice to the atmosphere was measured during sea ice growth (Fig. 5) in spite of the undersaturated

$pCO_{2(sw)}$. This suggests that air-ice $CO_2$ fluxes are only due to the concentration gradient between the ice

and the atmosphere (Geilfus et al., 2012; Nomura et al., 2013) but that sea ice exchanges $CO_2$ with the

atmosphere independently of the seawater concentration (Geilfus et al., 2014). The $pCO_{2(sw)}$ is highly

correlated with the seawater temperature (Fig. 2) with a rapid decrease of $pCO_{2(sw)}$ during the first days of

the experiment (13 to 15 January) and a relative constant $pCO_{2(sw)}$ until 27 January. However, on 26

January, the heat was turned back on affecting the seawater temperature on the same day (Fig. 2) while

the impact of increasing temperature on the $pCO_{2(sw)}$ appeared one day later (Fig. 3d). We normalized the

$pCO_{2(sw)}$ to a temperature of -1°C (after Copin-Montegut (1988), noted as $npCO_{2(sw)}$, blue line on Fig. 3d).

The $npCO_{2(sw)}$, does not show major variations during sea ice growth with values around 380 µatm.

However, once the heat is turned on and the seawater temperature increased (on 26 January), $npCO_{2(sw)}$

decreased from 383 µatm to 365 µatm, while $pCO_{2(sw)}$ did not change in response to increased seawater

temperatures until 27 January, suggesting that a process other than temperature change affected the

$pCO_{2(sw)}$. According to equation 1, the dissolution of calcium carbonate has the potential to reduce

$pCO_{2(sw)}$. Therefore, during sea ice growth and the associated release of salt, TA, $TCO_2$ and ikaite crystals

to the underlying seawater, ikaite dissolution within the seawater could be responsible for maintaining

stable $pCO_{2(sw)}$ values while seawater salinity, $TA_{(sw)}$ and $TCO_{2(sw)}$ are increasing. Once the seawater

temperature increased (26 January), sea ice melt likely released ikaite crystals to the underlying seawater

(Fig. 2, 8a) along with brine and meltwater, a process that would continuously export ikaite from the sea

ice as the volume interacting with the seawater via percolation or convection increased. The dissolution of

these crystals likely contributed to keeping the $pCO_{2(sw)}$ low and counterbalancing the effect of increased

temperature. We argued that once all the ikaite crystals are dissolved, the increase seawater temperature

increased the $pCO_{2(sw)}$ simultaneously with the $npCO_{2(sw)}$ (27 January, Fig. 3).

### 5.3. Ikaite export from the ice cover to the water column

We estimated the amount of ikaite precipitated and dissolved within sea ice and seawater based on the

sea ice (and seawater) volume (in $m^3$), the sea ice and seawater density, the concentration of ikaite

precipitated and dissolved within the ice cover (Fig. 7c), and the concentration of ikaite dissolved in the

water column (Fig. 8a). Within the ice cover, the amount of ikaite precipitated-dissolved ranged from -0.7

to 1.97 mol (Fig 8b, Table 2), with a maximum just after the snow was cleared on 23 January. In the
underlying seawater, the amount of ikaite dissolved in the pool increased from 0.47 mol on the first day
of the experiment to 11.5 mol on 25 January when sea ice growth ceased. Once the ice started to melt the
amount of dissolved ikaite increased up to 20.9 (28 Jan) and 26.7 mol (29 January, Table 2). The
estimation of ikaite dissolution in the pool is significantly higher than the estimated amount of ikaite
precipitated (and potentially exported) within the ice cover, especially during sea ice melt. Within the ice
cover, the ikaite values presented here represent a snapshot of the ikaite content in the ice at the time of
sampling. In the underlying seawater, ikaite dissolution increased $TA_{(sw)}$ cumulatively over time.
The difference between $TA_{(ice)}^*$ and $TA_{(ice)}$ provides an estimation of ikaite precipitated within the ice,
including potential ikaite export to the underlying seawater, so it cannot be used to determine how much
ikaite remained in the ice versus how much dissolved in the water column. However, Rysgaard et al.,
(2014) indicate ikaite precipitated within the ice based on direct observations. Using the ikaite
concentration reported in Rysgaard et al (2014) (and shown in Fig. 7c), the sea ice volume (in $m^3$) and
density, we calculate that 0 to 3.05 mol of ikaite precipitated within the ice cover during sea ice growth
(Fig. 8b and Table 2). This amount decreased to 0.46 and 0.55 mol during the sea ice melt (28 and 29
January, respectively). Increased ikaite dissolution in the water column when the ice began to melt (from
11.5 to 20.9 mol) indicates that 9.4 mol of ikaite were stored in the ice and rejected upon the sea ice melt.
This amount is about three times the amount of ikaite precipitated in the ice estimated by Rysgaard et al.,
(2014) at the end of the growth phase (3.05 mol, Table 2), suggesting more work is needed best estimate
ikaite precipitation within sea ice.
Once the ice started to melt, the increased ikaite dissolution from 11.5 mol to 20.9 mol (28 January) and
to 26.7 mol (29 January) suggests that about the same amount of ikaite is dissolved during the sea ice
growth as during the first two days of the sea ice melt. The amount of ikaite dissolved in the water
column after melt commenced continued to increase cumulatively, suggesting that ikaite is continuously
exported to the underlying seawater as increased sea ice temperatures permit more of the volume to
communicate with the underlying seawater. Therefore, we can assume that more than half of the amount
of ikaite precipitated within the ice remained in the ice cover before ice melt began.
### 5.4. Air-ice-seawater exchange of inorganic carbon
SERF is a semi-closed system where the only way for the surface (water or sea ice) to gain or lose $CO_2$
is through exchange with the atmosphere, making it reasonable to track the exchange of $TCO_2$ in the
atmosphere-sea ice-seawater system. The ice cover always had lower $TCO_{2(ice)}$ during the experiment
($TCO_{2(ice)}^* > TCO_{2(ice)}$) compared to what would be expected if the $CO_2$ simply followed brine rejection in
a conservative process (i.e. $TCO_{2(ice)}^*$) (Fig. 7b). This could be due to: (i) $CO_2$ released to the atmosphere
from sea ice, (ii) decreased $TCO_{2(ice)}$ due to the precipitation of ikaite within sea ice and/or (iii) sea ice
exchanging $TCO_2$ with the underlying seawater.
The number of moles of $TCO_2$ exchanged during this experiment was calculated using the sea ice (and
seawater) volume (in $m^3$) and density (in $kg/m^3$). The total amount of $TCO_{2(ice)}$ lost from the ice cover (the
difference between $TCO_{2(ice)}^*$ and $TCO_{2(ice)}$) ranged from 0.11 to 6.02 mol (average 2.38 mol, Fig. 9, black
dots). The greatest sea ice $TCO_2$ losses occurred on 15-16 January during initial sea ice growth and from
23 to 25 January, during ice cooling due to snow removal. The exchange of $CO_2$ between the ice and the
atmosphere is known (Fig. 5). The number of mole of $CO_2$ exchanges between the ice and the atmosphere
were calculated (noted as $CO_{2(air-ice)}$ in Table 2) using the time step between each flux measurement, the
ice thickness and density. During sea ice growth 0.01 to 0.42 mol of $CO_2$ were released from the ice-
covered pool to the atmosphere. During sea ice melt uptake of atmospheric $CO_2$ by the ice-covered pool
ranged from -0.15 to -0.93 (Fig. 9, white triangles). On average, over the duration of the experiment, the
ice cover released 0.08 mol of $CO_2$ to the atmosphere. Assuming we know how much ikaite is contained
within the ice cover (Fig. 8b), we can estimate how much $TCO_2$ is exported from the ice to the underlying
seawater (Fig. 9, blue triangles) by subtracting the air-ice $CO_2$ exchange and the ikaite precipitation from
the total reduction of $TCO_{2(ice)}$ observed within the ice cover (Fig. 9, black dots). The sea ice-to-seawater
$TCO_2$ export ranged from 0.2 to 3.98 mol (average = 1.7 mol), confirming that sea ice primarily exports
$TCO_2$ to the underlying seawater. $TCO_2$ export from the ice to the water column ranged from 23% of the
total sea ice $TCO_2$ early in the ice growth (14 January) to 100% after the onset of melt. These estimations
are comparable to the study of Sejr et al., (2011) who suggested that sea ice exports 99% of its total $TCO_2$
to the seawater below it. On average over the whole experiment, sea ice exported 1.7 mol of $TCO_2$ to the
underlying seawater (Fig. 9), which corresponds to a $TCO_{2(sw)}$ increase of 43.5 $\mu mol\ kg^{-1}$ considering the
average sea ice thickness and density during the experiment and the volume of the pool. However,
$TCO_{2(sw)}$ increased by 115 $\mu mol\ kg^{-1}$ over the whole experiment (Fig. 3b), leaving an increase of 71.5
$\mu mol\ kg^{-1}$ in the $TCO_{2(sw)}$ that cannot be explained by the sea ice-seawater exchange of $TCO_2$. We
postulate that as the ice melt advanced, patches of open water that opened at the surface of the pool which
were undersaturated compared to the atmosphere (Fig. 3d) imported the additional $TCO_2$ directly from the
atmosphere in the form of $CO_{2(g)}$. Considering the pool volume, the 71.5 µmol kg$^{-1}$ increase of $TCO_{2(sw)}$
could be explained by an air-sea water $CO_2$ uptake of 8.5 mmol m$^{-2}$ d$^{-1}$ over 3 days of sea ice melt in a
20% ice free pool. High air-sea gas exchanges rates have been observed over partially ice-covered seas
(Else et al., 2011; 2013). This mechanism is also corroborated by models that account for additional
sources of turbulence generated by the presence of sea ice (Loose et al., 2014).
The design of the experiment allowed for constrained measurements of inorganic carbon fluxes
between sea ice and the water column not possible in a natural environment where large volume, mixing
processes alter the underlying seawater making it more complicated to identify changes. We build a $CO_2$
budget based only on the sea ice growth phase because only two days of data for the melt phase are
available and the experiment stopped while the pool was 20% ice-free (Rysgaard et al., 2014; Else et al.,
2015). The initial seawater (origin point, t=0) contained 1040.9 mol of $TCO_{2(sw)}$ on 11 January while on
the last day of sea ice growth (25 January) the seawater contained 1017.3 mol of $TCO_{2(sw)}$ (Table 2) with
the difference, (23.6 mol of $TCO_2$) in all likelihood transferred from the water column to the ice cover or
the atmosphere. However, the $TCO_2$ content within the ice cover at the end of the growing phase was 15.6
mol and the ice cover released 3.1 mol of $CO_2$ to the atmosphere (Table 2). Therefore, 4.9 mol of the 23.6
mol of $TCO_2$ exchanged from the water column are unaccounted for, but may be explained by air-ice $CO_2$
fluxes. The chamber measurement technique for air-ice $CO_2$ flux may underestimate the exchange of
$CO_2$, and the air-seawater $CO_2$ fluxes are unknown until the ice started to grow (13 January). These
missing moles of $TCO_2$ may also be explained by our assumption of uniform sea ice thickness in the
SERF. Using the seawater conditions at the end of the experiment, 1-cm of seawater in the pool contains
4.21 mol of $TCO_2$, making it difficult to close our budget.
**5.5. Potential impact of sea ice growth and ikaite export on aragonite saturation state of the**

**underlying seawater.**

The Arctic Ocean is a region where calcifying organisms are particularly vulnerable to ocean
acidification since low temperatures and low salinity lower the carbonate saturation state. As a result
several areas of the Arctic Ocean are already undersaturated with respect to aragonite (Chierici and
Fransson 2009; Yamamoto-Kawai et al., 2009; Bates et al., 2011). This undersaturation is enhanced in
winter as the temperature decreases and $pCO_2$ increases as a result of respiration. Calcifying organisms
might therefore be most susceptible to the effects of acidification in the winter, corresponding to the
annual minimum in aragonite saturation state ($\Omega_{aragonite}$). Sea ice retreat is thought to enhance the impact
of ocean acidification by freshening and ventilating the surface water (Yamamoto-Kawai et al., 2008;
Yamamoto et al., 2012; Popova et al., 2014). However, any understanding of the effect of ikaite
precipitation in sea ice on ocean acidification is still in its infancy (e.g. Fransson et al., 2013).
Since the discovery of ikaite precipitation in sea ice (Dieckmann et al., 2008), research on its impact on
the carbonate system of the underlying seawater has been ongoing. Depending on the timing and location
of this precipitation within sea ice, the impact for the atmosphere and the water column in terms of $CO_2$
transport can be significantly different (Delille et al., 2014). Dissolution of ikaite within melting sea ice in
the spring and export of this related high TA:$T$CO$_2$ ratio meltwater from the ice to the water column will
decrease the $p$CO$_2$, increase pH and $\Omega_{aragonite}$ of the surface layer seawater. Accordingly, during sea ice
melt, an increase of $\Omega_{aragonite}$ in the surface water in the Arctic was observed (Chierici et al., 2011,
Fransson et al., 2013, Bates et al., 2014). However, it was difficult to ascribe this increase to the legacy of
excess TA in sea ice, ikaite dissolution or primary production.
The impact of ikaite precipitation on the surface seawater during sea ice growth is less clear. Fransson
et al., (2013) suggested in winter in the Amundsen Gulf that the release of brine decreased $\Omega_{aragonite}$ by 0.8
at the sea ice-seawater interface as a result of ikaite precipitation within sea ice and the related $CO_2$
enrichment of brine. Conversely, during ice melt, $\Omega_{aragonite}$ increased by 1.4 between March and May,
likely due to both calcium carbonate dissolution and primary production. This contrasts with the present
experiment. Figure 10 shows the evolution of $\Omega_{aragonite}$ and pH in the water column derived from TA$_{(sw)}$
and $T$CO$_{2(sw)}$ and the evolution of $\Omega_{aragonite}$ and pH predicted solely from salinity changes (i.e. using
TA$_{(sw)}{}^{*}$ and $T$CO$_{2(sw)}{}^{*}$, noted as $\Omega_{aragonite}{}^{*}$ and pH$^{*}$). We used the CO2sys_v2.1.xls spreadsheet (Pierrot et
al., 2006) with the dissociation constants from Goyet and Dickson (1989) and all other constants from
DOE (1994). This shows the complexity of ikaite and its impact on the carbonate system and $\Omega$ in the
underlying water.
During ice growth, sea ice brine rejection appears to increase both pH (from 8.00 to 8.06) and $\Omega_{aragonite}$
(from 1.28 to 1.65) of the underlying seawater, offsetting the effect of decreased temperature. A slight
increase of $\Omega_{aragonite}$ was predicted due to increased salinity and a proportional increase of TA and $T$CO$_2$
as depicted in $\Omega_{aragonite}{}^{*}$. However, the effect of ikaite rejection and subsequent changes in TA strongly
enhance the increase of $\Omega_{aragonite}$. Therefore, ikaite rejection from sea ice has a much stronger potential to
increase $\Omega_{aragonite}$ than brine rejection during fall and winter sea ice growth, suggesting ikaite exported to
seawater from sea ice may hamper the effect of oceanic acidification on $\Omega_{aragonite}$ in fall and winter in at
the time when $\Omega_{aragonite}$ is at its minimum (Chierici and Fransson 2009, Yamamoto-Kawai et al., 2009,
Chierici et al., 2011). Ice formation may therefore delay harmful effects of ocean acidification on
calcifying organisms by increasing $\Omega_{aragonite}$ in the critical winter period when $\Omega_{aragonite}$ reaches its minimal
values. As a corollary, ice removal acts to alleviate the effect of ikaite rejection and may therefore lowers
$\Omega_{aragonite.}$ This calls for an accounting of under-ice ikaite rejection in modeling predictions on the
consequences of Arctic Ocean acidification in the context of northern hemispheric annual multi-year sea
ice loss, as increased summer open water will lead to more first year sea ice formation in fall and winter
in the future.
## 6. Conclusion
We quantified the evolution of inorganic carbon dynamics from initial sea ice formation to its melt in
a sea ice-seawater mesocosm pool from 11 to 29 January 2013. Based on our analysis of TA and $T$CO$_2$ in
sea ice and seawater, the main processes affecting inorganic carbon within sea ice are ikaite precipitation
and $CO_2$ exchange with the atmosphere, while in the underlying seawater dissolution of ikaite was the
main process affecting the inorganic carbon system.
During this experiment, sea ice exchanged inorganic carbon components (e.g. $CO_2$, ikaite, $T$CO$_2$) with
both the atmosphere and the underlying seawater. During sea ice growth, $CO_2$ was released to the
atmosphere while during ice melt an uptake of atmospheric $CO_2$ was observed. We report ikaite
precipitation up to 167 $\mu$mol kg$^{-1}$ of sea ice, similar to previous estimates from Rysgaard et al., (2014)
based on microscopically observed values. In the underlying seawater, a net increase of $n$TA$_{(sw)}$ over the
whole experiment was observed (up to 128 $\mu$mol kg$^{-1}$), suggesting that a portion of the ikaite crystals
precipitated within sea ice were exported to the underlying seawater and then dissolved as the ice cover
evolved in time. Ikaite export from ice to the underlying seawater was associated with brine rejection
during sea ice growth, increased sea ice vertical connectivity due to the upward percolation of seawater,
and meltwater flushing during sea ice melt. Rysgaard et al., (2007) suggested that ikaite precipitation
within sea ice could act as a significant sink for atmospheric $CO_2$, however to act as a sink for
atmospheric $CO_2$, ikaite crystals must remain in the ice structure while the $CO_2$ produced by their
precipitation is expelled with dense brine rejection and entrained in deep seawater (Delille et al., 2014).
TA changes observed in the water column once the sea ice started to melt indicate that more than half of
the total amount of ikaite precipitated in the ice during the sea ice growth remained in the ice until the sea
ice began to melt. Derivation of air-sea $CO_2$ fluxes related to the sea ice carbon pump should take into
account ikaite export to the underlying ocean during sea ice growth, which might reduce the efficiency of
oceanic $CO_2$ uptake upon sea ice melt. As sea ice melts, ikaite is flushed downward out of the ice along
with the meltwater.
Ikaite export from sea ice and its dissolution had a strong impact on the underlying seawater. In this
semi-closed system, sea ice growth increased the seawater salinity, $TA_{(sw)}$, and $TCO_{2(sw)}$. In spite of those
increases, the $pCO_2$ of the underlying seawater remained undersaturated compared to the atmosphere. We
conclude that ikaite dissolution within the water column is responsible for the seawaters' continual $pCO_2$
undersaturation. In addition, we discuss that dissolution of ikaite crystals exported from sea ice in the
underlying seawater can potentially hamper the effect of oceanic acidification on $\Omega_{aragonite}$ in fall and
winter in ice-covered areas at the time when $\Omega_{aragonite}$ is smallest.
**7.  Acknowledgments**
We gratefully acknowledge the contributions of the Canada Excellence Research Chair (CERC) and
Canada Research Chair (CRC) programs.  Support was also provided by the Natural Sciences and
Engineering Research Council (NSERC), the Canada Foundation for Innovation and the University of
Manitoba. RJG thanks the NSERC Discovery Grant program. B.D. is a research associate researcher of
the F.R.S.-FNRS. This work is a contribution to the ArcticNet Networks of Centre of Excellence
and the Arctic Science Partnership (ASP) asp-net.org and the ARC-cake club. The authors are grateful to
the anonymous reviewers and to the editor whose comments greatly improved the quality of the
manuscript.

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

Table 1: Seawater conditions on 11 January, before any sea ice formation (t=0), on 25 January, just
before the heat was turned back on and on 29 January, at the end of the experiment. Note
that seawater salinity and $TA_{(sw)}$ do not reach the initial seawater values as sea ice was still
present at the end of the experiment.

| Date | Temperature (°C) | Salinity | TA ($\mu$mol kg$^{-1}$) | $n$TA ($\mu$mol kg$^{-1}$) | $T$CO$_2$ ($\mu$mol kg$^{-1}$) | $nT$CO$_2$ ($\mu$mol kg$^{-1}$) |
|---|---|---|---|---|---|---|
| 11 Jan. | -1.4 | 33.5 | 2453 | 2416 | 2341 | 2306 |
| 25 Jan. | -1.9 | 35.5 | 2659 | 2471 | 2524 | 2346 |
| 29 Jan. | -0.6 | 34.4 | 2607 | 2500 | 2461 | 2361 |



Table 2: Masses of $TCO_2$ in the water column ($TCO_{2(sw)}$) and in the ice cover ($TCO_{2(ice)}$), masses of ikaite within the ice cover estimated from this study and from Rysgaard et al., (2014), masses of ikaite dissolved in the water column (Ikaite$_{(sw)}$) and masses of $CO_2$ exchanged between the ice and the atmosphere over the whole pool (estimation based on the air-ice $CO_2$ fluxes). All units are in mole.

| January (DOY) | $TCO_{2(sw)}$ | $TCO_{2(ice)}$ | Ikaite$_{(ice)}$ from this study | Ikaite$_{(ice)}$ from Rysgaard et al., (2014) | Ikaite$_{(sw)}$ | $CO_{2(air\text{-}ice)}$ |
|---|---|---|---|---|---|---|
| t=0 | 1041 | | | | | |
| 13.75 | 1040 | 2.38 | 0.17 | 0.00 | 0.47 | |
| 13.88 | 1044 | 2.09 | 0.00 | 0.00 | 0.87 | |
| 14 | 1043 | 2.90 | 0.25 | 0.00 | 0.83 | 0.03 |
| 14.13 | 1043 | 3.29 | 0.62 | 0.00 | 2.57 | 0.02 |
| 14.25 | 1038 | 4.91 | -0.05 | 0.00 | 1.06 | 0.01 |
| 14.5 | 1037 | 4.77 | 0.18 | 0.00 | 3.75 | 0.12 |
| 14.75 | 1039 | 4.36 | 0.12 | 0.05 | 2.73 | 0.07 |
| 15 | 1037 | | | | 1.80 | 0.08 |
| 15.25 | 1032 | 4.67 | 0.98 | 0.68 | 1.28 | 0.01 |
| 15.5 | 1034 | 3.89 | 1.58 | 0.00 | -1.57 | 0.07 |
| 15.92 | 1034 | 4.47 | 0.69 | 0.00 | 1.63 | 0.12 |
| 16.38 | 1024 | 7.36 | 1.45 | 0.08 | 3.60 | 0.19 |
| 16.67 | 1028 | 8.17 | 1.87 | 0.00 | 6.00 | 0.10 |
| 17.38 | 1023 | 15.48 | 0.29 | 0.65 | 3.90 | 0.22 |
| 17.67 | 1026 | 13.26 | 0.04 | 0.46 | 4.50 | 0.13 |
| 18.38 | 1030 | 11.39 | 0.74 | 2.14 | 5.61 | 0.38 |
| 18.67 | 1027 | 12.06 | 0.21 | 0.21 | 7.16 | 0.10 |
| 19.38 | 1029 | 11.13 | 0.01 | 0.84 | 6.96 | 0.23 |
| 19.67 | 1030 | 10.75 | 0.03 | 0.09 | 1.97 | 0.11 |
| 20.38 | 1028 | 10.25 | -0.12 | 0.23 | 1.47 | 0.42 |
| 20.67 | 1022 | 10.36 | -0.70 | 0.71 | 3.48 | 0.12 |
| 21.38 | 1025 | 10.50 | 0.88 | 0.35 | 7.42 | 0.35 |
| 23.63 | 1034 | 12.60 | 1.34 | 2.14 | 11.18 | |
| 24.38 | 1026 | 14.84 | 1.30 | 1.94 | 9.75 | 0.21 |
| 25.38 | 1017 | 15.67 | 1.09 | 3.05 | 6.62 | |
| 25.5 | 1029 | | | | 11.51 | 0.02 |
| 28.67 | 1022 | 13.46 | -0.57 | 0.46 | 20.91 | -0.93 |
| 29.38 | 987.3 | 15.82 | -0.56 | 0.55 | 26.72 | -0.15 |

## 10.  Figure Captions

Figure 1: The Sea Ice Environmental Research Facility with thin sea ice covering the pond during the

2013 experiment. Photo: J. Sievers.

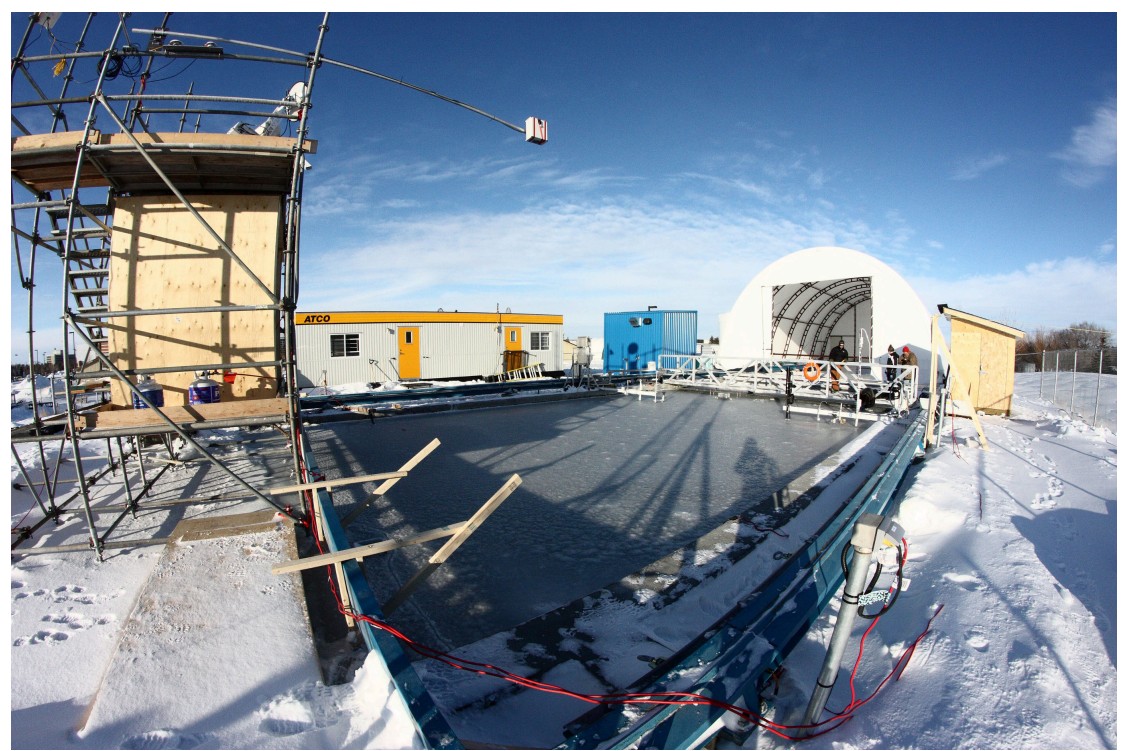

Figure 2: Evolution of (**a**) Air temperature (°C) at 2 m height, (**b**) snow thickness (black shaded areas) and sea ice/seawater temperature (°C), (**c**) bulk ice salinity, (**d**) brine volume content within sea ice, (**e**) seawater temperature (blue) and salinity (green). Measurements were performed at 30, 100, 175 and 245 cm water depths. The darker the color is, the closer to the surface. In panels (**b**), (**c**), (**d**) sea ice thickness is illustrated by black dots. Stars on panel (**b**) represent the depth at which the temperature profiles are derived from. Open squares in the lower part of (**d**) mark the sampling times. The dashed line on panel (e) indicates when the heat at the bottom of the pool was turned back on.

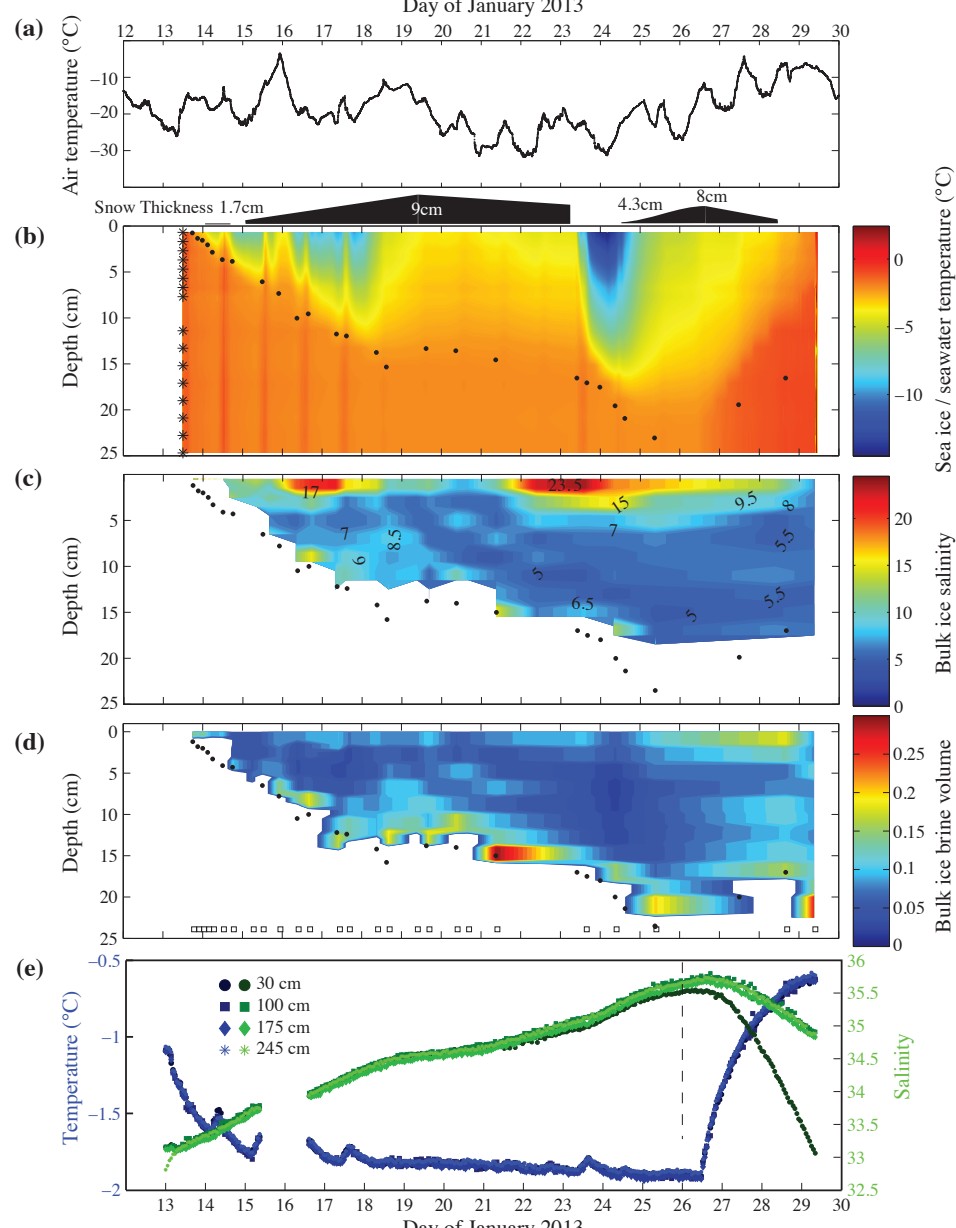

Figure 3: Evolution of (**a**) $TA_{(sw)}$ and $TA_{(sw)}^{*}$ (µmol kg$^{-1}$), (**b**) $TCO_{2(sw)}$ and $TCO_{2(sw)}^{*}$ (µmol kg$^{-1}$), (**c**) $nTA_{(sw)}$ (black) and $nTCO_{2(sw)}$ (green) (µmol kg$^{-1}$) and (**d**) the seawater $pCO_2$ (µatm) measured in situ (black) and corrected to a constant temperature of -1°C (blue). In panels (**a**) and (**b**) the black line is the average over the three depths while the dotted red line is the expected concentrations according to the variation of salinity observed and calculated from the mean values of the three depths ($TA_{(sw)}^{*}$ and $TCO_{2(sw)}^{*}$, respectively). The vertical black dotted line on 26 January mark when the heat was turned back ON.

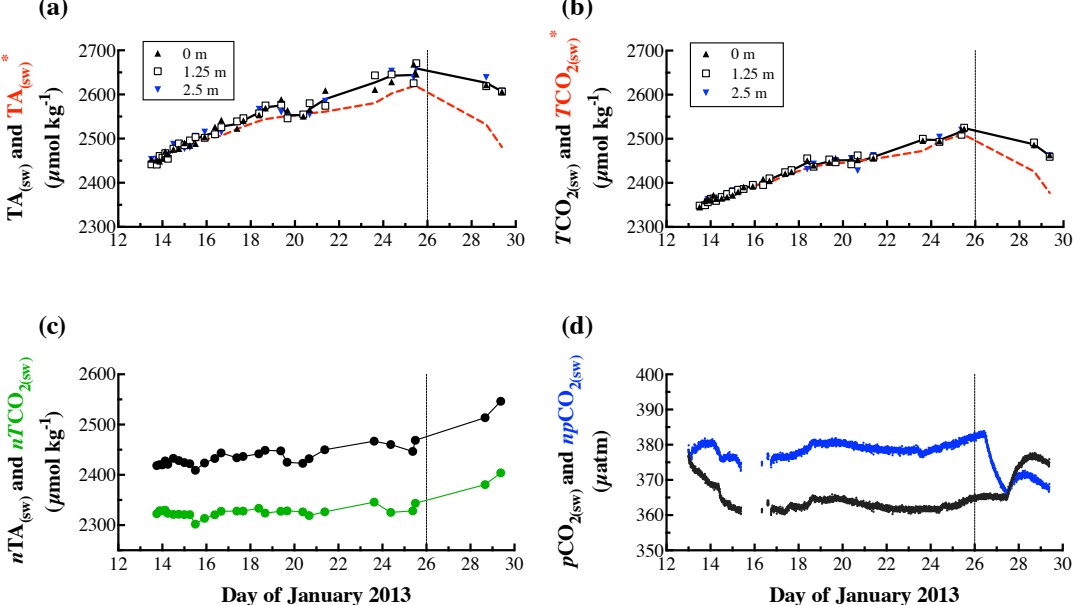

Figure 4: Evolution of (**a**) TA$_{(ice)}$ (µmol kg$^{-1}$), (**b**) $T$CO$_{2(ice)}$ (µmol kg$^{-1}$), (**c**) $n$TA$_{(ice)}$ (µmol kg$^{-1}$) and (**d**) $nT$CO$_{2(sw)}$ (µmol kg$^{-1}$). Sea ice thickness is illustrated by black dots. Open squares in the lower part of (**d**) mark the sampling times.

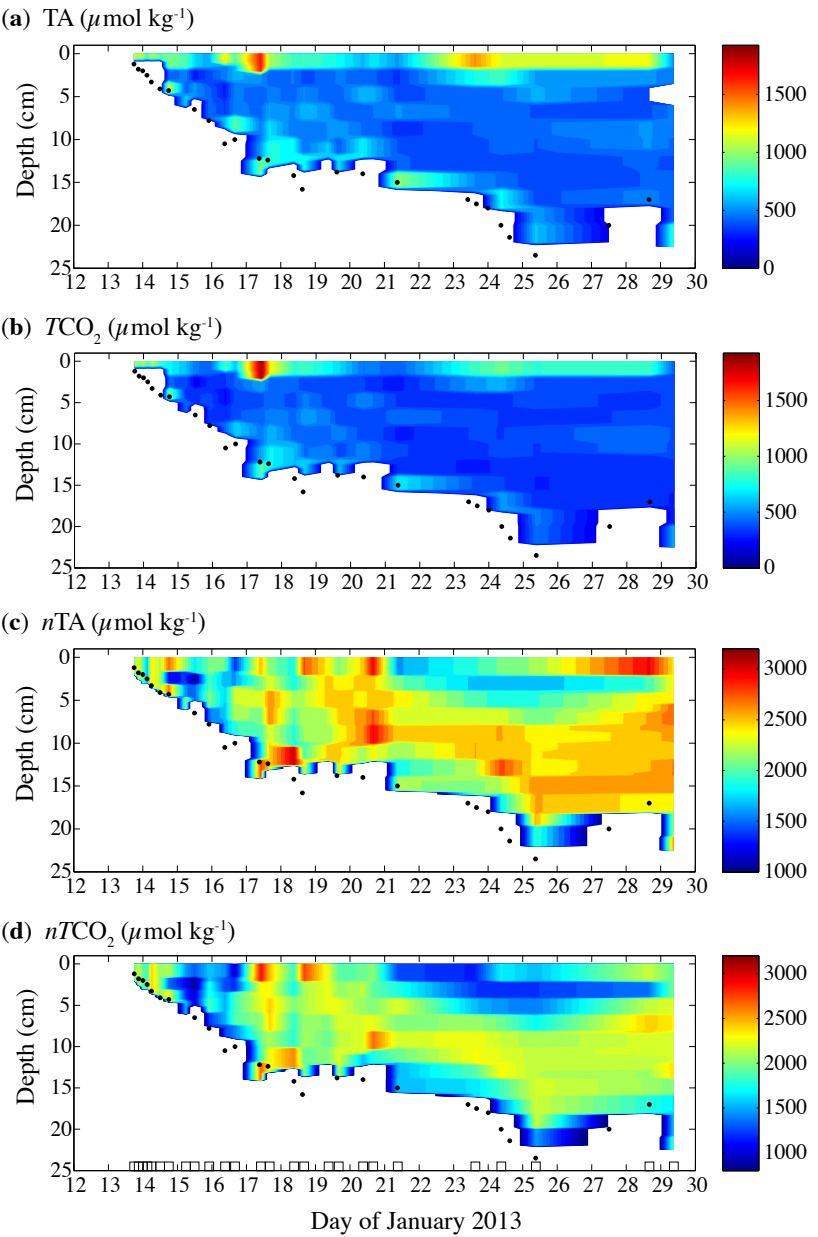

Figure 5: Air-ice $CO_2$ fluxes (mmol $m^{-2}$ $d^{-1}$). Positive air-ice $CO_2$ flux means outgassing from the ice and negative $CO_2$ flux means uptake of atmospheric $CO_2$. The vertical black dotted line on 26 January mark when the heat was turned back ON.

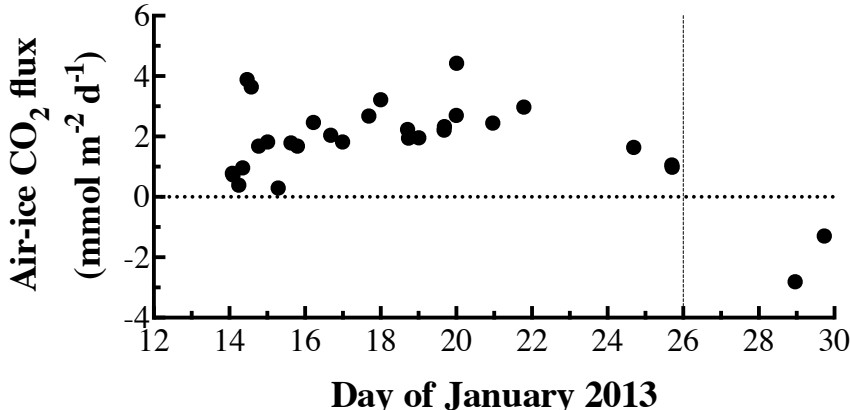

Figure 6: (**a**) Relationship between $nTCO_2$ and $nTA$ (µmol kg$^{-1}$) in bulk sea ice (white hexagons) and seawater (black dots), (**b**) Zoom on seawater data. The different dotted lines represent the theoretical evolution of $nTA$ and $nTCO_2$ ratio following the precipitation/dissolution of calcium carbonate and release/uptake of $CO_{2(g)}$. A linear regression is shown in green for the ice samples (a) and blue for seawater samples (b).

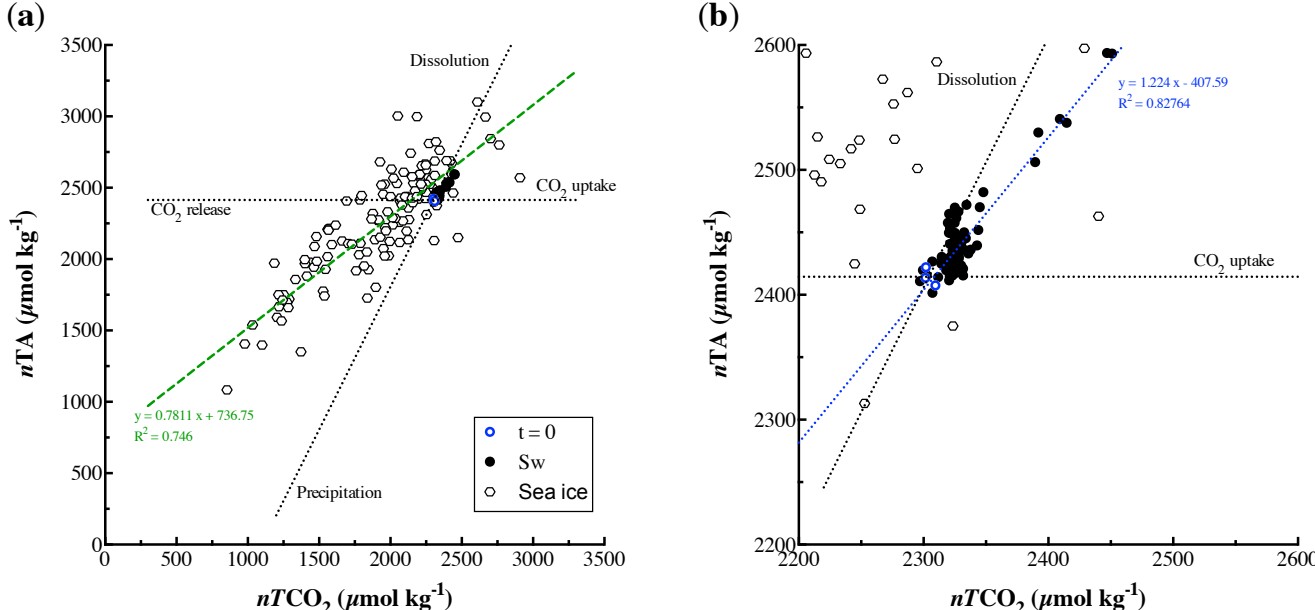

Figure 7: Evolution of (**a**) TA$_{(ice)}$ averaged throughout the ice thickness at each sampling day (black dots) and TA$_{(ice)}$* (dashed red line) (µmol kg$^{-1}$) and (**b**) $T$CO$_{2(ice)}$ averaged throughout the ice thickness at each sampling day (black diamonds) and $T$CO$_{2(ice)}$* (dashed red line) (µmol kg$^{-1}$), (**c**) Estimation of the ikaite precipitation/dissolution from half of the difference between TA$_{(ice)}$* and TA$_{(ice)}$ (µmol kg$^{-1}$) (black diamonds) compared to the average amount of ikaite precipitated throughout the ice thickness for each sampling day from Rysgaard et al., (2014) (white dots). The vertical black dotted line on 26 January mark when the heat was turned back ON.

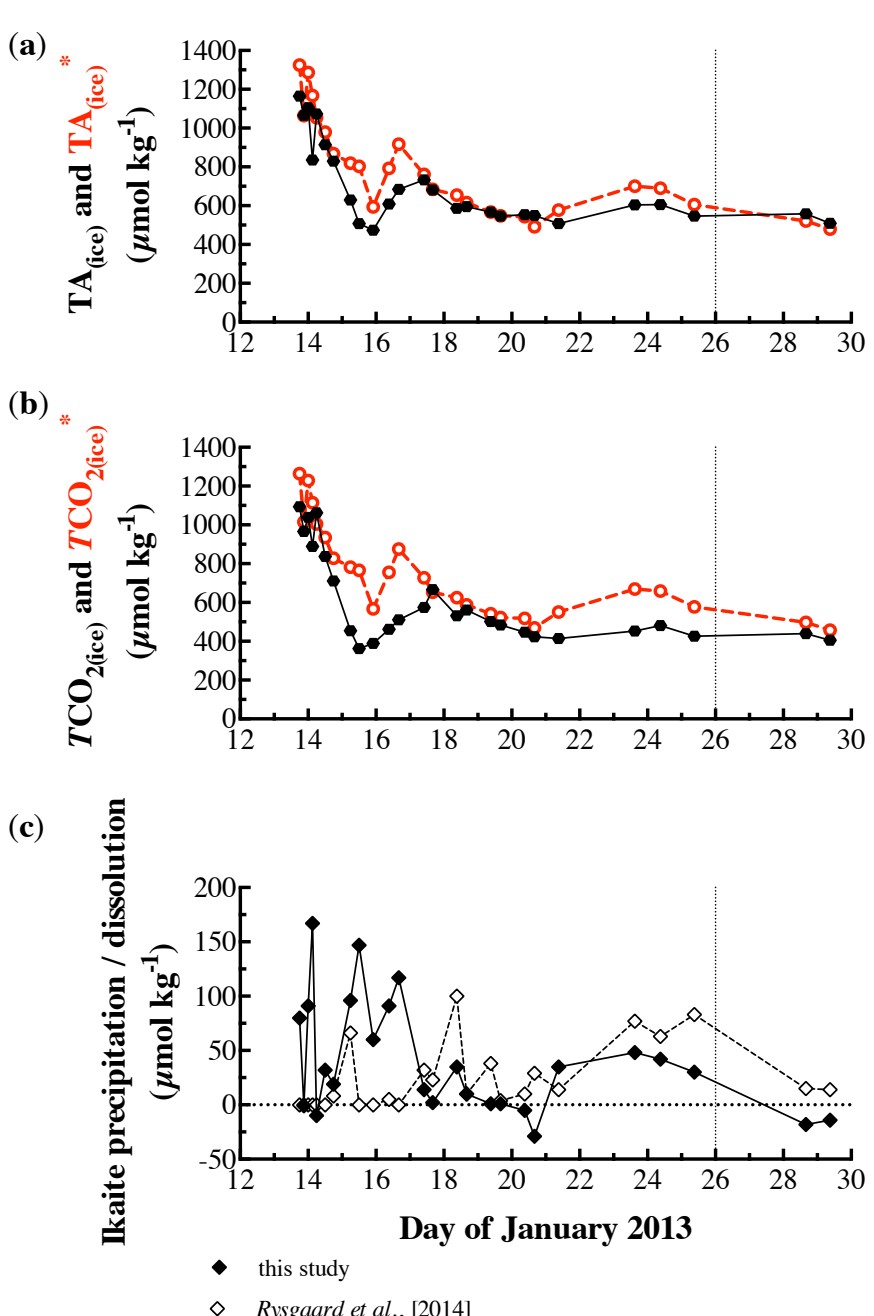

Figure 8: Evolution of (**a**) ikaite dissolution within the water column (in µmol kg⁻¹), (**b**) mass of ikaite dissolved in the underlying seawater (blue), mass of ikaite precipitated in sea ice (black) estimated from this study and estimated from Rysgaard et al., (2014) (white). The vertical black dotted line on 26 January mark when the heat was turned back on.

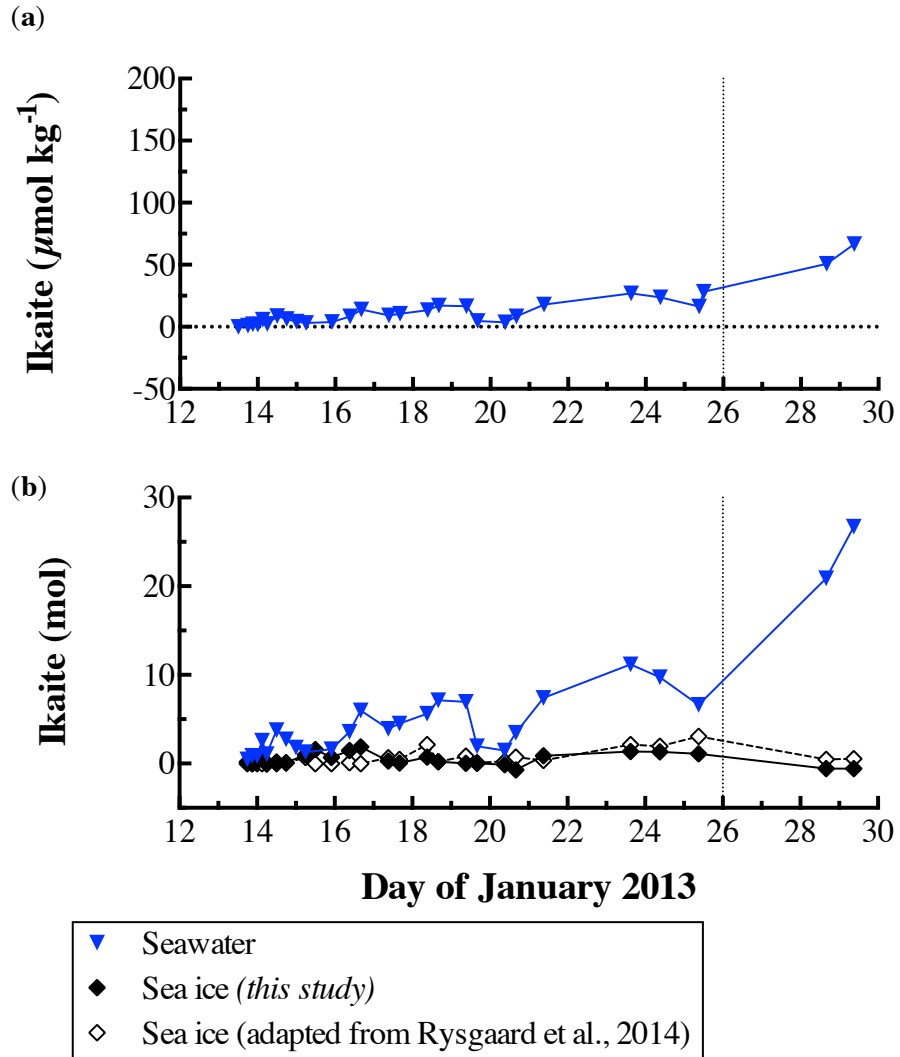

Figure 9: Total amount of $TCO_2$ lost from the ice cover (black dots), amount of $CO_2$ exchanges between the atmosphere and the ice cover ($CO_{2air-ice}$, white triangle) and sea ice-seawater $TCO_2$ exchanges (blue triangle). In mole for each day, integrated over the whole tank. The dotted line on 26 January mark when the heat was turned back ON.

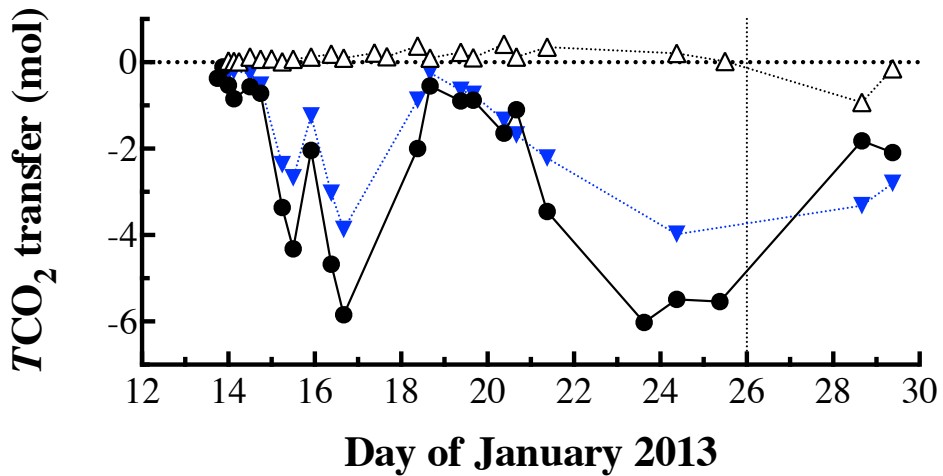

Figure 10: Evolution of (**a**) $\Omega_{\text{aragonite}}$ in the water column, calculated based on $TA_{(sw)}$ and $TCO_{2(sw)}$ (black dots) and calculated based on $TA_{(sw)}^{*}$ and $TCO_{2(sw)}^{*}$ (dashed red line) and (**b**) pH in the water column calculated based on $TA_{(sw)}$ and $TCO_{2(sw)}$ (black dots) and calculated based on $TA_{(sw)}^{*}$ and $TCO_{2(sw)}^{*}$ (dashed red line).

(a)

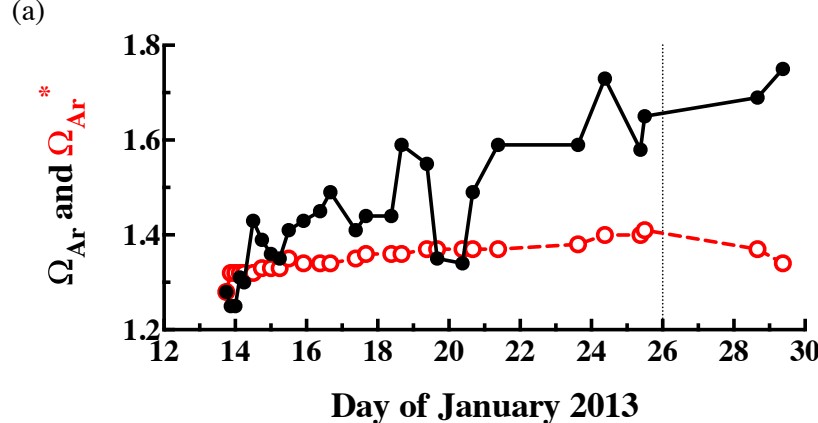

(b)

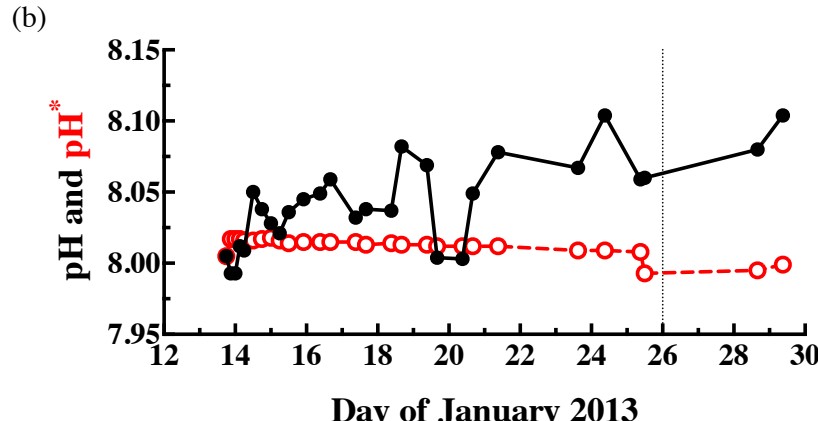