# Peer review of "Estimates of ikaite export from sea ice to the underlying seawater in a sea"

_The Cryosphere, 2016_

## Referee Comment (RC1) · Anonymous Referee #1 · 25 Mar 2016

I. General comments

The objective of the paper is to assess the impact of ikaite export from sea ice to the seawater, as clearly indicated by the title. Given that several studies propose that the inorganic carbonate chemistry in sea ice is of major influence for the polar air-sea CO2 exchange, quantifying the fate of ikaite in sea ice during sea ice growth and melt is of great interest.

To address their objective, the authors analyze temperature, salinity, total alkalinity (TA) and dissolved inorganic carbon (TCO2) in both the sea ice and the seawater in an artificial sea-ice pond located at the University of Manitoba, Canada. The method applied to analyze TA and TCO2 in the sea-ice is thought to measure dissolved TA and TCO2 only, specifically excluding TA and TCO2 bond in ikaite crystals (1 mol of ikaite

contains 2 mol of TA and 1 mol of TCO2). Ikaite concentrations in bulk sea ice and/or its export to the seawater was then derived by computing half of the difference between the theoretical TA concentrations in bulk sea ice if TA were to be conserved with salinity (this assumes no ikaite formation in sea ice) and the observed bulk sea-ice TA (without the ikaite-bond TA concentrations). Similar in seawater, ikaite concentrations are half the difference between the observed TA (which potentially would include imported and dissolved ikaite-bond TA) and the theoretical TA concentrations if TA were conservative with salinity.

Throughout large parts of the manuscript the authors claim that their methods would show ikaite concentrations in sea-ice, which, however is not entirely true since it could also be that ikaite was already exported from sea ice at the time when the observations were taken. In order to quantify the ikaite concentration in sea ice, the authors should have measured bulk sea ice TA concentrations of samples in which ikaite crystals were fully dissolved, and subsequently subtract from these 'new' measurements the already performed measurements of bulk sea-ice TA concentrations without dissolving ikaite. I am not sure to what extent the authors could preform additional measurements to account for this lack in observations.

Another way to approximate the exported amount of ikaite from the storage of ikaite in sea ice might be realized by subtracting the concentrations given in Rysgaard et al. 2014 for the same experiment from the observations presented in this study.

Despite of not being able to show ikaite concentrations in sea ice, the data presented on ikaite dissolution within seawater clearly supports the conclusion that ikaite is exported from the sea ice to the water column at various rates throughout the course of the experiment. However, the specific conclusion that up to 43 % of ikaite remain in sea ice while the rest is exported to the underlying water column is not supported by the presented data, and might be recomputed e.g. by using the ikaite content as given in Rysgaard et al. 2014.

Furthermore, the results showing that pCO2 below sea-ice remains undersaturated, which the authors relate to ikaite dissolution in sea-ice, is certainly a new results and important for assessing the influence of the inorganic chemistry in sea ice on air-sea-ice-sea CO2 exchange processes.

Beyond of analyzing the fate of ikaite in sea ice and seawater, the authors further consideration the influence of ikaite export to seawater on winter ocean aragonite saturation state. This approach is new, however, needs further clarification to understand the applicability of the experimental results on the influence of the process within the present-day and future Arctic Ocean, with the specific comments given in the corresponding section.

The overall structure and presentation of the paper is clear, and the language fluent. However, specific parts of the manuscript need some revisions concerning the clarity of the specific statements and the manuscripts needs a thorough read through concerning spelling and grammar mistakes.

Finally, the sectioning and structure of the manuscript is not following the guidelines for The Cryosphere, and thus needs some more attention.

After addressing the suggested revisions (see following comments) I think the paper is ready for publication.

II. Specific comments

1. Abstract

The Abstract provides a concise and brief summary. Specific questions, which have arisen in the results sections, potentially need to be re-considered in the Abstract.

L19: How do you derive the uncertainty range of $\pm 3$ umol/kg? No details about the uncertainty range are given in section 5.2.1.

L19: The specific percentage of the fraction of ikaite should be derived as detailed in

the comments to section 5.3.

2. Introduction

The introduction gives a solid general background on the processes related to the inorganic carbonate chemistry in sea ice and its influence on the air-sea CO2 exchange. Nevertheless, the authors need to state more detailedly why it is important to understand and quantify the fate of ikaite in the sea ice, specifically in the second last paragraph (L81-88).

L40-46: "Release of CO2 (from sea ice to the atmosphere) .. from open water." $\rightarrow$ Please be precise whether you mean air-sea-ice fluxes or air-sea fluxes.

L68: "..in the Arctic„": this should be also true for the Antarctic.

L79: "..will increase TA..": high TA in meltwater will certainly not increase the seawater TA, since the release of TA upon sea-ice melt dilutes surface ocean TA concentration since bulk sea ice TA concentrations (brine + ikaite) are always lower than TA concentrations of seawater, which is also supported by your data.

L87: "However, the..": This sentence would make more sense in the last paragraph, possibly after the second sentence.

L95-97: This is not a full summary of the paper, since you also look into $\Omega$aragonite.

3. Site description, sampling and analysis

L151-153: They way you write this sounds like you do the same as Rysgaard et al. 2014. However, you derive the 'ikaite' concentrations by the difference between (TAice*-TAice)/2, and only compare it to the concentrations estimated from microscope inspection as given in Rysgaard et al. 2014.

4.1. Sea ice and seawater physical conditions

L187: It seems strange that there is only a salinity stratification, however, in the temperature field the pool seems well mixed. How do you explain this?

4.2 Carbonate system

It is not really introduced why you look at nTA and nTCO2. To facilitate reading and understanding you should explain this a little more detailed.

L191-192: values given for seawater concentrations of TA and TCO2 at t=0 are different from the values given in Table 1.

L200: Additionally provide concentrations of nTA and nTCO2 at t=0.

5.1. Key processes affecting the carbonate system

L240: "up to 350 umol/kg", however, in Fig. 7c only up to to 100 umol/kg for the same dataset.

5.2. Estimation of the precipitation-dissolution of ikaite

L260: TCO2 is not conservative with salinity due to potential gas phase of CO2* (Eq. 2)

L271: Here the explanation of positive and negative signs is wrong: When the difference (TAice*-TAice )/2. is positive, then the observed TA concentration is lower than what would be expected from the theoretical conservation with S. Hence, TA is either in the form of ikaite crystals somewhere in the ice matrix or being exported potentially in ikaite form from the ice matrix to the seawater. In contrast, when the difference is negative, then it implies that more TA is observed in the brine than would be expected from the theoretical conservation with S, indicating dissolution of ikaite in brine. In contrast, in seawater, when the difference (TAsw*-TAsw)/(-2) is positive then ikaite is imported and dissolved in seawater releasing ikaite-bond TA, and if negative then TA is exported from seawater to somewhere else.

L272: "..implies a lack.." this is wrong, it implies that more TA is observed in sea ice. Please rewrite this sentence according to the changes above.

5.2.1 Sea ice

L279: "..is a result of ikaite precipitation..": it might also be the export of ikaite to the sea-water, see comments to L271 section 5.2, and L292, L302.

L296: "..found good agreement with small differences.." This statement is too positive for the time until the 17th of January 2013. As you state in L302 "ikaite crystals could have been formed and then exported into the underlying seawater.." In other words, for this period the comparison between the ikaite concentrations observed in Rysgaard et al. 2014 and your data suggests that your data shows concentration of TA*2 being exported to the seawater rather than the ikaite concentration stored in the sea ice matrix.

L309: "..negative difference .. indicating that ikaite dissolved in the ice or were exported to the water column.": A negative difference between TAice*and TAice does not necessarily mean that ikaite is exported, but it definitively implies that it is dissolved in brine. See comment to L271.

5.2.2. Water column

L324: Are the values of TCO2sw- TCO2sw* here similar to nTCO2 such as for TA?

L326-328: Please also explain the effects for the sea-ice covered period.

L329-351: Given that the dissolution of ikaite is a fast process in the order of seconds to minutes, a time delay in pCO2 rise of 1 day would suggest that during this day ikaite release to the sea-water must have been continuously large. Please discuss this more precisely.

5.3. Ikaite export from the ice cover to the water column

L360: "..of ikaite precipitated and remained within the ice cover.." See comment to L271, it could also be the amount of ikaite being exported.

L366-367: Since (TAice*-TAice )/2 could be ikaite crystals in sea ice and/or exported

ikaite crystals from sea ice it is not straight forward to derive the percentage of ikaite being exported or remained in the sea ice using this dataset. To derive ikaite concentrations in sea ice, you should measure sea ice concentrations with dissolved ikaite and subtract your observations from these new measurements. If this is unfeasible, you might instead consider to use the ikaite concentrations as given in Rysgaard et al. 2014 to compute an estimate of the fraction of ikaite remaining in sea ice.

5.4. Air-seawater exchange of inorganic carbon, attempt of CO2 budget

The calculation of the TCO2 export from ice to water is not clear, in particular, concerning the conversion of the air-seawater CO2 flux to units of mass? Opposed to the definition of ikaite concentrations in sea ice, you here nicely explain the potential processes that cause a negative difference between TCO2ice* and TCO2ice in lines 371-375: .."This could be due to different processes: (i) sea ice released CO2 to the atmosphere, (ii) the precipitation of ikaite within sea ice decreased TCO2(ice) and (iii) sea ice exchanges TCO2 with the underlying seawater." Unfortunately, for calculating the budget, you again assume that (TAice*-TAice )/2. would be ikaite concentrations (see comment to L271, explaining that for TA both (ii) and (iii) are relevant).

L388: "up to 99 %": please give mean range, or plot results as time series.

L389: "Between the beginning and the end of experiment, sea ice exported 2.8 mol." The methodology used to derive these values is unclear, it seems as if you give the value at the end of the experiment?

5.5. Impact of sea ice growth on aragonite saturation state of the Arctic Ocean in the context of ocean acidification

The title of this chapter implies that data from the Arctic Ocean are shown, however, the discussion is solely based on the SERF 2013 data. Hence, the chapter title is misleading.

Generally the data suggests that with or without ikaite export to the seawater the seawater would be supersaturated (Fig. 10), hence, in this experiment calcifying organisms should not face any problems. You should more clearly state that you base your conclusion "potentially hamper the effect of ocean acidification in fall to winter" on the differences between $\Omega$aragonite and $\Omega$aragonite*.

L441: Please clarify why $\Omega$aragonite reaches its minimum during winter, since this is not supported by the data on your sea-ice growth-melt cycle. Do other observations suggest this, if yes please refer to them?

L442: Why do you specifically only state the Arctic Ocean? In other words, do you expect different results for the Antarctic region, and if so, please explain the reasons.

L443-444: "..ice removal acts to impede the effect of ikaite rejection and therefore promote decreased $\Omega$aragonite." This calls for taking into account under-ice ikaite rejection in modeling predictions (..) in the context of sea ice rapid shrinking.": Several modeling studies have addressed the effects of future Arctic sea-ice decay on ocean acidification and report that Arctic surface ocean acidification is related to the rate of sea-ice reduction, and also to the responses of wind mixing and stratification under reduced sea-ice conditions (e.g. Steiner et al. 2014; Yamamoto et al. 2012). Please relate the increase in $\Omega$aragonite due to ikaite export from the ice to the surface ocean to the effects of rising atmospheric CO2 concentrations, global warming and associated Arctic sea-ice decay on $\Omega$aragonite projected by models.

6. Conclusion

The conclusions drawn are valid and well presented except of one part: The connection between L459-462 and L462-466 is not obvious. Please be more precise here. You should state that any attempt of deriving the air-sea CO2 flux related to the carbon pump should take into account that ikaite is exported to the underlying ocean during sea ice growth, which might reduce the efficiency of oceanic CO2 uptake upon sea ice melt related to the sea ice pump.

III. Technical comments in Figures and Tables

Caption Fig. 1.

Give year of experiment.

Fig. 2.

Put panel labels in brackets.

Panel (d): y-axis units for temperature overlain on bracket.

Panel (c) and (d): units for salinity not given.

Panel (d): legend is missing information of what is temperature and sanity.

Caption Fig. 2.

Use lower case letters for panel description.

".. snow cover": better to use same text as in figure: 'snow thickness'

"..black horizontal bars..": no bars visible: maybe better use 'black shaded areas'

Indicate colors for seawater temperature and sanity in panel (d)

Last sentence, final dot missing.

Caption Fig. 3.

Use lower case letters for panel description.

Indicate also that red line is calculated on the mean values of the three depth intervals.

Legend in panel (a) and (b): no need for 'Sw' abbreviation. One space between depth value and unite.

Caption Fig. 4.

Use lower case letters for panel description.

Caption Fig. 6.

Use lower case letters for panel description.

Caption Fig. 7.

Use lower case letters for panel description.

One space after the fist TA(ice)

Fig. 8.

y-axis labels: "Ikaite" This is to imprecise, see comments to L271.

Caption Fig. 8.

Use lower case letters for panel description.

"..amount of mol of ikaite.." mol is the unit: '..mass of ikaite in mol.'

Fig. 9.

A flux cannot have the unit mol.

"Total TCO2 lost by the sea ice" This is the mass of TCO2 of ice cover assuming the absence of ikaite.

"Air-ice CO2 fluxes" cannot have units of mol

Caption Fig. 9.

A flux cannot have the unit mol.

"Total TCO2 exchanges by the ice cover" This is the mass of TCO2 of ice cover assuming the absence of ikaite and CO2(g) in bubbles.

Fig. 3. to 5. and Fig. 7. to 9.

x-axis text: use for all plots the same x-axis text, e.g. the same as used in Fig. 2.:"Day

of January 2013".

Table 1.

Please also give the water column conditions just before the heat was turned ON, and at the end of the experiment.

Table 2.

"amount": better use 'mass'

"ikaite": See comment to Fig. 8. and L271.

"CO2 fluxes (mol)" fluxes are defined e.g. to be changes in mass over time. How do you derive these values, please clarify this in the text, see comment to section 5.4.

IV. Technical comments

L18-19: "..ikaite precipitated..", better: '..ikaite concentrations of up to..'

L19: "..within sea ice; up to..": wrong '..within sea-ice; at least..'

L29: "Each year..": Be more precise: e.g. 'Currently, each year..'

L30: "..through primary production and surface cooling..": please be slightly more precise here explaining the biological pump and the dissolution pump.

L33: Please give references, and clarify if you refer only to model and/or observational estimates.

L33: "..sea ice an impermeable.." grammar mistake: '..sea ice as an impermeable..'

L47: comma missing after "formation"

L50: please give reference.

L51: "..inorganic carbon.." $\rightarrow$ give abbreviation TCO2

L56: "..TA is reduced..": Be more precise: e.g. ..'TA in brine is reduced..'

L57: "..while TCO2 is..": Be more precise: e.g. '..while TCO2 in brine is..'

L59: "[CO2]": Spelling mistake: [CO2*]: and define in text what this means

L60: "[B(OH-)]": Spelling mistake: B(OH)4-]

L60: "[H-]": Spelling mistake: [H+]

L63: Also re-mention the pathway of brine to the sea-ice surface as was given in lines 47-48.

L67: "..fluxes..": Be more precise: e.g. '..air-sea fluxes..'

L67: "..seawater and incorporated..": Grammar mistake: e.g. '..seawater and its contribution to intermediate and deep-water formation..'

L77: "..melting sea ice promotes..": Be more precise: e.g. '..surface warming and melting sea ice promote..'

L80: "..underlying seawater..": be more precise: e.g. surface ocean $\rightarrow$ due to the enhanced summer stratification L78.

L90: "435 m$^3$": Give dimensions rather than the volume.

L167: "..15 cm on 18 January..": mistake: '..15 cm until 18 January..'

L170: "..This results in the increase of the sea ice temperature..": Be more precise: e.g. 'This resulted in repeated events of increased sea-ice temperatures..'

L197-198: it would be more intuitive to call it 'sw' and 'ice' instead of 'sample'. Delete brackets around 'sw', and 'ice', put brackets around 't' and do not write 't' in subscript.

L201: "..ice started the melt..": mistake: '..ice started to melt..'

L225: "..suggest..": this is rather weak, better to write that the measurements 'show' this.

L228: Is is not appropriate to give a range of two values.

L264-265: in the equation 'sample' means 'sw or ice'. It would facilitate reading if you would write for each component a separate equation. It is better to not put "sw" and "ice" within brackets, while the time variable should be written not in subscript and in brackets.

L270: "assume" this sounds like as if another value than the one-half would also make sense. You should clearly explain here were the one-half comes from: 1 mol of ikaite contains 2 mol of TA.

L278: "..assume.." see comment to L270

L299: "..precipitate..": grammar mistake: '..precipitated..'

L304: "..differences observed": be more precise: e.g. '..differences between both methods..'

L318: "..to the amount of ikaite..": be more precise: e.g. '..to the concentration of ikaite..'

L317: "..half the difference between TA(sw)* and TA(sw) .." mistake: '..half the difference between TAsw and TAsw*..'

L319: "..amount..": imprecise formulation: '..concentration..'

L322: ".. TA(sw)* and TA(sw) ..": see comment L317

L345: "..processes other than a the temperature.." → delete the 'a'

L346: Changes in pCO2 cannot not understood from equation 3.

L367: "..crystals remain contain within..": grammar mistake: '.. crystals remain within..'

L379: Fig. 9 not 8

L379: "maximum outgassing":logical mistake: '..maximum loss of TCO2..'

L386: "..substracting .. ikaite precipitation to the total": grammar mistake: '..precipitation from the total..'

L392: "Fig 3c" → Fig. 3B

L402: "..convection.." This should include all kind of transport mechanisms e.g. advection, mixing

L400: "..measurement..": spelling mistake: '..measurements..'

L424: "..area..": spelling mistake: '..areas..'

L426: "..as a result of respiration.": missing process: '..as a result of respiration and dissolution..'

L438: "..dramatically..": formulation too dramatic

L438: "..increase the..": grammar mistake 'increase..'

L439: "This suggest that..": spelling mistake: 'This suggests that..'

L444: "..and therefore promote decreased..": grammar mistake: '..and therefore promotes a decrease of..'

L457: "up to 66.." → '..up to 128..'

L462: Please give more references

L470: "..is responsible the..": '..is responsible for the'

L471: "..we project that..": '..we discuss that..'

References

Yamamoto, A., Kawamiya, M., Ishida, A., Yamanaka, Y., and Watanabe, S.: Impact of rapid sea-ice reduction in the Arctic Ocean on the rate of ocean acidification, Biogeosciences, 9, 2365-2375, 2012.

Rysgaard, S., Wang, F., Galley, R. J., Grimm, R., Notz, D., Lemes, M., Geilfus, N. X.,

Chaulk, A., Hare, A. A., Crabeck, O., Else, B. G. T., Campbell, K., Sørensen, L. L., Sievers, J., and Papakyriakou, T.: Temporal dynamics of ikaite in experimental sea ice, The Cryosphere, 8, 1469-1478, 2014.

Steiner, N.S., Christian, J.R., Six, K.D., Yamamoto, A., Yamamoto-Kawai, M. 2014. Future ocean acidification in the Canada Basin and surrounding Arctic Ocean from CMIP5 earth system models, J. Geophys. Res.-Oceans, 119, 332-347, 2014.

---

## Referee Comment (RC2) · Anonymous Referee #2 · 2 Apr 2016

**Review on Geilfus et al. (2016)**

Geilfus et al. (2016) discuss data from a most interesting sea ice formation (and a bit of melting) experiment performed at the Sea-ice Environmental Research Facility (SERF) site from 13 to 30 January 2013 at the University of Manitoba, Winnipeg, Canada. Several articles have been published already using data from this experiment (Hare et al., 2013; Rysgaard et al., 2014, Else et al., 2015). Geilfus and colleagues use measurements of total alkalinity (TA), dissolved inorganic carbon ($TCO_2$, total $CO_2$), salinity, temperature, and a few other measurements to estimate the carbon budgets in sea ice and the underlying (artificial) sea water, especially the precipitation, transfer, and dissolution of ikaite. The conservative components of the marine carbonate system, namely TA and $TCO_2$, vary due to three processes: (1) Change in salinity due to formation and melting of sea ice, (2) precipitation or dissolution of calcium carbonate, here in the form of ikaite, and (3) gas-exchange. The size of the processes can be estimated in the following sequence: (1) can be quantified by scaling TA and $TCO_2$ using salinity (Eqs. 6 & 7). (2) can be estimated from changes of TA whereby the amount of calcium carbonate precipitation (and associated $TCO_2$ decrease) is equal to half of the TA reduction; the dissolution of calcium carbonate precipitation has the opposite effect. (3) The residual $TCO_2$ variation should be due to gas-exchange which might be, however, difficult to estimate because of uncertainties when calculating small differences.

The data (TA, $TCO_2$, T, S) seem to be of high quality, however, a detailed discussion of the time evolution of measured and derived quantities is largely missing; often only wide ranges ('0.47 to 26.71 mol') are given. A proper analysis of the data, estimates of uncertainties, identification of surprising or contradicting findings and a proper overall budget (How to close the TA budget?) for the whole pool is largely missing. Thus I cannot recommend publication.

General comments & suggestions:
Units: the partial pressure of $CO_2$, $pCO_2$, should be given in $\mu$atm (and not ppm; ppm refers to the mixing ratio of $CO_2$, $xCO_2$)
Which program/package do you apply for carbonate system calculations? Which equilibrium values do you use? For a recent discussion compare Orr, Epitalon & Gattuso (2015).

Specific comments & suggestions:

1. L 30: $CO_2$ emissions & oceanic uptake: Sabine et al., 2004 is an excellent paper, however, I suggest to cite more recent estimates (for example, IPCC 2013, or Global Carbon Project)

2. L 31: 5-14% of the global ocean $CO_2$ uptake: based on which values?

3. L 47-48: 'During the earliest stages of sea ice formation a small fraction of $CO_2$-supersaturated brine is expelled upward onto the ice surface promoting a release of $CO_2$ to the atmosphere (Geilfus et al., 2013a).' It might be interesting to elaborate a bit more on 'expelling brine': When does it occur? How much brine can be expelled? Level of $CO_2$-supersaturation? Salinity of the expelled brine?

4. L 50: 'physical concentration'??? I suggest dropping 'physical'

5. L 60: Eq. (3) is an approximation to the TA definition given by Dickson (1981). In your experiment you use a special form of artificial seawater (ASW). It would be interesting how much total borate is in the ASW and how this is taken into account in the calculation of $pCO_2$ from TA and $TCO_2$.

6. L 78-80 'The mixing of meltwater, that is low in $TCO_2$, $pCO_2$, and high in TA due to brine dilution and ikaite dissolution, with seawater will increase TA and decrease the $pCO_2$ of the underlying seawater, enhancing the air-sea $CO_2$ fluxes (Rysgaard et al., 2007; 2009).'
$pCO_2$ of seawater is not a 'substance' that can be 'mixed': it is the equilibrium partial pressure of seawater and does not follow a linear mixing relationship. $TCO_2$ in meltwater is low compared to (artificial) seawater. Meltwater $pCO_2$ is low compared to atmospheric $CO_2$ because of low $TCO_2$ and not enough time for gas-exchange and equilibration with the atmosphere. I don't know why meltwater TA should be higher than in ASW, because the ikaite was precipitated from ASW and then dissolves again.

7. L 92-95 'We gain the ability to carefully track carbon parameters in the ice, in the atmosphere, and in the underlying seawater, while growing sea ice in a large enough volume of seawater, so that conditions closely mimic the natural system.'

However, there are various differences to the natural system; to name only a few: no leads for heat & gas-exchange, no horizontal ice movement impacting mixing of the underlying water, no 'biology' (which here simplifies the analysis of the carbonate system), the pressure build-up during the first part of the experiment. These differences should be mentioned and possible consequences for data interpretation should be discussed, especially with respect to comparison with the real world.

8. L 104 '(ASW) formulated by dissolving large quantities': formulated $\Rightarrow$ generated, fabricated

9. L 189-191 'TA and TCO$_2$ in seawater, noted as TA(sw) and TCO$_2$(sw), were sampled at the sea ice-seawater interface, 1.25 and 2.5 m depth. However, as the variations of TA and TCO$_2$ over the 3 depths are quite small (SD = 8.75 and 4.5 $\mu$mol kg$^{-1}$, respectively), we consider the average concentration.'
Do you really mean 'variations' of TA (with a standard deviation of 8.75 $\mu$mol kg$^{-1}$) or differences of TA between the 3 levels. If the latter: give mean difference $\pm$ SD.

10. L 204-205
'The pCO$_2$(sw) then oscillated from 360 to 365 ppm during sea ice growth.' $\Rightarrow$
'The pCO$_2$(sw) then varies from 360 to 365 $\mu$atm during sea ice growth.'

11. L 219 'minimums' $\Rightarrow$ minima

12. L224-228: Air-ice CO$_2$ fluxes:
Although it's good to know the ranges of CO$_2$-fluxes, in the current context it would be even more interesting the fluxes integrated over time.

13. L 238-240 'For this 2013 experiment, Rysgaard et al. (2014) discussed the precipitation of ikaite within the ice cover in detail, reporting high concentrations of ikaite ($>$ 2000 $\mu$mol kg$^{-1}$) at the surface of the ice and ikaite precipitation up to 350 $\mu$mol kg$^{-1}$ in bulk sea ice.'
The concentrations, especially at the surface, are impressive. In the

current context (TA and $TCO_2$ budgets for the whole pool) it would be good to obtain integrated values, at least rough estimates.

14. L 244 please drop 'Therefore'

15. L 255 please drop 'However,'

16. L 256-257 Try to avoid repetition ('2:1 ratio'): 'As illustrated in Figure 6, an exchange of $CO_2$ does not affect TA while the precipitation-dissolution of ikaite affect TA and $CO_2$ in a ratio 2:1.'

17. L 271-274 'A negative difference (i.e. $TA(sample)^* <$ TA(sample)), implies that a lack of TA is observed in the sample compared to what is expected based on the observed salinity changes (Fig. 2). This suggests that ikaite crystals were either dissolved or exported out of the sample (sea ice or seawater).'
    difference = $TA(sample)^*$ - TA(sample)
    I don't understand the sentence: 'negative difference' means TA(sample) > $TA(sample)^*$, i.e. there is more TA in the sample than expected from salinity scaling; dissolution of ikaite (that was imported from somewhere else) would indeed increase TA; export of ikaite (that has been precipitated in the sample) would imply a decrease of sample TA.

18. L 278 '... both processes reduce and $TCO_{2(ice)}$': outgassing of $CO_2$ (one of the two processes) does not change $TA_{(ice)}$, please rewrite sentence accordingly.

19. Figure 7:
    (1) $TA^*_{(ice)}$ looks like you have continuous (or at least many) measurements. Please give some info.
    (2) I'm wondering how much of the difference between $TCO_{2}^*{}_{(ice)}$ - $TCO_{2(ice)}$ can be explained by ikaite precipitation alone and suggest to show this in another panel added to the Fig. 7.

20. Table 1: to display 4 values only, a table is not required, however, it would be good to extend the table and give values of $TA_{(sw)}$, $TCO_{2(sw)}$, $TA_{(ice)}$, $TCO_{2(ice)}$, $S_{(sw)}$, $T_{(sw)}$, $S_{(ice)}$, $T_{(ice)}$ for the time points at which you took $TA_{(ice)}$ samples.

21. L 286-288 'The upward percolation of seawater observed from 15 to 18 January might complicate the picture of the effect of sea ice temperature on ikaite formation.'
I bit more detailed description what happened here would be useful (or can it be found somewhere else, reference?). 15 to 18 January is the period with large differences $TA^*_{(ice)}$ - $TA_{(ice)}$, $TCO2^*_{(ice)}$ - $TCO2_{(ice)}$, and large discrepancy between estimates of ikaite precipitation by Rysgaad et al. (2014) and the current investigation (Fig. 7).

22. L 293-297 'So, we compared the direct microscopy observations by averaging the amount of ikaite precipitated throughout the ice thickness for each sampling day from Rysgaard et al., (2014) (Fig. 7c, white dots) with our estimation of the amount of ikaite based on the difference between $TA^*_{(ice)}$ and $TA_{(ice)}$ (Fig. 7c, black dots) and found good agreement, with some small differences likely due to methodological differences.'
please give a correlation coefficient.

23. L 298-301 'During melting of the sea ice samples, ikaite crystals may have dissolved, leading to an underestimation of the total amount of ikaite precipitate [precipitation] in the ice. This bias is avoided during direct microscopic observation of the crystals (Rysgaard et al., 2014) if crystals are large enough to allow optical detection.'
Do you see a significant difference in the mean values of ikaite precipitation estimated by the two methods?

24. L 315-317 'According to equations 1 to 3, lower $TA^*_{(sw)}$ and $TCO2^*_{(sw)}$ compared to $TA^*_{(sw)}$ and $TCO2^*_{(sw)}$ (Fig. 3b, c) confirm the dissolution of ikaite in the underlying seawater.'
Eqs. (1)–(3) do not contain the quantities $TA^*_{(sw)}$ and $TCO2^*_{(sw)}$: please rewrite accordingly

25. Fig. 8A does not make sense to me because you compare ikaite precipitation and dissolution using concentrations in one reservoir (sea ice) which shows large relative changes in volume and in another huge reservoir (seawater). I suggest to drop Fig. 8A.

26. According to Fig. 8B much more ikaite has been dissolved in seawater than precipitated in sea ice: What's your explanation?

27. L 338-340 'Using the equation from Copin-Montegut (1988), we normalized the $pCO_2$(sw) to a temperature of -1°C (noted as $npCO_2$(sw), blue line on Fig. 3d).'
No motivation is given for this 'normalization' and I don't see why to do so. Once again: $pCO_2$(sw) is not a substance. The gas-exchange depends on the actual $pCO_2$(sw) (strongly dependent on temperature!).

28. L 361 'Within the water column, 0.47 to 26.71 mol of ikaite dissolved.'
Please give a proper discussion of the evolution in time (Fig. 8B) and how this evolution is related to various processes. What might have caused the drop of ikaite dissolution in seawater around 20 January? How to close the TA budget? Compare also Fig. 3

29. L 375-377 'To estimate the amount of $TCO_2$ exchanged during this experiment, we convert our units to moles, using the sea ice (and seawater) thickness (in meter) and density (in kg/m3) and the pool dimension (in meter).'
This is not just a conversion of units! Instead of concentrations you consider reservoir contents!

30. L 418-419 'Using the seawater conditions at the end of the experiment, a layer of 1cm of seawater in the pool contains 4.21 mol of $TCO_2$, making it difficult to close our budget.'
It's good that you mention this uncertainty. I would like to see more uncertainty estimates in the manuscript.

**References**

[1] Dickson, A.G. An exact definition of total alkalinity and a procedure for the estimation of alkalinity and total inorganic carbon from titration data. *Deep Sea Research Part A. Oceanographic Research Papers*, 28(6):609–623, 1981.

[2] Else, BGT and Rysgaard, Søren and Attard, Karl and Campbell, K and Crabeck, O and Galley, RJ and Geilfus, N-X and Lemes, M and Lueck, R and Papakyriakou, T and F. Wang. Under-ice eddy covariance flux measurements of heat, salt, momentum, and dissolved oxygen in an

artificial sea ice pool. *Cold Regions Science and Technology*, 119:158–169, 2015.

[3] Geilfus, N.-X. and Galley, R. J. and Else, B. G. T. and Papakyriakou, T. and Crabeck, O. and Lemes, M. and Delille, B. and Rysgaard, S. Impacts of ikaite export from sea ice to the underlying seawater in a sea ice-seawater mesocosm. *The Cryosphere Discussions*, 2016:1–33, 2016.

[4] Hare, AA and Wang, Fei and Barber, Dave and Geilfus, N-X and Galley, RJ and Rysgaard, Søren. pH evolution in sea ice grown at an outdoor experimental facility. *Marine Chemistry*, 154:46–54, 2013.

[5] Orr, J.C., J.-M. Epitalon, and J.-P. Gattuso. Comparison of ten packages that compute ocean carbonate chemistry. *Biogeosciences*, 12(5):1483–1510, 2015.

[6] Rysgaard, Søren and Wang, F and Galley, RJ and Grimm, Rosina and Notz, Dirk and Lemes, M and Geilfus, N-X and Chaulk, A and Hare, AA and Crabeck, O and others. Temporal dynamics of ikaite in experimental sea ice. *Cryosphere*, 8:1469–1478, 2014.

---

## Referee Comment (RC3) · Anonymous Referee #3 · 11 Apr 2016

Review of the manuscript by Geilfus et al. Impact of ikaite export from sea ice to underlying seawater in a sea ice-seawater mesocosm

The manuscript describes a mesocosm experiment with artificial sea ice and seawater and the precipitation of ikaite and the impact of the exported ikaite on the underlying water using the SERF artificial outside seawater tank, the University of Manitoba in Winnipeg, Canada. The authors show data and results mainly as the changes and evolution in measured seawater TA and $TCO_2$ and salinity-normalized TA and $TCO_2$ during a 17 day-period. Measured air-ice $CO_2$ exchange during the study is also presented. The investigation of sea ice processes and underlying water in a confined setup in an outside environment with mainly the processes of salinity changes, ikaite precipitation/dissolution and $CO_2$ gas exchange affecting the carbonate chemistry (assuming insignificant effect of biological processes) is new and interesting. However, the idea of solid ikaite export to the water column and the effect of ikaite on the underlying water such as aragonite saturation state has been presented and discussed in a few publications, which should be referred to. These publications also describe sea ice processes and evolution of the sea ice and underlying water in natural sea ice. However, the estimates of the amount of ikaite exported out from the sea ice to the water beneath compared to the ikaite precipitation in sea ice are new and valuable. I think it is an interesting approach and important study in a controlled environment but it needs improvements. There are too many unclear calculations, figures, statements and missing uncertainty discussions. Hence, the manuscript requires substantial revision and cannot be published in its present form. However, I encourage publication after major revision.

*General comments:*
Parts of the results are not convincing with measured TA in the seawater being higher at the end of the experiment (melt) than at the start of the experiment. Important discussion and uncertainty investigations are missing regarding the contradictions of the results. Some figures are unclear, and calculations are not well described and are sometimes difficult to follow and reconstruct, such as the mole calculations of ikaite as well as the result of 57% of ikaite exported from the ice. Essential data are missing and a description of the evolution of the TA in the underlying water is missing. The uncertainty discussion on ikaite dissolution during analysis and not in the water column is missing and not mentioned in the method section. There are also unclear explanations of some of the contradicting results.

I also have concerns about the statement and conclusion about ikaite dissolution in seawater as ikaite probably does not dissolve at temperatures <0°C, such as the temperature in the underlying water. The seawater samples were stored at +4°C so the ikaite was probably dissolved or near dissolution before or at analysis, and not in the water column. The water column temperatures were between -3°C and -7°C during the study and about -1°C at the end of the study. There is lack of information on temperature, salinity, TA and $TCO_2$ at the end of the experiment when the ice was melted. This should be mentioned in the method and

discussion sections, to be able to close the TA seawater budget from start to end, which seems to be a problem. The seawater salinity and TA could change during the study since freshwater in the form of sea ice is removed every time an ice core is collected, and same for seawater. What about TA and $TCO_2$ in the snow, brine and brine-skim, where these analysed? These factors could be discussed if they impact the results and maybe also used correcting the calculations.

In parts of the result, the air-sea $CO_2$ flux is not considered and left out in the statement of processes when calculating the changes in $TCO_2$, which is an important process driving the changes in $TCO_2$ (except for biological production) although with relatively small effect. However, this is later discussed in the manuscript. The information on wind speed is missing, it is essential for gas exchange to occur between ice and air. Metrological data could perhaps be presented in a table and moved to site description/method since this is not a result of this paper and already presented by Rysgaard et al. (2014).

Important and highly relevant references are missing in the introduction and discussion sections, such as Fransson et al. (2013) and Chierici el al. (2011), which performed the first studies of the carbonate chemistry and aragonite saturation (ocean acidification) in natural sea ice and underlying water during a full ice season in the Arctic. I suggest that these references are cited and mentioned in the discussion section. There are other relevant references that I suggest to be included, see Specific comments.

The manuscript would benefit from language correction by English native person.

*Specific comments*
Line 1. The title may not inform the reader what this manuscript is about. I suggest changing "Impacts" (on what?) to "Estimates" or "indications".

**Abstract**
Line 12. This sentence suits better in introduction, it is not the result of the manuscript. I suggest removing the sentence and start the next sentence with "The fate ….".

Line 14 and throughout the manuscript. As far as I can see, the experiment was performed during 17 days, not month-long experiment or three weeks, as is also written at various places in the manuscript. Please change to "17 days long" or just mentioned the dates.

Lines 16, 20, 25 and throughout the manuscript: "dissolution of ikaite" has to be explained or used properly. Ikaite will probably not dissolve in the cold water (<0°C), so please add information to explain what you mean with "dissolved ikaite". You may write "presence of ikaite dissolved during analysis". Perhaps you have proofs on the dissolved ikaite in the underlying water (before storage or analysis), then please add that information.

**Introduction**

Lines 35-36. The references mentioned, do they report on sequestration of atmospheric $CO_2$ below the mixed layer or only into the surface mixed layer? Do they have evidence that the ice-brine pump actually exports atmospheric $CO_2$ below the mixed layer (i.e. sequestration for longer periods)? There are other studies (not so recent) of $CO_2$ sequestering which are more relevant, e.g. $CO_2$ uptake in the Arctic Ocean due to brine rejection (e.g. Anderson et al., 2004) from brine rejection, and (and very recent) Brown et al. (2016) that may also be referenced? There may also be modeling results. Maybe use other reference or change to "sequestration into the mixed layer below the ice".

Line 39. I suggest removing "$CO_2$", and start the sentence with "The carbonate chemistry…". What do you mean with "heterogeneous"? Do you mean that the distribution or concentrations are heterogeneous?

Lines 41-45. I suggest to add the reference of Fransson et al. (2013) for both $CO_2$ release in winter and $CO_2$ uptake during ice melt.
Line 46. What is the sea ice pump, please explain why and how $pCO_2$ is controlled?
Line 53. Please add the reference Nomura et al. (2013) (after Dieckmann et al., 2008), they also found ikaite crystals in natural Arctic sea ice.
Lines 54-55. Please add the reference Fransson et al. (2013).
Lines 56-60, Equations. The definitions of some parameters are missing, please add.

Lines 67-69. Please add the reference Fransson et al. (2013) for mentioning the study of brine rejection (with $CO_2$ and $TCO_2$) and effect on the carbonate chemistry in under-ice water (upper 10 m) after the studies by Semiletov et al., (2004); Rysgaard et al., (2007; 2009). Fransson et al. (2013) performed a seasonal study of natural sea ice and under-ice water covering a period from ice formation to ice melt in the Canadian Arctic. I suggest that this reference has to be cited and later discussed.

Line 75. Change (Eq. 3) to (Eq 1).
Line 82. What do you mean with "carbon-bearing materials". Please explain.

Lines 86-88. The carbonate chemistry was examined by Fransson et al. (2013) in the under-ice water where the signal of brine rejection and ikaite was observed at 2 m beneath the sea ice, so please add this information and reference. However, deeper down in the water column, this signal was gone.

Line 93. Change "carbon" parameter to "carbonate" parameters.
Line 94. What is "large enough volume"?
Line 95. Change "a 3 weeks experiment" to "17-days experiment".
Lines 96-97. after "main processes…" please add "…assuming no biological processes" .

Lines 99-109. Tank and experiment descriptions: I suggest adding a table with salinity, temperature, TA, $TCO_2$ of the artificial seawater. Are there any nutrients in the artificial seawater? What has been debated is that ikaite has shown a relationship to nutrient

concentrations (phosphate, nitrate?). I suggest mentioning this in the description of the site and in the discussion, and I suggest adding the reference Hu et al. (2014) for the discussion, where they found that phosphate is perhaps not essential for ikaite precipitation, that was previously thought.

What was the volume of the water in pool at the start (open water) and end (melt) dates? Did you track the changes in volume of the water during ice formation and ice growth, and when removing the seawater samples?

Did you have artificial mixing in the tank? Was the tank water well mixed so that all solid ikaite (and TA) was well distributed in the water column? Did you check if there was solid ikaite at the bottom of the tank or are you sure that all ikaite was well mixed and distributed over the entire water column, and was later dissolved (in the sample)?

How much of the pool ice cover was used for the experiment, sampling all over the ice cover?

Lines 110-114. How was salinity in the seawater sample and melted sea ice measured?

Lines 115-119. Do you have data on wind speed? That is important for the discussion of ice-air $CO_2$ fluxes.

Lines 127-130. Method of samples: the procedure of the TA analyses after removing ikaite shpuld be mentioned in this section and later discussed. Did ikaite dissolve during strorage and analysis of the seawater sample or in the water column? Please define when dissolution took place. This is valid throughout the manuscript.

Did you analysed TA in snow and brine? What about frost flowers? I assume that at the high TA occasions at the ice surface on the 16-17 January and 22-23 January, there were probably brine skim on top of the ice, including TA and maybe ikaite, which may be lost when you remove the snow and/or the ice core. Did you sample the bottom of the tank? Could there have been solid ikaite?

Could solid ikaite have escaped from the sea ice to the underlying water during the collection of sea ice? This was discussed in Fransson et al. (2013) as a possible factor of the high TA values found at 5-15 m under the sea ice, apart from the natural ikaite export from the ice.

Lines 130 and 144. How much $HgCl_2$ did you add to the samples and what was the volume of the sample?

Lines 141-142 and throughout the manuscript. You mentioned that the seawater and melted sea ice samples were stored in +4°C to avoid the dissolution of ikaite. How do you explain why the ikaite was dissolved in the water column under the sea ice?

Lines 153-155. I suggest a figure or table with brine volume for each day during the study. That is needed to understand why and when the ikaite can escape from the sea ice. This specific data should be mentioned in the discussion section as well. Have you checked brine-volume corrected TA?

**Results**

Lines 166-171. The metrological and salinity data is not part of the results, is already presented by Rysgaard et al. (2014) and could be moved to the site description and methods as background data.

Line 191. SD = 8.75 should have only one decimal (due to the accuracy and precision of the measurements), please change to SD=8.8. Do you mean SD="standard deviation"? "variations of …. are quite small", do you mean that "they are almost within the uncertainty of the analytical methods"?

Lines 191-195. I suggest to also write the TA and $TCO_2$ differences from start to end. That helps to understand the figures.

Line 208. Same as earlier, is this melted sea ice with or without ikaite? If this is in melted sea ice including ikaite crystals, you need to clarify the bulk sea ice as "melted (including ikaite)". Please explain and add to method.

Line 208-223. I would like you to present the averaged salinity used for sea ice.

Lines 225-226. Did you measure $CO_2$ ice-air exchange on top of the snow or did you remove the snow? This will give different flux results. Please explain.

Line 227. Add "from source" to get "switched from source to sink for…."

Line 225-228. It seems that from the measured $CO_2$-flux measurements, the sea ice acts as a net $CO_2$ source, and not a net $CO_2$ sink for atmospheric $CO_2$. This is contradictory to what is discussed about sea ice as a $CO_2$ sink. Please explain in discussion section.

Lines 229-230. The references mentioned confirm that the measured $CO_2$ fluxes are in the same order of magnitude. Please add numbers and direction of the $CO_2$ flux in their studies and perhaps discuss more in the discussion section.

**Discussion**
Line 237. What do you mean with "very low"?
Lines 236-238. The biological processes are assumed to have insignificant effect on the carbonate system. Did you check the bacterial activity in bulk sea ice both the start and the end of the experiment? I suggest that this is mentioned in the method description. It would be valuable to relate the estimated microbial activity ($gCL^{-1}h^{-1}$) and algal Chl a ($\mu g\ L^{-1}$) to the changes you measure in $TCO_2$ (in $\mu mol\ kg^{-1}$) to obtain a better idea of the biological impact of $TCO_2$.
What is the biological activity and effect of $TCO_2$ in the underlying water, particularly the microbial activity could be significant? Did you measure microbial activity in the seawater before and after the experiment?

Line 252. Same as earlier about "dissolution of ikaite in water column and sea ice".

Lines 260-261. This statement is not valid as is. Please change to: "Assuming no biological effect, ikaite precipitation/dissolution and gas exchange ($TCO_2$), TA and $TCO_2$ are considered conservative with salinity. Thus we can calculate…"

Line 257. Repetition: a ratio 2:1.
Line 269. Add "assumed to be only due to…" after "…this experiment are..".

Lines 271-274. Please explain better what you mean with "lack of TA". What do you mean with either dissolved or exported out of the sample? What means "exported out of the seawater sample"?

Lines 277-278. "ikaite is precipitated and $CO_2$ released from the ice to the atmosphere ; both processes reduce $TA_{(ice)}$ and $TCO_{2(ice)}$." This statement should be changed since $TA_{(ice)}$ is not reduced by $CO_2$ exchange.

Line 285-286: What is "relatively high sea ice temperatures"? Is this temperature high enough for ikaite dissolution, "likely promote ikaite dissolution"? Please explain. I would think that it is more likely that ikaite is rejected from the sea ice to the underlying water due to increased brine volume and dissolved later (storage, analysis?). It would be good to relate this temperature increase in the sea ice to brine volume values (e.g. >5%) when the brine channels connect to each other and promote solutes and gases to escape from the ice. Presenting the evolution of the brine volume fractions in a table or figure during the study would improve some of the understanding of the results, as was suggested earlier in this review.

Line 296. What do you mean with "good agreement"? Please specify.
Lines 298-300. This sentence could perhaps be moved to the method description.

Lines 311-313. Please add "in this study" between "underlying seawater" and "is the dissolution…". Also add "export of ikaite from the ice" before "dissolution of…" so the sentence will be: " ….carbonate system in the underlying water in this study is the export of ikaite from the ice and dissolution of calcium carbonate". Please change the next sentence to: "While a few studies of ikaite precipitation….".

Lines 315-318. Please add: "according to the study by Fransson et al. (2013)" after where the crystals are dissolved". This study needs to be mentioned since this is one of the first studies describing the carbonate chemistry (such as TA, $TCO_2$) evolution of the sea ice and underlying water (upper 10m) and the sea ice processes such as precipitation and dissolution of ikaite, affecting TA, $TCO_2$ and aragonite saturation from ice formation (in November) to ice melt (in June). They suggested that the high TA found in the upper 10 m under the sea ice was a result of solid ikaite rejected from the ice, dissolved in the water or in the sample before analysis.

Line 319. Please explain how you obtained the 66 $\mu$mol kg$^{-1}$ maximum concentration.

Line 320. Change to "17-days long".

Lines 336-345. I am concerned about the 1-day delay of the measured $pCO_{2sw}$ compared to the $npCO_{2sw}$-normalized values in Figure 3d after turning on the heat. This is unclear to me since this temperature increase should be directly discerned in $pCO_{2sw}$ and it has to be explained or discussed. Why is there a delay?

The sentence "process other than a the temperature change affected the $pCO_{2(sw)}$". Do you have any suggestions on what other processes affected $pCO_{2(sw)}$"?

Lines 355-357. Compare with brine volume fraction.

Lines 358-367 and Figures 8a, b. The calculation procedure is difficult to follow and information on volumes of water and sea ice are missing. I am not convinced why the ikite (mole) in seawater is so large. It is mentioned in the Figure 7c caption (almost same figure as Figure 8a) that "the ikaite is estimated from half of the difference between $TA_{(ice)*}$ and $TA_{(ice)}$", but in the figures it seem that data is not presented as "half". Could you explain?

How was "0 to 43% of ikaite crystals remain" calculated?

Lines 376-377. Please provide numbers of your parameters such as volume, density, and pool dimensions used in the calculations.

Line 380. Was the $CO_2$ fluxes measured on snow and on ice from removed snow?

Line 388. Add "(up to 99% as brine)"…

Line 396-398. What was the wind speed during the study? It would be interesting to know since $CO_2$ fluxes are highly dependent on wind speed.

Line 430-431. This statement is not right. The effect of processes in sea ice such as ikaite precipitation and dissolution affecting the carbonate chemistry and aragonite saturation state (ocean acidification) in the under-ice water has been address in the seasonal study by Fransson et al. (2013). This study should be mentioned. However, in natural sea ice, there is also advection and other processes acting on the under-ice water, which makes the artificial mesocosm experiment a suitable environment to study effects in a more confined and controlled way.

Line 436. "sea ice decreases pH and increases $\Omega_{aragonite}$". Could you please explain why they change in opposite directions?

Lines 438-446. There are few studies such as Chierici el al. (2011) that I suggest should be mentioned in this discussion since this is the first study of the changes of the carbonate chemistry and aragonite saturation state in the underlying water (mixed layer) during a full annual cycle in the Arctic, covering all seasons (autumn, winter, spring and summer). They found relatively low $\Omega_{aragonite}$ in winter under the ice, explained mainly by remineralisation and brine rejection. In spring, $\Omega_{aragonite}$ increased mainly as a result of primary production. Fransson et al. (2013) also studied the carbonate chemistry and $\Omega_{aragonite}$ in underlying water but focused on the upper 10m, showing more of the impacts of sea ice processes.

**Conclusion**

Line 448. "17-day".

Line 451. Change the sentence to "….while export of ikaite from the ice and dissolution of ikaite was the main …."

**Tables**

Table 1. should include more information such as all sampling occasions, not only start conditions.

Table 2. This table could perhaps also include "ikaite (mol) seawater". In the header it should be added "sea ice" in the ikaite (mol) column.

**Figures and figure captions:**

Figure 2. This figure is not the result of this manuscript and has been presented in Rysgaard et al. (2014). I suggest moving it to background information for the site description. Figure 2d is very unclear and it is impossible to discern the different parameters, and should be changed. Figure 2d caption is unclear of what is what with the different colors and depths shown in the figure.  I suggest separating salinity and temperature in two different figures for clarity.

It is difficult to see if the salinity is higher at the end of the experiment or not. This has to be more evident in the method and discussed, if the salinity never returns to start salinity.

Please decide if you use big (A) or small letter (a) in caption and figure, be consistent.

Figure 3. TA* and $TCO_2$* are defined when this figure is referred to. The parameters should be defined in the result section (when the figure is firstly mentioned) to understand the results shown in Figure 3. Figure 3d has also very unclear colors. The "blue" line should be defined in the figure caption. In addition, add and define TA* and $TCO_2$* in the caption (a, b) as well as add the color "black" (a,b) and "red" and "black" and "green" (c) for more consistent presentations of the data.

I am concerned about the 1-day delay of the measured $pCO_{2sw}$ compared to the $npCO_{2sw}$-normalized values in Figure 3d after turning on the heat. This is unclear to me since this temperature increase should be directly discerned in $pCO_{2sw}$ and it has to be explained or discussed. Why is there a delay?

Figure 5. Add "positive air-ice $CO_2$ flux means outgassing from the ice and negative $CO_2$ flux means uptake of atmospheric $CO_2$.

Figure 6. Define the green dotted line in caption.

Figures 7. The figure 7b of changes in $TCO_2$ includes $CO_2$ flux but it does not say in the text.

Figure 7c does not show "half the TA" as I can see. Please explain or I missed something.

Figure 9. What explains the large difference on the 24-25 January between ice-water exchange of $CO_2$ and total $TCO_2$ loss from sea ice?

**Added references:**

Anderson, L.G., E. Falck., E. P. Jones., S. Jutterström and J. H. Swift. 2014 Enhanced uptake of atmospheric $CO_2$ during freezing of seawater: A field study in Storfjorden, Svalbard. JGR Vol. 109, C06004, doi:10.1029/2003JC002120, 2004

Brown et al. (2016)

Chierici, M., Fransson, A., Lansard, B., Miller, L.A., A. Mucci., E. Shadwick., H. Thomas, J E. Tremblay., T. Papakyriakou. 2011. *The impact of biogeochemical processes and environmental factors on the calcium carbonate saturation state in the Circumpolar Flaw Lead in the Amundsen Gulf, Arctic Ocean*. JGR-Oceans. 116, C00G09, doi:10.1029/2011JC007184.

Fransson, A., Chierici, M., Miller, L.A., Carnat, G. Papakyriakou T, et al., 2013. Impact of sea-ice processes on the carbonate system and ocean acidification at the ice-water interface in the Arctic Ocean. 2013. Journal of Geophysical Research-Oceans, 118, 1–23, doi:10.1002/2013JC009164.

Hu, Y., D.A..Wolf-Gladrow, G.S. Dieckmann, C. Völker, G. Nehrke 2014. A laboratory study of ikaite ($CaCO3 \cdot 6H2O$) precipitation as a function of pH, salinity, temperature and phosphate concentration, Marine Chemistry 162 (2014) 10–18, http://dx.doi.org/10.1016/j.marchem.2014.02.003

Nomura D, Assmy P, Nehrke G, Granskog MA, Fischer M, Dieckmann GS, Fransson A, Hu Y, Schnetger B, 2013. Characterization of ikaite ($CaCO32 \cdot 6H2O$) crystals in first- year Arctic sea ice north of Svalbard. Annals of Glaciology, 54(63)doi:10.3189/2013AoJ62A034

---

## Author Comment (AC1) · 13 May 2016

I. General comments

The objective of the paper is to assess the impact of ikaite export from sea ice to the seawater, as clearly indicated by the title. Given that several studies propose that the inorganic carbonate chemistry in sea ice is of major influence for the polar air-sea $CO_2$ exchange, quantifying the fate of ikaite in sea ice during sea ice growth and melt is of great interest.

To address their objective, the authors analyze temperature, salinity, total alkalinity (TA) and dissolved inorganic carbon ($TCO_2$) in both the sea ice and the seawater in an artificial sea-ice pond located at the University of Manitoba, Canada. The method applied to analyze TA and $TCO_2$ in the sea-ice is thought to measure dissolved TA and $TCO_2$ only, specifically excluding TA and $TCO_2$ found in ikaite crystals (1 mol of ikaite contains 2 mol of TA and 1 mol of $TCO_2$). Ikaite concentrations in bulk sea ice and/or its export to the seawater was then derived by computing half of the difference between the theoretical TA concentrations in bulk sea ice if TA were to be conserved with salinity (this assumes no ikaite formation in sea ice) and the observed bulk sea-ice TA (without the ikaite-bond TA concentrations). Similar in seawater, ikaite concentrations are half the difference between the observed TA (which potentially would include imported and dissolved ikaite-bond TA) and the theoretical TA concentrations if TA were conservative with salinity.

Throughout large parts of the manuscript the authors claim that their methods would show ikaite concentrations in sea-ice, which, however is not entirely true since it could also be that ikaite was already exported from sea ice at the time when the observations were taken. In order to quantify the ikaite concentration in sea ice, the authors should have measured bulk sea ice TA concentrations of samples in which ikaite crystals were fully dissolved, and subsequently subtract from these 'new' measurements the already performed measurements of bulk sea-ice TA concentrations without dissolving ikaite. I am not sure to what extent the authors could preform additional measurements to account for this lack in observations.

In the section 5.2 (Estimation of the precipitation-dissolution of ikaite) we clarify our

methods used, based on TA, to estimate the precipitation-dissolution of ikaite within sea ice and seawater. We also add more precision in the section 5.2.1 where we compare our estimation with previous estimation from Rysgaard et al (2014). Then finally, when we try to estimate how much ikaite remain in the ice cover compared to the amount exported to the underlying seawater, we used our estimation and then used estimation from Rysgaard et al (2014), as suggested.

Another way to approximate the exported amount of ikaite from the storage of ikaite in sea ice might be realized by subtracting the concentrations given in Rysgaard et al. 2014 for the same experiment from the observations presented in this study.

We have implemented this suggestion in the manuscript. Thanks.

Despite of not being able to show ikaite concentrations in sea ice, the data presented on ikaite dissolution within seawater clearly supports the conclusion that ikaite is exported from the sea ice to the water column at various rates throughout the course of the experiment. However, the specific conclusion that up to 43 % of ikaite remain in sea ice while the rest is exported to the underlying water column is not supported by the presented data, and might be recomputed e.g. by using the ikaite content as given in Rysgaard et al. 2014.

We have revised our discussion of ikaite export to the underlying seawater in this light. You'll see that this revision is explained in detail in the specific comments.

Furthermore, the results showing that $pCO_2$ below sea-ice remains undersaturated, which the authors relate to ikaite dissolution in sea-ice, is certainly a new results and important for assessing the influence of the inorganic chemistry in sea ice on air-sea ice-sea $CO_2$ exchange processes.

Thank you.

Beyond of analyzing the fate of ikaite in sea ice and seawater, the authors further consideration the influence of ikaite export to seawater on winter ocean aragonite saturation state. This approach is new, however, needs further clarification to understand the applicability of the experimental results on the influence of the process within the present-day and future Arctic Ocean, with the specific comments given in the corresponding section.

We have substantially revised section 5.5 on the impact of sea ice growth on the Aragonite saturation state in Arctic waters. We have addressed specific comments below in this regard.

The overall structure and presentation of the paper is clear, and the language fluent. However, specific parts of the manuscript need some revisions concerning the clarity of the specific statements and the manuscripts needs a thorough read through concerning spelling and grammar mistakes.

We took a better look at the manuscript and hopefully correct the spelling and grammar

mistakes. Thanks for pointing out most of them in your review.

Finally, the sectioning and structure of the manuscript is not following the guidelines for The Cryosphere, and thus needs some more attention.

We have endeavored to correct this in the revised version. Thanks.

After addressing the suggested revisions (see following comments) I think the paper is ready for publication.

We'd like to thank the reviewer for their insightful and constructive comments, which we believe, have improved the manuscript.

II. Specific comments

1. Abstract

The Abstract provides a concise and brief summary. Specific questions, which have arisen in the results sections, potentially need to be re-considered in the Abstract.

L19: How do you derive the uncertainty range of ±3 umol/kg? No details about the uncertainty range are given in section 5.2.1.

Since estimation of ikaite precipitation-dissolution is based on TA, we used the uncertainty of the TA measurement, ±3 $\mu$mol/kg. However, as the precision our TA measurement is clearly mentioned in the "site description, sampling and analysis" section, we have removed this information from the abstract.

L19: The specific percentage of the fraction of ikaite should be derived as detailed in the comments to section 5.3.

We deleted this part as this estimation was wrong. See comments in the section 5.3 further in the review.
We also modified the abstract and conclusion to fit better with our discussion.

2. Introduction

The introduction gives a solid general background on the processes related to the inorganic carbonate chemistry in sea ice and its influence on the air-sea $CO_2$ exchange. Nevertheless, the authors need to state more detailed why it is important to understand and quantify the fate of ikaite in the sea ice, specifically in the second last paragraph (L81-88).

We modify the structure of the introduction to highlight the link between precipitation of ikaite within sea ice and the potential sea ice pump for atmospheric $CO_2$ associated with it, depending on the fate of the crystals within sea ice.

L40-46: "Release of $CO_2$ (from sea ice to the atmosphere)... from open water." → Please be precise whether you mean air-sea-ice fluxes or air-sea fluxes.

We have amended this text for clarity. It now reads:

"Release of $CO_2$ from sea ice to the atmosphere has been reported during sea ice formation from open water (Geilfus et al., 2013a) and in winter (Miller et al., 2011; Fransson et al., 2013) while uptake of $CO_2$ by sea ice from the atmosphere has been reported after sea ice melt onset (e.g. Semiletov et al., 2004; Nomura et al., 2010; Geilfus et al., 2012; Nomura et al., 2013; Fransson et al., 2013; Geilfus et al., 2014; 2015)."

L68: "…in the Arctic...": this should be also true for the Antarctic.

We deleted "in the Arctic" to make it a more general statement.

L79: "…will increase TA...": high TA in meltwater will certainly not increase the seawater TA, since the release of TA upon sea-ice melt dilutes surface ocean TA concentration since bulk sea ice TA concentrations (brine + ikaite) are always lower than TA concentrations of seawater, which is also supported by your data.

We changed the manuscript as followed: "Melting sea ice stratifies surface seawater leading to decreased TA, $TCO_2$ and $pCO_2$, in the sea surface, enhancing air-sea $CO_2$ fluxes (Rysgaard et al., 2007; 2009)."

L87: "However, the…": This sentence would make more sense in the last paragraph, possibly after the second sentence.

We would like to keep it that way. In the last paragraph we are mentioning: "We gain the ability to carefully track carbonate parameters in the ice, in the atmosphere, and in the underlying seawater, while growing sea ice in a large volume of seawater, so that conditions closely mimic the natural system." Which is our solution to the difficulties to detect the signal of carbon component release in a water column of several hundred meters (lines 87-88).

L95-97: This is not a full summary of the paper, since you also look into $\Omega$aragonite.

We added few the following: "We also discuss that dissolution of ikaite crystals exported from sea ice in the underlying seawater can potentially hamper the effect of oceanic acidification on $\Omega_{aragonite}$."

3. Site description, sampling and analysis

L151-153: They way you write this sounds like you do the same as Rysgaard et al. 2014. However, you derive the 'ikaite' concentrations by the difference between (TAice*-TAice)/2, and only compare it to the concentrations estimated from microscope inspection as given in Rysgaard et al. 2014.

We deleted the section "The abundance and concentration of ikaite crystals precipitated within sea ice has been estimated by inspection under microscope as the samples melted." So that it does not seem like we have replicated the method and analysis provided by Rysgaard et al., (2014).

4.1. Sea ice and seawater physical conditions

L187: It seems strange that there is only a salinity stratification, however, in the temperature field the pool seems well mixed. How do you explain this?

Maybe the stratification related to the temperature is too small to be observed. The y-axis ranged in only 1.5°C.

4.2 Carbonate system

It is not really introduced why you look at $n$TA and $n$TCO$_2$. To facilitate reading and understanding you should explain this a little more detailed.

The reason to normalize TA and $T$CO$_2$ is to remove the potential impact of salinity changes on both parameters and to estimate the role of other processes such as precipitation/dissolution of calcium carbonate and/or gas exchanges. We are not sure how to more clearly impress the dependence of TA and $T$CO$_2$ on salinity with more clarity.

L191-192: values given for seawater concentrations of TA and TCO2 at t=0 are different from the values given in Table 1.

The seawater TA and $T$CO$_2$ (TA$_{sw}$ and $T$CO$_{2sw}$) are the concentrations reported during sea ice growth. These values are different from the values reported at t=0 on 11 January (see Table 1, TA = 2453 and $T$CO$_2$=2341) because the sea ice started to grow on 13 January. The differences between 11 Jan (t=0) and 13 Jan (when ice growth commenced) may be due cooling of the seawater, gas exchanges…
We decided not to include these values in the plot because only the plots related to TA and $T$CO$_2$ in seawater will start on 11 January while all the others plots will start on 13 January when data collection commenced.

L200: Additionally provide concentrations of nTA and nTCO$_2$ at t=0.

We have added the values in the table 1. Thanks.

5.1. Key processes affecting the carbonate system

L240: "up to 350 umol/kg", however, in Fig. 7c only up to to 100 umol/kg for the same dataset.

In line 240 of the original manuscript, 350 $\mu$mol kg$^{-1}$ is a maximum reported by Rysgaard et al (2014) using their method of microscopically observing crystals. We mean to state their results here so that our data showing average ikaite concentrations in the ice cover ranged from 0 to 100 $\mu$mol kg$^{-1}$ reported in figure 7 is put in some context. To clarify, we have revised this text to read:
"We compared the direct microscopy observations by averaging the amount of ikaite precipitated throughout the ice thickness for each sampling day from Rysgaard et al., (2014) (Fig. 7c, white dots) with our estimation of the amount of ikaite based on the difference between TA$_{(ice)}$* and TA$_{(ice)}$ (Fig. 7c, black dots)."
We have also revised the figure caption of the Figure 7 for the sake of clarity:
"Evolution of (**a**) TA$_{(ice)}$ averaged throughout the ice thickness at each sampling day (black dots) and TA$_{(ice)}$* (dashed red line) ($\mu$mol kg$^{-1}$) and (**b**) $T$CO$_{2(ice)}$ averaged throughout the ice thickness at each sampling day (black dots) and $T$CO$_{2(ice)}$* (dashed red line) ($\mu$mol kg$^{-1}$), (**c**) Estimation of the ikaite precipitation/dissolution from half of the difference between TA$_{(ice)}$* and TA$_{(ice)}$ ($\mu$mol kg$^{-1}$) (black dots) compared to the average amount of ikaite precipitated throughout the ice thickness for each sampling day from Rysgaard et al., (2014) (white dots). The vertical black dotted line on 26 January mark when the heat was turned back on."

5.2. Estimation of the precipitation-dissolution of ikaite

L260: TCO2 is not conservative with salinity due to potential gas phase of CO2* (Eq. 2)
We revised the text to read:
"Assuming no biological effect, ikaite precipitation/dissolution and gas exchange, TA and $T\mathrm{CO}_2$ are considered conservative with salinity."

L271: Here the explanation of positive and negative signs is wrong: When the difference (TAice*-TAice )/2 is positive, then the observed TA concentration is lower than what would be expected from the theoretical conservation with S. Hence, TA is either in the form of ikaite crystals somewhere in the ice matrix or being exported potentially in ikaite form from the ice matrix to the seawater. In contrast, when the difference is negative, then it implies that more TA is observed in the brine than would be expected from the theoretical conservation with S, indicating dissolution of ikaite in brine. In contrast, in seawater, when the difference (TAsw*-TAsw)/2 is positive then ikaite is imported and dissolved in seawater releasing ikaite-bond TA, and if negative then TA is exported from seawater to somewhere else.
We agree that this section was unclear. We changed the text as: "We assume that the difference between $\mathrm{TA}_{(sample)}^{*}$ and the observed TA is only due to the precipitation or dissolution of ikaite crystals. In case of ikaite precipitation (*i.e.* $\mathrm{TA}_{(sample)}^{*} > \mathrm{TA}_{(sample)}$), half of this positive difference corresponds to the amount of ikaite precipitated within the ice. This ikaite may either remain or may be exported out of the ice. A negative difference (*i.e.* $\mathrm{TA}_{(sample)}^{*} < \mathrm{TA}_{(sample)}$), indicates ikaite dissolution."

L272: "...implies a lack..." this is wrong, it implies that more TA is observed in sea ice. Please rewrite this sentence according to the changes above.
See previous comment for Line 271 above. Thanks.

5.2.1 Sea ice

L279: "…is a result of ikaite precipitation...": it might also be the export of ikaite to the sea-water, see comments to L271 section 5.2, and L292, L302.
See previous comment. For Line 271 above. Thanks.

L296: "...found good agreement with small differences..." This statement is too positive for the time until the 17th of January 2013. As you state in L302 "ikaite crystals could have been formed and then exported into the underlying seawater..." In other words, for this period the comparison between the ikaite concentrations observed in Rysgaard et al. 2014 and your data suggests that your data shows concentration of TA*2 being exported to the seawater rather than the ikaite concentration stored in the sea ice matrix.
We have revised the text to read:
"We compared the direct microscopy observations by averaging the amount of ikaite precipitated throughout the ice thickness for each sampling day from Rysgaard et al., (2014) (Fig. 7c, white dots) with our estimation of the amount of ikaite based on the difference between $\mathrm{TA}_{(ice)}^{*}$ and $\mathrm{TA}_{(ice)}$ (Fig. 7c, black dots). Both ikaite measurements are of the same order of magnitude however the average (22 µmol kg$^{-1}$) and maximum (100 µmol kg$^{-1}$) of direct observations presented by Rysgaard et al. (2014) were lower than our estimated average (40 µmol kg$^{-1}$) and maximum of up to 167 µmol kg$^{-1}$ over

this whole experiment. Deviations are likely due to methodological differences. Here, sea ice samples were melted to subsample for TA and $TCO_2$, Ikaite crystals may have dissolved during melting, leading to an underestimation of the total amount of ikaite precipitated in the ice. However, the difference between $TA_{(ice)}^*$ and $TA_{(ice)}$ provides an estimation of how much ikaite is precipitated in the ice cover, including those crystals potentially already exported to the underlying seawater. The method used by Rysgaard et al., (2014) avoid the bias of ikaite dissolution during sea ice melt with the caveat that crystals need to be large enough to be optically detected. If no crystals were observed, Rysgaard et al., (2014) assumed that no crystals were precipitated in the ice, though ikaite crystals could have been formed and then exported into the underlying seawater prior to microscopic observation of the sample, which may explain the difference observed between both methods during initial sea ice formation (15-18 January) when the ice was still very thin. In addition, the succession of upward percolation events could have facilitated the ikaite export from the ice cover to the underlying seawater. Estimations from both methods show similar concentrations when the ice (i) warmed due to snowfall (18-23 January) and (ii) cooled once the snow was removed (on 23 January). Once the ice started to melt (26 January), Rysgaard et al., (2014) reported a decrease in the ikaite precipitation while in this study we reported a negative difference between $TA_{(ice)}^*$ and $TA_{(ice)}$, possibly indicating that ikaite dissolved in the ice."

L309: "…negative difference ... indicating that ikaite dissolved in the ice or were exported to the water column.": A negative difference between TAice* and TAice does not necessarily mean that ikaite is exported, but it definitively implies that it is dissolved in brine. See comment to L271.

We have revised the text to read:

"Once the ice started to melt (26 January), Rysgaard et al., (2014) reported a decrease in the ikaite precipitation while in this study we reported a negative difference between $TA_{(ice)}^*$ and $TA_{(ice)}$, possibly indicating that ikaite dissolved in the ice."

5.2.2. Water column

L324: Are the values of TCO2sw-TCO2sw* here similar to nTCO2 such as for TA?
I don't understand the question.

L326-328: Please also explain the effects for the sea-ice covered period.
We've added: "During sea ice melt, increased vertical permeability resulting in increased liquid communication through the sea ice volume from below likely in part dissolved ikaite crystals still residing in the ice at that time, and also will have created a downward crystal export mechanism. As the ice melt advanced, patches of open water occurred at the surface of the pool. Therefore, uptake of atmospheric $CO_2$ by the undersaturated seawater likely occurred, increasing the $TCO_{2(sw)}$."

L329-351: Given that the dissolution of ikaite is a fast process in the order of seconds to minutes, a time delay in pCO2 rise of 1 day would suggest that during this day ikaite release to the seawater must have been continuously large. Please discuss this more precisely.

We are not so sure that the dissolution of ikaite crystals occurs that quickly. At -1°C (average temperature of the water column) the dissolution of ikaite will be a slow

process. In Dieckmann et al (2008), the first publication reporting ikaite within sea ice, they melted bulk sea ice over night at +4°C. This method was used by Dieckmann et al (2008, 2010), Geilfus et al (2013), Nomura et al (2013) and this study. This strongly suggests that ikaite dissolution is not a fast process, especially not in a matter of seconds.
We agree that ikaite release from the ice to the water column must have been quite large and happening continuously during the melt. We added this idea in the manuscript in the section 5.3 'Ikaite export from the ice cover to the water column'

5.3. Ikaite export from the ice cover to the water column

L360: "...of ikaite precipitated and remained within the ice cover..." See comment to L271, it could also be the amount of ikaite being exported.
We changed the text accordingly.

L366-367: Since (TAice*-TAice)/2 could be ikaite crystals in sea ice and/or exported ikaite crystals from sea ice it is not straight forward to derive the percentage of ikaite being exported or remained in the sea ice using this dataset. To derive ikaite concentrations in sea ice, you should measure sea ice concentrations with dissolved ikaite and subtract your observations from these new measurements. If this is unfeasible, you might instead consider to use the ikaite concentrations as given in Rysgaard et al. 2014 to compute an estimate of the fraction of ikaite remaining in sea ice.
We did this exercise and change the whole section. It now reads:
"The difference between $TA_{(ice)}^{*}$ and $TA_{(ice)}$ provides an estimation of ikaite precipitated within the ice, including potential ikaite export to the underlying seawater, so it cannot be used to determine how much ikaite remained in the ice versus how much dissolved in the water column. However, Rysgaard et al., (2014) indicate ikaite precipitated within the ice based on direct observations. Using the ikaite concentration reported in Rysgaard et al (2014) (and shown in Fig. 7c), the sea ice volume (in $m^3$) and density, we calculate that 0 to 3.05 mol of ikaite precipitated within the ice cover during sea ice growth (Fig. 8b and Table 2). This amount decreased to 0.46 and 0.55 mol during the sea ice melt (28 and 29 January, respectively). Increased ikaite dissolution in the water column when the ice began to melt (from 11.5 to 20.9 mol) indicates that 9.4 mol of ikaite were stored in the ice and rejected upon the sea ice melt. This amount is about three times the amount of ikaite precipitated in the ice estimated by Rysgaard et al., (2014) at the end of the growth phase (3.05 mol, Table 2), suggesting more work is needed best estimate ikaite precipitation within sea ice."

5.4. Air-seawater exchange of inorganic carbon, attempt of $CO_2$ budget

The calculation of the $TCO_2$ export from ice to water is not clear, in particular, concerning the conversion of the air-seawater $CO_2$ flux to units of mass?
The air-ice $CO_2$ fluxes are presented in mmol $m^{-2}$ $d^{-1}$. As we know the time step between each measurement, we can estimate the number of mole of $CO_2$ exchanged between the ice and the atmosphere for each day in mol $m^{-2}$. From there, using the sea ice thickness and density we can estimate how many moles are exchanged over the whole pool. We added few lines in the manuscript to clarify:

"The number of mole of $CO_2$ exchanges between the ice and the atmosphere were calculated (noted as $CO_{2(air-ice)}$ in Table 2) using the time step between each flux measurement, the ice thickness and density. During sea ice growth 0.01 to 0.42 mol of $CO_2$ were released from the ice-covered pool to the atmosphere. During sea ice melt uptake of atmospheric $CO_2$ by the ice-covered pool ranged from -0.15 to -0.93 (Fig. 9, white triangles)."

Opposed to the definition of ikaite concentrations in sea ice, you here nicely explain the potential processes that cause a negative difference between $TCO_{2ice}$* and $TCO_{2ice}$ in lines 371-375: ..."This could be due to different processes: (i) sea ice released $CO_2$ to the atmosphere, (ii) the precipitation of ikaite within sea ice decreased $TCO_{2(ice)}$ and (iii) sea ice exchanges $TCO_2$ with the underlying seawater." Unfortunately, for calculating the budget, you again assume that $(TA_{ice}*-TA_{ice})/2$ would be ikaite concentrations (see comment to L271, explaining that for TA both (ii) and (iii) are relevant).

Agreed, so we have revised the text to read: "Assuming we know how much ikaite is contained in the ice cover…"

L388: "up to 99 %": please give mean range, or plot results as time series.
We have revised the text here to read: "$TCO_2$ export from the ice to the water column ranged from 23% of the total sea ice $TCO_2$ early in the ice growth (14 January) to 100% after the onset of melt. These estimations are comparable to the study of Sejr et al., (2011) who suggested that sea ice exports 99% of its total $TCO_2$ to the seawater below it."

89: "Between the beginning and the end of experiment, sea ice exported 2.8 mol." The methodology used to derive these values is unclear, it seems as if you give the value at the end of the experiment?

Agreed. We now use the average sea ice export of $TCO_2$ to the underlying seawater and have made this necessary correction in the text:
"On average over the whole experiment, sea ice exported 1.7 mol of $TCO_2$ to the underlying seawater (Fig. 9), which corresponds to a $TCO_{2(sw)}$ increase of 43.5 µmol kg$^{-1}$ considering the average sea ice thickness and density during the experiment and the volume of the pool. However, $TCO_{2(sw)}$ increased by 115 µmol kg$^{-1}$ over the whole experiment (Fig. 3b), leaving an increase of 71.5 µmol kg$^{-1}$ in the $TCO_{2(sw)}$ that cannot be explained by the sea ice-seawater exchange of $TCO_2$. We postulate that as the ice melt advanced, patches of open water that opened at the surface of the pool which were undersaturated compared to the atmosphere (Fig. 3d) imported the additional $TCO_2$ directly from the atmosphere in the form of $CO_{2(g)}$. Considering the pool volume, the 71.5 µmol kg$^{-1}$ increase of $TCO_{2(sw)}$ could be explained by an air-sea water $CO_2$ uptake of 8.5 mmol m$^{-2}$ d$^{-1}$ over 3 days of sea ice melt in a 20% ice free pool. High air-sea gas exchanges rates have been observed over partially ice-covered seas (Else et al., 2011; 2013). This mechanism is also corroborated by models that account for additional sources of turbulence generated by the presence of sea ice (Loose et al., 2014)."

5.5. Impact of sea ice growth on aragonite saturation state of the Arctic Ocean in the

context of ocean acidification

The title of this chapter implies that data from the Arctic Ocean are shown, however, the discussion is solely based on the SERF 2013 data. Hence, the chapter title is misleading.

We changed the title of this section into: "Potential impact of sea ice growth and ikaite export on aragonite saturation state of the underlying seawater.

Generally the data suggests that with or without ikaite export to the seawater the seawater would be supersaturated (Fig. 10), hence, in this experiment calcifying organisms should not face any problems. You should more clearly state that you base your conclusion "potentially hamper the effect of ocean acidification in fall to winter" on the differences between $\Omega$aragonite and $\Omega$aragonite*.

We added this precision and changed the text as followed: "During ice growth, sea ice brine rejection appears to increase both pH (from 8.00 to 8.06) and $\Omega_{aragonite}$ (from 1.28 to 1.65) of the underlying seawater, offsetting the effect of decreased temperature. A slight increase of $\Omega_{aragonite}$ was predicted due to increased salinity and a proportional increase of TA and $T$CO$_2$ as depicted in $\Omega_{aragonite}$*. However, the effect of ikaite rejection and subsequent changes in TA strongly enhance the increase of $\Omega_{aragonite}$."

- L441: Please clarify why $\Omega$aragonite reaches its minimum during winter, since this is not supported by the data on your sea-ice growth-melt cycle. Do other observations suggest this, if yes please refer to them?
  We added references in the text.

- L442: Why do you specifically only state the Arctic Ocean? In other words, do you expect different results for the Antarctic region, and if so, please explain the reasons.
  We deleted "in the Arctic ocean".

- L443-444: "...ice removal acts to impede the effect of ikaite rejection and therefore promote decreased $\Omega$aragonite." This calls for taking into account under-ice ikaite rejection in modeling predictions (...) in the context of sea ice rapid shrinking.": Several modeling studies have addressed the effects of future Arctic sea-ice decay on ocean acidification and report that Arctic surface ocean acidification is related to the rate of sea-ice reduction, and also to the responses of wind mixing and stratification under reduced sea-ice conditions (e.g. Steiner et al. 2014; Yamamoto et al. 2012). Please relate the increase in $\Omega$aragonite due to ikaite export from the ice to the surface ocean to the effects of rising atmospheric CO2 concentrations, global warming and associated Arctic sea-ice decay on $\Omega$aragonite projected by models.
  This will be a really interesting topic, but this is far beyond the scoop of this manuscript.

6. Conclusion

The conclusions drawn are valid and well presented except of one part: The connection between L459-462 and L462-466 is not obvious. Please be more precise here. You should state that any attempt of deriving the air-sea CO$_2$ flux related to the carbon pump

should take into account that ikaite is exported to the underlying ocean during sea ice growth, which might reduce the efficiency of oceanic $CO_2$ uptake upon sea ice melt related to the sea ice pump.

We changed the text accordingly.

III. Technical comments in Figures and Tables Caption

Fig. 1 Give year of experiment.

We add the information.

Fig. 2. Put panel labels in brackets.

Done.

Panel (d): y-axis units for temperature overlain on bracket.

Thanks, we fixed that.

Panel (c) and (d): units for salinity not given.

Salinity do not have units and we did not provide any units through the whole manuscript.

Panel (d): legend is missing information of what is temperature and sanity.

We fixed that.

Caption Fig. 2.
Use lower case letters for panel description.
 "... snow cover": better to use same text as in figure: 'snow thickness'
"... black horizontal bars..": no bars visible: maybe better use 'black shaded areas'
Indicate colors for seawater temperature and sanity in panel (d)
Last sentence, final dot missing.

We did the changes as suggested.

Caption Fig. 3.
Use lower case letters for panel description. Indicate also that red line is calculated on the mean values of the three depth intervals. Legend in panel (a) and (b): no need for 'Sw' abbreviation. One space between depth value and unite.

We did the changes as suggested.

Caption Fig. 4.
Use lower case letters for panel description.

We did the changes as suggested.

Caption Fig. 6.
Use lower case letters for panel description.

We did the changes as suggested.

Caption Fig. 7.
Use lower case letters for panel description.
One space after the fist $TA_{(ice)}$

We did the changes as suggested.

Fig. 8.
y-axis labels: "Ikaite" This is to imprecise, see comments to L271.

The figure caption of figure 8 now reads:
"Evolution of (**a**) ikaite dissolution within the water column (in $\mu$mol kg$^{-1}$), (**b**) mass of ikaite dissolved in the underlying seawater (blue), mass of ikaite precipitated in sea ice (black) estimated from this study and estimated from Rysgaard et al., (2014) (white). The vertical black dotted line on 26 January mark when the heat was turned back on."
The figure is showing the amount of ikaite within sea ice or sweater, no matter if it dissolved, precipitated or exported. Therefore, "Ikaite" as label of the y-axis should be enough.

Caption Fig. 8.
Use lower case letters for panel description.
"..amount of mol of ikaite.." mol is the unit: '..mass of ikaite in mol..'
We did the changes as suggested.

Fig. 9.
A flux cannot have the unit mol.
"Total $T$CO$_2$ lost by the sea ice" This is the mass of $T$CO$_2$ of ice cover assuming the absence of ikaite.
"Air-ice CO$_2$ fluxes" cannot have units of mol
We changed it into: amount of CO$_2$ exchanges between the atmosphere and the ice cover (CO$_{2air\text{-}ice}$, white triangle).

Caption Fig. 9.
A flux cannot have the unit mol.
"Total $T$CO$_2$ exchanges by the ice cover" This is the mass of $T$CO$_2$ of ice cover assuming the absence of ikaite and CO$_{2(g)}$ in bubbles.
As mentioned in the manuscript:
The total amount of $T$CO$_{2(ice)}$ lost from the ice cover is estimated by the difference between $T$CO$_{2(ice)}^{*}$ and $T$CO$_{2(ice)}$.
And
According to Figure 7b, the ice cover always had lower $T$CO$_{2(ice)}$ during the experiment ($T$CO$_{2(ice)}^{*}$ > $T$CO$_{2(ice)}$) compared to what would be expected if the CO$_2$ simply followed brine rejection in a conservative process (i.e. $T$CO$_{2(ice)}^{*}$). This could be due to different processes: (i) sea ice released CO$_2$ to the atmosphere, (ii) the precipitation of ikaite within sea ice decreased $T$CO$_{2(ice)}$ and (iii) sea ice exchanges $T$CO$_2$ with the underlying seawater.
Therefore, the total exchanges of TCO$_2$ from the ice take into account the precipitation of ikaite and the gas exchanges.

Fig. 3. To 5. And Fig. 7. To 9
x-axis text: use for all plots the same x-axis text, e.g. the same as used in Fig. 2.:"Day of January 2013".
We did the change as suggested.

Table 1.
Please also give the water column conditions just before the heat was turned ON, and at the end of the experiment.

We followed the suggestion.

Table 2.

"amount": better use 'mass'

"ikaite": See comment to Fig. 8. and L271.

"$CO_2$ fluxes (mol)" fluxes are defined e.g. to be changes in mass over time. How do you derive these values, please clarify this in the text, see comment to section 5.4.

The figure caption now read:

"Masses of $TCO_2$ in the water column ($TCO_{2(sw)}$) and in the ice cover ($TCO_{2(ice)}$), masses of ikaite within the ice cover estimated from this study and from Rysgaard et al., (2014), masses of ikaite dissolved in the water column (Ikaite$_{(sw)}$) and masses of $CO_2$ exchanged between the ice and the atmosphere over the whole pool (estimation based on the air-ice $CO_2$ fluxes). All units are in mole."

IV. Technical comments

L18-19: "... ikaite precipitated…", better: '… ikaite concentrations of up to..'
   Thanks for the correction.

L19: "… within sea ice; up to…": wrong '… within sea-ice; at least...'
   This sentence does not exist anymore.

L29: "Each year...": Be more precise: e.g. 'Currently, each year..'
   Thanks for the suggestion.

L30: "... through primary production and surface cooling…": please be slightly more precise here explaining the biological pump and the dissolution pump.
   If the reader wants to know more about the biological and dissolution pump, he could use the references provided in the manuscript.

L33: Please give references, and clarify if you refer only to model and/or observational estimates.
   I'm not sure what the reviewer is referring to and there are a couple of references in the sentence.

L33: "… sea ice an impermeable…" grammar mistake: '… sea ice as an impermeable…'
   Thanks for the correction.

L47: comma missing after "formation"
   Thanks for the correction.

L50: please give reference.
   I'm not what the reviewer is referring to as the end of the sentence L50 has a reference.

L51: "…inorganic carbon..." → give abbreviation TCO2
   No, inorganic carbon refer here to either $TCO_2$ or $CO_2$ (dissolved or as gas bubbles), which is all "inorganic carbon".

L56: "…TA is reduced...": Be more precise: e.g. ..'TA in brine is reduced...'

Thanks for the correction.

L57: "...while TCO2 is...": Be more precise: e.g. '...while TCO2 in brine is..'
Thanks for the correction.

L59: "[CO2]": Spelling mistake: [CO2*]: and define in text what this means
The equation as shown in the original manuscript is correct and is displayed as in Zeebe and Wolf Gladrow (2001), $CO_2$ in seawater: equilibrium, kinetics, isotopes.

L60: "[B(OH-)]": Spelling mistake: B(OH)4-]
Thanks for the correction.

L60: "[H-]": Spelling mistake: [H+]
Thanks for the correction.

L63: Also re-mention the pathway of brine to the sea-ice surface as was given in lines 47-48.
We won't repeat something wrote less than 15 lines above. In addition, the reading of the text will be less smooth.

L67: "...fluxes...": Be more precise: e.g. '…air-sea fluxes..'
Thanks for the suggestion.

L67: "…seawater and incorporated…": Grammar mistake: e.g. '…seawater and its contribution to intermediate and deep-water formation..'
Thanks for the correction.

L77: "…melting sea ice promotes…": Be more precise: e.g. '...surface warming and melting sea ice promote..'
We choose to keep "melting sea ice". Sea ice can melt from the top (surface melting) due to the increase of the radiation and from the bottom (warmer seawater, as during this experiment). Both cases lead to the melt of the ice. And the sea ice melt promotes the stratification of the underlying seawater (as during this experiment).

L80: "…underlying seawater...": be more precise: e.g. surface ocean → due to the enhanced summer stratification L78.
Thanks for the suggestion.

L90: "435 m3": Give dimensions rather than the volume.
The dimensions are provided few lines below, in the section "Site description, sampling and analysis".

L167: "…15 cm on 18 January...": mistake: '..15 cm until 18 January...'
If it please to the reviewer.

L170: "…This results in the increase of the sea ice temperature…": Be more precise: e.g. 'This resulted in repeated events of increased sea-ice temperatures..'
We followed the suggestion.

L197-198: it would be more intuitive to call it 'sw' and 'ice' instead of 'sample'. Delete brackets around 'sw', and 'ice', put brackets around 't' and do not write 't' in subscript.
We decided to use 'sample' in the formulas (4) and (5) to make them valid for both seawater and sea ice. In the formula description, we specifically explained that 'sample' is either sea ice or seawater. The use of the brackets around 'sw' and 'ice' is

to be consistent with the writing of $TA_{(ice)}$ and/or $TA_{(sw)}$ used through the whole manuscript.

L201: "…ice started the melt..": mistake: '…ice started to melt..'
Thanks for the correction.

L225: "...suggest...": this is rather weak, better to write that the measurements 'show' this.
Thanks for the suggestion.

L228: Is is not appropriate to give a range of two values.
Agreed, we have made the appropriate correction.

L264-265: in the equation 'sample' means 'sw or ice'. It would facilitate reading if you would write for each component a separate equation.
We decided to use the notation "sample" to avoid the repetition of the same equation and we will keep it that way.

It is better to not put "sw" and "ice" within brackets, while the time variable should be written not in subscript and in brackets.
We put "sw" and "ice" within bracket to be consistent with the notation of $TA_{(ice)}$, $TA_{(sw)}$, $TCO_{2(ice)}$ and $TCO_{2(sw)}$ thought the whole manuscript.

L270: "assume" this sounds like as if another value than the one-half would also make sense. You should clearly explain here were the one-half comes from: 1 mol of ikaite contains 2 mol of TA.
The text now read: "The difference between $TA_{(sample)}^{*}$ and the observed TA is only due to the precipitation or dissolution of ikaite crystals."

L278: "...assume…" see comment to L270
The text now read: "Half the difference between $TA_{(ice)}^{*}$ and $TA_{(ice)}$ is a result of ikaite precipitation (Fig. 7c, black dots)."

L299: "…precipitate...": grammar mistake: '…precipitated..'
Thanks for the correction.

L304: "…differences observed": be more precise: e.g. '…differences between both methods…'
If it please to the reviewer.

L318: "…to the amount of ikaite...": be more precise: e.g. '...to the concentration of ikaite..'
If it please to the reviewer.

L317: "... half the difference between $TA_{(sw)}^{*}$ and $TA_{(sw)}$ ..." mistake: '... half the difference between $TA_{sw}$ and $TA_{sw}^{*}$…'
No, the text as shown in the manuscript is correct, we are always looking at the difference between $TA_{(sw)}^{*}$ and $TA_{(sw)}$.

L319: "…amount…": imprecise formulation: '...concentration...'
Thank you.

L322: "... TA(sw)* and TA(sw) ..": see comment L317

See previous comment.

L345: "…processes other than a the temperature..." → delete the 'a'
Thanks for the correction.

L346: Changes in $pCO_2$ cannot not understood from equation 3.
Right, we changed it to equation 1.

L367: "…crystals remain contain within...": grammar mistake: '.. crystals remain within..'
Thanks for the correction.

L379: Fig. 9 not 8
Thanks for the correction.

L379: "maximum outgassing": logical mistake: '… maximum loss of $TCO_2$…'
Thanks for the correction.

L386: "... substracting … ikaite precipitation to the total": grammar mistake: '…precipitation from the total...'
Thanks for the correction.

L392: "Fig 3c" → Fig. 3B
Thanks for the correction.

L402: "...convection..." This should include all kind of transport mechanisms e.g. advection, mixing
We changed it into "mixing" which we take to include convection, diffusion, advection etc.

L400: "... measurement…": spelling mistake: '…measurements..'
Thanks for the correction.

L424: "... area ..": spelling mistake: '... areas ...'
Thanks for the correction.

L426: "… as a result of respiration.": missing process: '… as a result of respiration and dissolution..'
The undersaturation in respect with aragonite can't be enhanced by the dissolution of calcium carbonate…

L438: "…dramatically...": formulation too dramatic
We changed dramatically for strongly.

L438: "…increase the...": grammar mistake 'increase...'
Thanks for the correction.

L439: "This suggest that...": spelling mistake: 'This suggests that..'
Thanks for the correction.

L444: "… and therefore promote decreased..": grammar mistake: '… and therefore promotes a decrease of...'
Thanks for the correction.

L457: "up to 66..." → '… up to 128…'

Thanks for the correction.

L462: Please give more references
The reference provided is the original reference regarding this problematic. There is no need for additional references.

L470: "… is responsible the...": '… is responsible for the'
Thanks for the correction.

L471: "…we project that...": '… we discuss that..'
Thanks for the correction.

References

Yamamoto, A., Kawamiya, M., Ishida, A., Yamanaka, Y., and Watanabe, S.: Impact of rapid sea-ice reduction in the Arctic Ocean on the rate of ocean acidification, Biogeosciences, 9, 2365-2375, 2012.

Rysgaard, S., Wang, F., Galley, R. J., Grimm, R., Notz, D., Lemes, M., Geilfus, N. X., Chaulk, A., Hare, A. A., Crabeck, O., Else, B. G. T., Campbell, K., Sørensen, L. L., Sievers, J., and Papakyriakou, T.: Temporal dynamics of ikaite in experimental sea ice, The Cryosphere, 8, 1469-1478, 2014.

Steiner, N.S., Christian, J.R., Six, K.D., Yamamoto, A., Yamamoto-Kawai, M. 2014. Future ocean acidification in the Canada Basin and surrounding Arctic Ocean from CMIP5 earth system models, J. Geophys. Res.-Oceans, 119, 332-347, 2014.

---

## Author Comment (AC2) · 13 May 2016

**Review on Geilfus et al. (2016)**

Geilfus et al. (2016) discuss data from a most interesting sea ice formation (and a bit of melting) experiment performed at the Sea-ice Environmental Research Facility (SERF) site from 13 to 30 January 2013 at the University of Manitoba, Winnipeg, Canada. Several articles have been published already using data from this experiment (Hare et al., 2013; Rysgaard et al., 2014, Else et al., 2015). Geilfus and colleagues use measurements of total alkalinity (TA), dissolved inorganic carbon ($TCO_2$ total $CO_2$), salinity, temperature, and a few other measurements to estimate the carbon budgets in sea ice and the underlying (artificial) sea water, especially the precipitation, transfer, and dissolution of ikaite. The conservative components of the marine carbonate system, namely TA and $TCO_2$, vary due to three processes: (1) Change in salinity due to formation and melting of sea ice, (2) precipitation or dissolution of calcium carbonate, here in the form of ikaite, and (3) gas-exchange. The size of the processes can be estimated in the following sequence: (1) can be quantified by scaling TA and $TCO_2$ using salinity (Eqs. 6 & 7). (2) can be estimated from changes of TA whereby the amount of calcium carbonate precipitation (and associated $TCO_2$ decrease) is equal to half of the TA reduction; the dissolution of calcium carbonate precipitation has the opposite effect. (3) The residual $TCO_2$ variation should be due to gas-exchange, which might be, however, difficult to estimate because of uncertainties when calculating small differences.

The data (TA, $TCO_2$, T, S) seem to be of high quality, however, a detailed discussion of the time evolution of measured and derived quantities is largely missing; often only wide ranges ('0.47 to 26.71 mol') are given. A proper analysis of the data, estimates of uncertainties, identification of surprising or contradicting findings and a proper overall budget (How to close the TA budget?) for the whole pool is largely missing. Thus I cannot recommend publication.

General comments & suggestions:

Units: the partial pressure of $CO_2$, $pCO_2$, should be given in μatm (and not ppm; ppm refers to the mixing ratio of $CO_2$, $xCO_2$).

We have replaced "ppm" with "μatm" in each instance where the $p$CO$_2$ is discussed in the manuscript. Thanks.

Which program/package do you apply for carbonate system calculations? Which equilibrium values do you use? For a recent discussion compare Orr, Epitalon & Gattuso (2015).

We made our carbonate system calculations using the CO2sys_v2.1.xls spreadsheet [*Pierrot et al.*, 2006] with the dissociation constants from Goyet and Dickson (1989) and others constants advocated by DOE (1994). We refer the reviewer to line 434 of the original manuscript. This information is still present in the revised version of the manuscript (L 502).

Specific comments & suggestions:

1. L 30: $CO_2$ emissions & oceanic uptake: Sabine et al., 2004 is an excellent paper, however, I suggest to cite more recent estimates (for example, IPCC 2013, or Global

Carbon Project)

Thanks for the suggestion. In the IPCC 2013 report, the introduction for Ocean biogeochemical changes, including Anthropogenic Ocean Acidification (Chap. 3, Observation Oceans, pages 291) read: "The oceans can store large amounts of $CO_2$. The reservoir of inorganic carbon in the ocean is roughly 50 times that of the atmosphere (Sabine et al., 2004)". So we feel that the Sabine citation is original work and therefore the most relevant one to use in this instance.

2. L 31: 5-14% of the global ocean $CO_2$ uptake: based on which values?

We've added the actual Tg C $yr^{-1}$ values to the text.

The manuscript now reads: "The Arctic Ocean plays a key role in these processes, taking up from -66 to -199 Tg C $year^{-1}$, contributing 5-14% to the global ocean $CO_2$ uptake (Bates and Mathis, 2009), primarily through primary production and surface cooling (MacGilchrist et al., 2014)."

3. L 47-48: 'During the earliest stages of sea ice formation a small fraction of $CO_2$-supersaturated brine is expelled upward onto the ice surface promoting a release of $CO_2$ to the atmosphere (Geilfus et al., 2013a).' It might be interesting to elaborate a bit more on 'expelling brine': When does it occur? How much brine can be expelled? Level of $CO_2$-supersaturation? Salinity of the expelled brine?

Not much is known about upward brine expulsion. During sea ice formation, salty brine is expelled upward to the ice surface, but mainly downward into the ocean below. Often the formation of a brine skim is associated with the formation of frost flowers. A complete description of frost flowers formation during the same experiment as this study could be found in Galley et al (2015), Micrometeorological and thermal control of frost flower growth and decay on young sea ice, Arctic, vol 68, n°1, pp79-92. In Galley et al., (2015), salinity of the brine skim is up to 85. However higher salinity (100<) are reported in Geilfus et al (2013) or Barber et al (2014), doi:10.1002//2014JD021736.

If we know about the salinity of the brine skim, we don't know how much brine is expelled upward compared to what is expulsed downward. However, the millimetric layer of brine skim reported at the surface of the ice and its ephemeral nature during the very onset of ice growth suggests that brine is mainly exported downward.

4. L 50: 'physical concentration'??? I suggest dropping 'physical'

We followed the suggestion, thanks.

5. L 60: Eq. (3) is an approximation to the TA definition given by Dickson (1981). In your experiment you use a special form of artificial seawater (ASW). It would be interesting how much total borate is in the ASW and how this is taken into account in the calculation of $pCO_2$ from TA and $TCO_2$.

We have modified equation 3 to fix a mistake in the original version. As mentioned in the original manuscript (line 105), the composition of the ASW can be found in Rysgaard et al., (2014). The borate concentration in the ASW was not measured, and therefore we can't discuss its influence on TA and $TCO_2$. Since 96.5% of the carbonate in seawater is accounted for by carbonate and bicarbonate (eq 3), while the rest is comprised of protons, hydroxides as well as borate, silicate and phosphate, we are confident in the assessment of TA presented here.

6. L 78-80 'The mixing of meltwater, that is low in $TCO_2$, $pCO_2$, and high in TA due to brine dilution and ikaite dissolution, with seawater will increase TA and decrease the $pCO_2$ of the underlying seawater, enhancing the air-sea $CO_2$ fluxes (Rysgaard et al., 2007; 2009).' $pCO_2$ of seawater is not a 'substance' that can be 'mixed': it is the equilibrium partial pressure of seawater and does not follow a linear mixing relationship. $TCO_2$ in meltwater is low compared to (artificial) seawater. Meltwater $pCO_2$ is low compared to atmospheric $CO_2$ because of low $TCO_2$ and not enough time for gas-exchange and equilibration with the atmosphere. I don't know why meltwater TA should be higher than in ASW, because the ikaite was precipitated from ASW and then dissolves again.

When sea ice melts, it does not return it its original seawater composition. Melt water is different from seawater in many ways, including its TA. It is also substantially less saline, for example.

We changed the text in the manuscript as followed: "Melting sea ice stratifies surface seawater leading to decreased TA, $TCO_2$ and $pCO_2$, in the sea surface, enhancing air-sea $CO_2$ fluxes (Rysgaard et al., 2007; 2009)."

7. L 92-95 'We gain the ability to carefully track carbon parameters in the ice, in the atmosphere, and in the underlying seawater, while growing sea ice in a large enough volume of seawater, so that conditions closely mimic the natural system.'

However, there are various differences to the natural system; to name only a few: no leads for heat & gas-exchange, no horizontal ice movement impacting mixing of the underlying water, no 'biology' (which here simplifies the analysis of the carbonate system), the pressure build-up during the first part of the experiment. These differences should be mentioned and possible consequences for data interpretation should be discussed, especially with respect to comparison with the real world.

It is true that the SERF mesocosm does not exactly mimic the natural environment. The main goal of the SERF, and of this experiment is to have a simplified or constrained version of an ice-covered ocean. We aim in this way to gain an improved understanding of inorganic carbon dynamics from the initial sea ice growth to its melt. Once the physical-chemical processes are completely understood given the constrained SERF system, we will endeavour to add complexity to the SERF system in future experiments.

We have conducted this experiment in a way to purposefully exclude biology from the system to focus on the physical and chemical controls of the carbonate system. In particular, we aimed to i) determine what the main processes responsible for the changes in the inorganic carbon system during a event of sea ice growth and melt and ii) determine the exchanges between the ice, the underlying seawater and the atmosphere. We were mainly focused on the precipitation of ikaite within sea ice and its fate in the system in order to follow on and augment the results of previous and concurrent SERF experiments (e.g. Rysgaard et al., 2014). In this regard, SERF is a made useful by the "constraints" it imposes, for example its volume is fixed. This allows us to look at the potential exchange between the ice cover and the underlying seawater, which so far has proved too complicated to do in the natural environment.

8. L 104 '(ASW) formulated by dissolving large quantities': formulated $\Rightarrow$ generated, fabricated

We changed "formulated" to "made ".

9. L189-191 'TA and $TCO_2$ in seawater, noted as $TA_{(sw)}$ and $TCO_{2(sw)}$, were sampled at the sea ice-seawater interface, 1.25 and 2.5 m depth. However, as the variations of TA and $TCO_2$ over the 3 depths are quite small (SD = 8.75 and 4.5 $\mu$mol kg$^{-1}$, respectively), we consider the average concentration.' Do you really mean 'variations' of TA (with a standard deviation of 8.75 $\mu$mol kg$^{-1}$) or differences of TA between the 3 levels. If the latter: give mean difference ± SD.
   The text now reads: "An ANOVA test over the 3 depths revealed that the means are not statistically different (p<0.01) so we consider the average concentration of the three depths in the following analysis."

10. L 204-205' The $pCO_{2(sw)}$ then oscillated from 360 to 365 ppm during sea ice growth.' ⇒'The $pCO_{2(sw)}$ then varies from 360 to 365 $\mu$atm during sea ice growth.'
    Thanks we have made the suggested change.

11. L 219 'minimums' ⇒ minima
    Thanks we have made the suggested change.

12. L224-228: Air-ice $CO_2$ fluxes: Although it's good to know the ranges of $CO_2$-fluxes, in the current context it would be even more interesting the fluxes integrated over time.
    We can't integrate the fluxes over time as if more fluxes were measured, we will have more $CO_2$ released to the atmosphere, which doesn't make sense. In the section 5.4, we averaged the fluxes over the whole experiment to estimate how much $CO_2$ is exchange between the ice and the atmosphere and in the table 2, the number of mole of $CO_2$ exchanges are indicated day by day.

13. L 238-240 'For this 2013 experiment, Rysgaard et al. (2014) discussed the precipitation of ikaite within the ice cover in detail, reporting high concentrations of ikaite (> 2000 $\mu$mol kg$^{-1}$) at the surface of the ice and ikaite precipitation up to 350 $\mu$mol kg$^{-1}$ in bulk sea ice.' The concentrations, especially at the surface, are impressive. In the current context (TA and $TCO_2$ budgets for the whole pool) it would be good to obtain integrated values, at least rough estimates.
    Ikaite precipitation concentrations have been integrated through the entire ice thickness and are provided in the original manuscript at the figure 7, as explained in the manuscript and in the figure caption. They are also reported in Rysgaard et al (2014).

14. L 244 please drop 'Therefore'
    Thanks we have made the suggested change

15. L 255 please drop 'However,'
    Thanks we have made the suggested change

16. L 256-257 Try to avoid repetition ('2:1 ratio'): 'As illustrated in Figure 6, an exchange of $CO^*$ does not affect TA while the precipitation- dissolution of ikaite affect TA and $CO_2$ in a ratio 2:1.'
    We want to keep this precision to make sure that any reader (even those not familiar with this concept) will understand how and why we can estimate the precipitation-dissolution of ikaite through the changes in TA observed during this experiment.

17. L 271-274 'A negative difference (i.e. TA(sample)$^*$ < TA(sample)), implies that a lack of TA is observed in the sample compared to what is expected based on the observed salinity changes (Fig. 2). This suggests that ikaite crystals were either dissolved or exported out of the sample (sea ice or seawater).' difference = TA(sample)$^*$ - TA(sample). I don't understand the sentence: 'negative difference' means TA(sample) > TA(sample)$^*$, i.e. there is more TA in the sample than expected from salinity scaling; dissolution of ikaite (that was imported from somewhere else) would indeed increase TA; export of ikaite (that has been precipitated in the sample) would imply a decrease of sample TA.

    This section was unclear. If $TA_{(sample)}^*$ is higher than $TA_{(sample)}$ (positive difference), it implies that a process is responsible for decreasing the $TA_{(sample)}$. In this case study, ikaite precipitation will decrease $TA_{(sample)}$. If the difference between $TA_{(sample)}^*$ and $TA_{(sample)}$ is negative, this suggests a process is responsible for increased $TA_{(sample)}$, in our case that will be the dissolution of ikaite crystals.

    We changed the text as follows: "We assume that the difference between $TA_{(sample)}^*$ and the observed TA is only due to the precipitation or dissolution of ikaite crystals. In case of ikaite precipitation (*i.e.* $TA_{(sample)}^* > TA_{(sample)}$), half of this positive difference corresponds to the amount of ikaite precipitated within the ice. This ikaite may either remain or may be exported out of the ice. A negative difference (*i.e.* $TA_{(sample)}^* < TA_{(sample)}$), indicates ikaite dissolution."

18. L 278 '... both processes reduce and $TCO_{2(ice)}$': outgassing of $CO_2$ (one of the two processes) does not change $TA_{(ice)}$, please rewrite sentence accordingly.

    We changed the text as followed: "Greater $TA_{(ice)}^*$ and $TCO_{2(ice)}^*$ compared to the averaged observed $TA_{(ice)}$ and $TCO_{2(ice)}$ (Fig. 7a, b) are expected as ikaite is precipitated and $CO_2$ released from the ice to the atmosphere (Fig. 5, 6). Half the difference between $TA_{(ice)}^*$ and $TA_{(ice)}$ is a result of ikaite precipitation (Fig. 7c, black dots)."

19. Figure 7: (1) $TA_{(ice)}^*$ looks like you have continuous (or at least many) measurements. Please give some info.

    As explained in the manuscript, $TA_{(ice)}^*$ is calculated based on the actual measurements. We changed the representation to make it easier to understand for each reader.

    (2) I'm wondering how much of the difference between $TCO_{2(ice)}^* - TCO_{2(ice)}$ can be explained by ikaite precipitation alone and suggest to show this in another panel added to the Fig. 7.

    We have a full discussion regarding the exchanges of $TCO_2$ between the ice cover, the underlying seawater and the atmosphere (Section 5.4). In this section we discuss how much $CO_2$ is released to the atmosphere and how much is exchanged with the underlying seawater when ikaite precipitation is subtracted from that calculation in the budget. We therefore refrain from adding additional panels to Figure 7.

20. Table 1: to display 4 values only, a table is not required, however, it would be good to extend the table and give values of $TA_{(sw)}$, $TCO_{2\ (sw)}$, $TA_{(ice)}$, $TCO_{2\ (ice)}$, $S_{(sw)}$, $T_{(sw)}$, $S_{(ice)}$, $T_{(ice)}$ for the time points at which you took $TA_{(ice)}$ samples.

    The goal of the table 1, as explained in the original manuscript on line 247 and in the table caption is to show the initial seawater conditions at the beginning of the

experiment prior to any sea ice formation (at t=0, the origin point) on 11 January.
The table now shows the pool conditions at t=0, (11 January), on 25 January (prior to the beginning of sea ice melt) and on 29 January (at the end of the experiment), as asked by reviewer 1. We refrain from extending the table as requested by this reviewer as that would be duplication of the data already found in the figures provided.

21. L 286-288 'The upward percolation of seawater observed from 15 to 18 January might complicate the picture of the effect of sea ice temperature on ikaite formation.'I bit more detailed description what happened here would be useful (or can it be found somewhere else, reference?).
On lines 166-171 (of the original manuscript) please note that we discuss this in greater detail:
"The air temperature at the beginning of the experiment ranged from -2°C to -26°C, which initiated rapid sea ice growth to 15 cm until 18 January (Fig. 2). During this initial sea ice growth, the sea ice was attached to the side of the pool resulting in the development of a hydrostatic pressure head that caused percolation of seawater at the freezing point upwards through the sea ice volume as the sea ice grew downwards. This resulted in repeated events of increased sea ice temperature from the bottom to the surface observed between 15 and 18 January (Fig. 2)."

15 to 18 January is the period with large differences $TA^{*}_{(ice)}$ - $TA_{(ice)}$, $TCO_2{}^{*}{}_{(ice)}$-$TCO_{2(ice)}$, and large discrepancy between estimates of ikaite precipitation by Rysgaard et al. (2014) and the current investigation (Fig. 7).
We discuss the differences between our methods and estimation by Rysgaard et al (2014) in the original manuscript, from line 293 to line 305, with a specific focus on the beginning of the sea ice growth (15-18 January). Please also see the next comment for further precision.

22. L 293-297 'So, we compared the direct microscopy observations by averaging the amount of ikaite precipitated throughout the ice thickness for each sampling day from Rysgaard et al., (2014) (Fig. 7c, white dots) with our estimation of the amount of ikaite based on the difference between $TA^{*}_{(ice)}$ and $TA_{(ice)}$ (Fig. 7c, black dots) and found good agreement, with some small differences likely due to methodological differences.' Please give a correlation coefficient.
We have amended the text as follows to clarify what was initially meant by good agreement:
"We compared the direct microscopy observations by averaging the amount of ikaite precipitated throughout the ice thickness for each sampling day from Rysgaard et al., (2014) (Fig. 7c, white dots) with our estimation of the amount of ikaite based on the difference between $TA_{(ice)}{}^{*}$ and $TA_{(ice)}$ (Fig. 7c, black dots). Both ikaite measurements are of the same order of magnitude however the average (22 µmol kg$^{-1}$) and maximum (100 µmol kg$^{-1}$) of direct observations presented by Rysgaard et al. (2014) were lower than our estimated average (40 µmol kg$^{-1}$) and maximum of up to 167 µmol kg$^{-1}$ over this whole experiment. Deviations are likely due to methodological differences. Here, sea ice samples were melted to subsample for TA and $T$CO$_2$, Ikaite crystals may have dissolved during melting, leading to an

underestimation of the total amount of ikaite precipitated in the ice. However, the difference between $TA_{(ice)}^*$ and $TA_{(ice)}$ provides an estimation of how much ikaite is precipitated in the ice cover, including those crystals potentially already exported to the underlying seawater. The method used by Rysgaard et al., (2014) avoid the bias of ikaite dissolution during sea ice melt with the caveat that crystals need to be large enough to be optically detected. If no crystals were observed, Rysgaard et al., (2014) assumed that no crystals were precipitated in the ice, though ikaite crystals could have been formed and then exported into the underlying seawater prior to microscopic observation of the sample, which may explain the difference observed between both methods during initial sea ice formation (15-18 January) when the ice was still very thin. In addition, the succession of upward percolation events could have facilitated the ikaite export from the ice cover to the underlying seawater. Estimations from both methods show similar concentrations when the ice (i) warmed due to snowfall (18-23 January) and (ii) cooled once the snow was removed (on 23 January). Once the ice started to melt (26 January), Rysgaard et al., (2014) reported a decrease in the ikaite precipitation while in this study we reported a negative difference between $TA_{(ice)}^*$ and $TA_{(ice)},$ possibly indicating that ikaite dissolved in the ice."

23. L 298-301 'During melting of the sea ice samples, ikaite crystals may have dissolved, leading to an underestimation of the total amount of ikaite precipitate [precipitation] in the ice. This bias is avoided during direct microscopic observation of the crystals (Rysgaard et al., 2014) if crystals are large enough to allow optical detection.' Do you see a significant difference in the mean values of ikaite precipitation estimated by the two methods?
See previous comment.

24. L 315-317 'According to equations 1 to 3, lower $TA_{(sw)}^*$ and $TCO_{2\,(sw)}^*$ compared to $TA_{(sw)}^*$ and $TCO_{2\,(sw)}^*$ (Fig. 3b, c) confirm the dissolution of ikaite in the underlying seawater.' Eqs. (1)–(3) do not contain the quantities $TA_{(sw)}^*$ and $TCO_{2\,(sw)}^*$: please rewrite accordingly
Now the manuscript reads: "Lower $TA_{(sw)}^*$ and $TCO_{2(sw)}^*$ compared to $TA_{(sw)}$ and $TCO_{2(sw)}$ (Fig. 3) confirm the dissolution of ikaite in the underlying seawater as the dissolution of ikaite crystals will decrease both TA and $TCO_2$ (equations 1 to 3)."

25. Fig. 8A does not make sense to me because you compare ikaite precipitation and dissolution using concentrations in one reservoir (sea ice) which shows large relative changes in volume and in another huge reservoir (seawater). I suggest to drop Fig. 8A.
Figure 8a shows ikaite precipitated in the ice cover (black diamonds) and the dissolution of ikaite in the underlying seawater (blue triangle), both expressed in $\mu$mol kg$^{-1}$. It may not be appropriate to show both the ikaite precipitation within sea ice and the ikaite dissolution within the water column in a single plot due to the difference in reservoir. Since ikaite precipitation within sea ice is already illustrated in the figure 7, we decided to remove this information from figure 8a, but we choose to keep the figure because it is the only plot illustrating ikaite dissolution in the water column.

26. According to Fig. 8B much more ikaite has been dissolved in seawater than

precipitated in sea ice: What's your explanation?

We add some precision about this in the manuscript in the section 5.3:

"The estimation of ikaite dissolution in the pool is significantly higher than the estimated amount of ikaite precipitated (and potentially exported) within the ice cover, especially during sea ice melt. Within the ice cover, the ikaite values presented here represent a snapshot of the ikaite content in the ice at the time of sampling. In the underlying seawater, ikaite dissolution increased $TA_{(sw)}$ cumulatively over time."

27. L 338-340 'Using the equation from Copin-Montegut (1988), we normalized the $pCO_{2(sw)}$ to a temperature of -1°C (noted as $npCO_{2(sw)}$, blue line on Fig. 3d).'No motivation is given for this 'normalization' and I don't see why to do so. Once again: $pCO_{2(sw)}$ is not a substance. The gas-exchange depends on the actual $pCO_{2(sw)}$ (strongly dependent on temperature!).

The lines 334-338 of the original manuscript read: "The $pCO_{2(sw)}$ is highly correlated with the seawater temperature (Fig. 2) with a rapid decrease of $pCO_{2(sw)}$ during the first days of the experiment (13 to 15 January) and a relative constant $pCO_{2(sw)}$ until 27 January. However, on 26 January, the heat was turned back ON affecting the seawater temperature on the same day (Fig. 2) while the impact on the $pCO_{2(sw)}$ only appeared one day later (Fig. 3d)."

This led us to normalize the data because as you say and as it's mentioned above, the $pCO_2$ is strongly dependent on the temperature. So the question we want to answer is why, when we heated the pool, the in-situ $pCO_2$ didn't change. To answer that question, we normalized the $pCO_2$ to a temperature of -1°C to remove the effect of temperature.

28. L361 'Within the water column, 0.47 to 26.71 mol of ikaite dissolved.' Please give a proper discussion of the evolution in time (Fig. 8B) and how this evolution is related to various processes. What might have caused the drop of ikaite dissolution in seawater around 20 January? How to close the TA budget? Compare also Fig. 3

We changed section 5.3 "Ikaite export from the ice cover to the water column", to include a discussion about the evolution of the amount of ikaite dissolved in the water column and some possible conclusions.

In the first submitted manuscript, we did not attempt to do a TA budget... We did try to close a $TCO_2$ budget, including the ikaite precipitation-dissolution, air-ice gas exchanges and ice-seawater $TCO_2$ exchange. Unfortunately, as stated in the manuscript we could not close the budget and uncertainty in the methods were too big (see your comment 30). This conclusion will be the same in an attempt to do a TA budget.

29. L 375-377 'To estimate the amount of $TCO_2$ exchanged during this experiment, we convert mol $kg^{-1}$ to moles, using the sea ice (and seawater) thickness (in meter) and density (in kg/m3) and the pool dimension (in meter).' This is not just a conversion of units! Instead of concentrations you consider reservoir contents!

The text now read: "To estimate the amount of $TCO_2$ exchanged during this experiment, we convert our units to moles, using the sea ice (and seawater) volume (in $m^3$) and density (in $kg/m^3$)."

30. L 418-419 'Using the seawater conditions at the end of the experiment, a layer of

1cm of seawater in the pool contains 4.21 mol of TCO$_2$, making it difficult to close our budget.'It's good that you mention this uncertainty. I would like to see more uncertainty estimates in the manuscript.
Thanks

**References**

. [1] Dickson, A.G. An exact definition of total alkalinity and a procedure for the estimation of alkalinity and total inorganic carbon from titration data. *Deep Sea Research Part A. Oceanographic Research Papers*, 28(6):609–623, 1981.

. [2] Else, BGT and Rysgaard, Søren and Attard, Karl and Campbell, K and Crabeck, O and Galley, RJ and Geilfus, N-X and Lemes, M and Lueck, R and Papakyriakou, T and F. Wang. Under-ice eddy covariance flux measurements of heat, salt, momentum, and dissolved oxygen in an artificial sea ice pool. *Cold Regions Science and Technology*, 119:158–169, 2015.

. [3] Geilfus, N.-X. and Galley, R. J. and Else, B. G. T. and Papakyriakou, T. and Crabeck, O. and Lemes, M. and Delille, B. and Rysgaard, S. Impacts of ikaite export from sea ice to the underlying seawater in a sea ice-seawater mesocosm. *The Cryosphere Discussions*, 2016:1–33, 2016.

. [4] Hare, AA and Wang, Fei and Barber, Dave and Geilfus, N-X and Gal- ley, RJ and Rysgaard, Søren. pH evolution in sea ice grown at an outdoor experimental facility. *Marine Chemistry*, 154:46–54, 2013.

. [5] Orr, J.C., J.-M. Epitalon, and J.-P. Gattuso. Comparison of ten packages that compute ocean carbonate chemistry. *Biogeosciences*, 12(5):1483– 1510, 2015.

. [6] Rysgaard, Søren and Wang, F and Galley, RJ and Grimm, Rosina and Notz, Dirk and Lemes, M and Geilfus, N-X and Chaulk, A and Hare, AA and Crabeck, O and others. Temporal dynamics of ikaite in experimental sea ice. *Cryosphere*, 8:1469– 1478, 2014.

---

## Author Comment (AC3) · 13 May 2016

Review of the manuscript by Geilfus et al. Impact of ikaite export from sea ice to underlying seawater in a sea ice-seawater mesocosm

The manuscript describes a mesocosm experiment with artificial sea ice and seawater and the precipitation of ikaite and the impact of the exported ikaite on the underlying water using the SERF artificial outside seawater tank, the University of Manitoba in Winnipeg, Canada. The authors show data and results mainly as the changes and evolution in measured seawater TA and $TCO_2$ and salinity-normalized TA and $TCO_2$ during a 17 day-period. Measured air-ice $CO_2$ exchange during the study is also presented. The investigation of sea ice processes and underlying water in a confined setup in an outside environment with mainly the processes of salinity changes, ikaite precipitation/dissolution and $CO_2$ gas exchange affecting the carbonate chemistry (assuming insignificant effect of biological processes) is new and interesting. However, the idea of solid ikaite export to the water column and the effect of ikaite on the underlying water such as aragonite saturation state has been presented and discussed in a few publications, which should be referred to. These publications also describe sea ice processes and evolution of the sea ice and underlying water in natural sea ice. However, the estimates of the amount of ikaite exported out from the sea ice to the water beneath compared to the ikaite precipitation in sea ice are new and valuable. I think it is an interesting approach and important study in a controlled environment but it needs improvements. There are too many unclear calculations, figures, statements and missing uncertainty discussions. Hence, the manuscript requires substantial revision and cannot be published in its present form. However, I encourage publication after major revision.

*General comments:*

Parts of the results are not convincing with measured TA in the seawater being higher at the end of the experiment (melt) than at the start of the experiment.

We stopped the experiment when sea ice was still present in the tank, which explain why the initial seawater conditions are not met at the end of the experiment. We have endeavored to make this clear in the revised version of the manuscript.

Important discussion and uncertainty investigations are missing regarding the contradictions of the results.

There are no contradictions of the results presented here as far as we can tell.

Some figures are unclear, and calculations are not well described and are sometimes difficult to follow and reconstruct, such as the mole calculations of ikaite as well as the result of 57% of ikaite exported from the ice.

We have made every attempt to clarify these and other points in the specific comments below.

Essential data are missing and a description of the evolution of the TA in the underlying water is missing. The uncertainty discussion on ikaite dissolution during analysis and not in the water column is missing and not mentioned in the method section. There are also unclear explanations of some of the contradicting results.

Essential data are not missing from the work, and we see no specific comments to that effect. Revision of the section regarding in the seawater TA in light of the comments of the other two reviewers should clarify this for the reviewer.

I also have concerns about the statement and conclusion about ikaite dissolution in seawater as ikaite probably does not dissolve at temperatures <0°C, such as the temperature in the underlying water. The seawater samples were stored at +4°C so the ikaite was probably dissolved or near dissolution before or at analysis, and not in the water column.

Rysgaard et al., (2014) simulated the precipitation of ikaite using SERF seawater and the FREZChem model (Marion 2001) and show that ikaite will start precipitate at -4°C. This suggests that the temperature of the water column, during the whole experiment, is warm enough to dissolve ikaite crystals.

Regarding our methods to estimate the ikaite precipitation within sea ice, we used the same technique as presented by Dieckmann et al., (2008) who is the first to report ikaite within sea ice. We recognize that melting sea ice at +4°C over night is probably not the best method, but it is widely used in different studies (e.g. Dieckmann et al 2008, 2010, Geilfus et al 2013 and Nomura et al 2013). We can't affirm that no ikaite dissolution take place during the melt of the ice samples. This is also discussed in the section "estimation of the precipitation-dissolution of ikaite" as a possible bias of the method. However, the melt of the ice samples is fast (happen over night) compared to the duration of 17 days for the whole experiment where the ikaite have more time to dissolved.

The water column temperatures were between -3°C and -7°C during the study and about -1°C at the end of the study.

The water column temperature can't be ranging from -3 to -7°C, that's sea ice (see figure 2).

There is lack of information on temperature, salinity, TA and $TCO_2$ at the end of the experiment when the ice was melted. This should be mentioned in the method and discussion sections, to be able to close the TA seawater budget from start to end, which seems to be a problem.

We stopped the experiment when sea ice was still present in the tank which explain why we are not going back to the initial salinity in the pool and why we can' close the TA budget. In the first submitted manuscript, we did not attempt to do a TA budget. We did try to close a $TCO_2$ budget, including the ikaite precipitation-dissolution, air-ice gas exchanges and ice-seawater $TCO_2$ exchange. Unfortunately, as stated in the manuscript we could not close the budget and uncertainty in the methods were too big. This conclusion will be the same in an attempt to do a TA budget.

The seawater salinity and TA could change during the study since freshwater in the form of sea ice is removed every time an ice core is collected, and same for seawater.

We did some estimation and the lost of water due to the sea ice/seawater sampling is negligible (see further comments).

To estimate how much water is removed from the pool due to sea ice sampling, we consider that we collected 5 ice cores during each sampling day. Therefore, we will remove 103 L of seawater out of the pool. According to the dimension of the pool (line 100, L= 18.3 m, l=9.1 m and depth= 2.6m) 103 L corresponds to 0.023% of the total volume of the pool. The impact on the salinity and TA will be negligible.

What about TA and $TCO_2$ in the snow, brine and brine-skim, where these analyzed?

These factors could be discussed if they impact the results and maybe also used correcting the calculations.

We did not measure TA and $TCO_2$ in the snow or in the brine, and data on TA and $TCO_2$ on brine skim and frost flowers were too scarce to conclude anything and were therefore not presented in this study.

In parts of the result, the air-sea $CO_2$ flux is not considered and left out in the statement of processes when calculating the changes in $TCO_2$, which is an important process driving the changes in $TCO_2$ (except for biological production) although with relatively small effect. However, this is later discussed in the manuscript.

We disagree with the reviewer about the location of the air-sea flux discussion, leaving it where we feel it's most applicable and pertinent in the discussion about the $CO_2$ fluxes and how they affect the $TCO_2$ exchanges we make the $TCO_2$ budget.

The information on wind speed is missing, it is essential for gas exchange to occur between ice and air. Metrological data could perhaps be presented in a table and moved to site description/method since this is not a result of this paper and already presented by Rysgaard et al. (2014).

The wind data presented in more than one other paper to which we refer in the work and are not presented here because we used the chamber technique to measure the $CO_2$ flux at the sea ice surface. The reviewer will understand of course that the chamber technique, prevents the impact of wind on the flux measurement. See a more complete response in the specific comments section.

Important and highly relevant references are missing in the introduction and discussion sections, such as Fransson et al. (2013) and Chierici el al. (2011), which performed the first studies of the carbonate chemistry and aragonite saturation (ocean acidification) in natural sea ice and underlying water during a full ice season in the Arctic. I suggest that these references are cited and mentioned in the discussion section. There are other relevant references that I suggest to be included, see Specific comments.

Thanks for the references; we made sure to cite these works properly in the manuscript.

The manuscript would benefit from language correction by English native person.

We've endeavored to improve the written English in the work.

*Specific comments*

Line 1. The title may not inform the reader what this manuscript is about. I suggest changing "Impacts" (on what?) to "Estimates" or "indications".

We changed it into "estimates".

**Abstract**

Line 12. This sentence suits better in introduction, it is not the result of the manuscript. I suggest removing the sentence and start the next sentence with "The fate ....".

The first sentence now reads:" The precipitation and fate of these ikaite crystals within sea ice is still poorly understood."

Line 14 and throughout the manuscript. As far as I can see, the experiment was performed during 17 days, not month-long experiment or three weeks, as is also written at various places in the manuscript. Please change to "17 days long" or just mentioned the dates.

We have made the proposed revision.

Lines 16, 20, 25 and throughout the manuscript: "dissolution of ikaite" has to be explained or used properly. Ikaite will probably not dissolve in the cold water (<0°C), so please add information to explain what you mean with "dissolved ikaite". You may write "presence of ikaite dissolved during analysis". Perhaps you have proofs on the dissolved ikaite in the underlying water (before storage or analysis), then please add that information.

See previous comments.

**Introduction**

Lines 35-36. The references mentioned, do they report on sequestration of atmospheric $CO_2$ below the mixed layer or only into the surface mixed layer? Do they have evidence that the ice-brine pump actually exports atmospheric $CO_2$ below the mixed layer (i.e. sequestration for longer periods)? There are other studies (not so recent) of $CO_2$ sequestering which are more relevant, e.g. $CO_2$ uptake in the Arctic Ocean due to brine rejection (e.g. Anderson et al., 2004) from brine rejection, and (and very recent) Brown et al. (2016) that may also be referenced? There may also be modeling results. Maybe use other reference or change to "sequestration into the mixed layer below the ice".

Thanks for the reference of Brown et al (2016), I did not see the final version of the publication yet as it is just accepted for publication (end of March 2016).
We changed our sentence as suggested.

Line 39. I suggest removing "$CO_2$", and start the sentence with "The carbonate chemistry...". What do you mean with "heterogeneous"? Do you mean that the distribution or concentrations are heterogeneous?

We followed the suggestion and change the sentence by: "The carbonate chemistry in sea ice and brine is spatially and temporally variable, which leads to complex $CO_2$ dynamics with the potential to affect the air-sea $CO_2$ flux (Parmentier et al., 2013)."

Lines 41-45. I suggest to add the reference of Fransson et al. (2013) for both $CO_2$ release in winter and $CO_2$ uptake during ice melt.

Reference added

Line 46. What is the sea ice pump, please explain why and how $pCO_2$ is controlled?

We changed the structure of the text to make sure the sea ice pump is clearly explained: "The specific conditions leading to ikaite precipitation as well as the fate of these precipitates in sea ice are still not fully understood. Ikaite crystals may remain within the ice structure while the $CO_2$ formed during their precipitation is likely rejected with dense brine to the underlying seawater and sequestered below the mixed layer. During sea ice melt, the dissolution of these crystals triggered by increased ice temperatures and decreased bulk ice salinity will consume $CO_2$ and drive a $CO_2$

uptake from the atmosphere to the ice. Such mechanism could be an effective sea ice pump of atmospheric $CO_2$ (Delille et al., 2014). In addition, ikaite stored in the ice matrix could become a source of TA to the near-surface ocean upon its subsequent dissolution during sea ice melt (Rysgaard et al., 2007; 2009)."

Line 53. Please add the reference Nomura et al. (2013) (after Dieckmann et al., 2008), they also found ikaite crystals in natural Arctic sea ice.
Reference added.

Lines 54-55. Please add the reference Fransson et al. (2013).
Reference added.

Lines 56-60, Equations. The definitions of some parameters are missing, please add.
Yes, thanks we now define all parts of the equations that are mentioned in the text.

Lines 67-69. Please add the reference Fransson et al. (2013) for mentioning the study of brine rejection (with $CO_2$ and $TCO_2$) and effect on the carbonate chemistry in under-ice water (upper 10 m) after the studies by Semiletov et al., (2004); Rysgaard et al., (2007; 2009). Fransson et al. (2013) performed a seasonal study of natural sea ice and under-ice water covering a period from ice formation to ice melt in the Canadian Arctic. I suggest that this reference has to be cited and later discussed.
Reference added.

Line 75. Change (Eq. 3) to (Eq 1).
Thanks for the correction.

Line 82. What do you mean with "carbon-bearing materials". Please explain.
Now the sentence reads: "One of the major unknowns is the fate of ikaite, $TCO_2$ and $CO_2$ released from sea ice during winter."

Lines 86-88. The carbonate chemistry was examined by Fransson et al. (2013) in the under- ice water where the signal of brine rejection and ikaite was observed at 2 m beneath the sea ice, so please add this information and reference. However, deeper down in the water column, this signal was gone.
Thanks for the reference. The work of Fransson et al (2013) show how difficult it is to detect the signal of carbon components release into the water column, as stated in our manuscript. Especially as they only found evidences on 4 stations compared to the 18 stations sampled in their studies (Chap. 50, p20 on the Fransson et al (2013) publication).

Line 93. Change "carbon" parameter to "carbonate" parameters.
Thanks for the correction.

Line 94. What is "large enough volume"?
We deleted the word "enough". Now the sentence reads: "We gain the ability to carefully track carbonate parameters in the ice, in the atmosphere, and in the underlying seawater, while growing sea ice in a large volume of seawater, so that conditions closely mimic the natural system."

Line 95. Change "a 3 weeks experiment" to "17-days experiment".

We changed it into "During this experiment'.

Lines 96-97. After "main processes..." please add "...assuming no biological processes".
Thanks, we have made the recommended addition.

Lines 99-109. Tank and experiment descriptions: I suggest adding a table with salinity, temperature, TA, $TCO_2$ of the artificial seawater. Are there any nutrients in the artificial seawater? What has been debated is that ikaite has shown a relationship to nutrient concentrations (phosphate, nitrate?). I suggest mentioning this in the description of the site and in the discussion, and I suggest adding the reference Hu et al. (2014) for the discussion, where they found that phosphate is perhaps not essential for ikaite precipitation, that was previously thought.
We did not measure any nutrients in the artificial seawater. Since they were not measure during this experiment, it's difficult make any meaningful comments on the subject and so we refrain from doing so.
Table 1 shows the seawater conditions at 3 stages of the experiment: 1) the initial seawater conditions prior to sea ice formation 2) at the end of sea ice growth prior to melt and 3) the last measurements made in the pool, once the ice was melting.

What was the volume of the water in pool at the start (open water) and end (melt) dates? Did you track the changes in volume of the water during ice formation and ice growth, and when removing the seawater samples? Did you have artificial mixing in the tank? Was the tank water well mixed so that all solid ikaite (and TA) was well distributed in the water column? Did you check if there was solid ikaite at the bottom of the tank or are you sure that all ikaite was well mixed and distributed over the entire water column, and was later dissolved (in the sample)?
We did not track the changes of volume in the pool during the experiment. However, considering that we collected 5 ice cores during each sampling day, we will remove 103 L of seawater out of the pool. According to the dimension of the pool (line 100, L= 18.3 m, l=9.1 m and depth= 2.6m) 103 L corresponds to 0.023% of the total volume of the pool.
We did have artificial mixing in the tank. This detail was added to the manuscript in the "Site description" section: "Four 375 W pumps were installed on the bottom of the pool at each of the corners to induce a consistent current. The pumps were configured to draw water from their base and then propel it outward parallel to the bottom of the pool. The pumps were oriented successively at right angles to one another, which created a counterclockwise circulation of 2-3 cm s$^{-1}$ (Else et al., 2015)."
The pool was well mixed as suggested by the T and S profile observed during the experiment (as explained line 184-187) so the distribution of TA and ikaite should be homogeneous.
We only had access to the bottom of the pool in the spring once the pool was drained, having no mechanism to look for ikaite crystals while the pool was full of water.

How much of the pool ice cover was used for the experiment, sampling all over the ice cover?
We add the precision in the manuscript and a reference to Else et al (2015) who presented a schematic view of SERF. The manuscript now reads: "Sea ice and seawater samples were obtained from a confined area on the North side of the pool to

minimize effects on other experiments (e.g. Else et al., 2015).”

Lines 110-114. How was salinity in the seawater sample and melted sea ice measured?

The manuscript, lines 184 reads: “Bulk ice and seawater samples salinity was measured on bulk ice and seawater samples using a Thermo Orion 3-star with an Orion 013610MD conductivity cell and values were converted to bulk salinity (Grasshoff et al., 1983).”

Lines 115-119. Do you have data on wind speed? That is important for the discussion of ice- air $CO_2$ fluxes.

The wind could be an important component in the amplitude of the air-ice $CO_2$ fluxes measured by eddy covariance. However, in this study air-ice $CO_2$ fluxes were measured using the chamber technique in the purposeful absence of wind. Therefore we can't link the magnitude of the air-ice $CO_2$ fluxes to any wind speed.

Lines 127-130. Method of samples: the procedure of the TA analyses after removing ikaite should be mentioned in this section and later discussed. Did ikaite dissolve during storage and analysis of the seawater sample or in the water column? Please define when dissolution took place. This is valid throughout the manuscript.

The lines 127-130 states on how we took seawater samples with a peristaltic pump through an ice core hole. We did not remove any ikaite crystals from our seawater samples and never mentioned anything like that in the manuscript.

It may be possible that ikaite was present as crystals in the seawater samples and dissolved therein during storage. Since both TA and $TCO_2$ were measured adding acid in the samples, in all likelihood any ikaite "present" in the sample will have been dissolved.

Did you analysed TA in snow and brine?

We did not measure TA in snow or brine.

What about frost flowers? I assume that at the high TA occasions at the ice surface on the 16-17 January and 22-23 January, there were probably brine skim on top of the ice, including TA and maybe ikaite, which may be lost when you remove the snow and/or the ice core.

We did some measurements of TA and $TCO_2$ in the frost flowers. But we don't have enough data to support anything, which is why this is not presented in the manuscript. Yes the high TA and $TCO_2$ reported on 16-17-22-23 January are due to the presence of brine skim. Which is why, in the manuscript we are linking these high concentration to the high salinity at the surface of the ice.

Did you sample the bottom of the tank? Could there have been solid ikaite?

At the end of the experiment we did not look for ikaite or any other precipitates at the bottom of the pool. In addition, the heating coil sitting at the bottom of the pool will make the dissolution of ikaite very likely. The pool was ice-free for a few days after the end of our measurement period before another experiment took place at SERF. We only had access to the bottom of the pool in the spring once the pool was drained.

Could solid ikaite have escaped from the sea ice to the underlying water during the collection of sea ice? This was discussed in Fransson et al. (2013) as a possible factor of

the high TA values found at 5-15 m under the sea ice, apart from the natural ikaite export from the ice.

The section from Fransson et al (2013) discussing the possible lost of ikaite during the ice core collection read: "However, if we assume that bacterial respiration occurred in the entire ice core during the study period, this would result in more negative $CO_2$-gas flux ($C_T$ loss) or less positive flux. In addition, the effect of solid $CaCO_3$ may be underestimated due to the loss of $CaCO_3$, $A_T$ and brine at ice-core extraction."

I believe the authors are referring to the possibility of lost of brine, and by extension ikaite, due to core extraction from the ice cover. This is of course a possibility, as you are pulling an ice core out of the ice cover, you may lose brine, gases and ikaite crystals. This is not a new problem in sea ice research and we, like many others, have not developed a coring method that overcomes this problem.

According to Rysgaard et al., (2012) brine loss during the core extraction could be approximately 10 percent (±5 %) based on unpublished data collected during the IPY-CFL project in 2008.

Lines 130 and 144. How much $HgCl_2$ did you add to the samples and what was the volume of the sample?

We now include the requested information as follows:

"Samples were stored in 12 ml gas-tight vials (Exetainer, Labco High Wycombe, UK) and poisoned with 12 $\mu$l of saturated $HgCl_2$ solution and stored in the dark at 4°C until analysed."

Lines 141-142 and throughout the manuscript. You mentioned that the seawater and melted sea ice samples were stored in +4°C to avoid the dissolution of ikaite. How do you explain why the ikaite was dissolved in the water column under the sea ice?

No, the lines 141-142 reads: "The bagged sea ice samples were then melted in the dark at 4°C to **minimize** the dissolution of calcium carbonate precipitates (meltwater temperature never rose significantly above 0°C)." We can't affirm that no ikaite dissolution take place during the melt of the ice samples. This is also discussed in the section "estimation of the precipitation-dissolution of ikaite" as a possible bias of the method.

Lines 153-155. I suggest a figure or table with brine volume for each day during the study. That is needed to understand why and when the ikaite can escape from the sea ice. This specific data should be mentioned in the discussion section as well. Have you checked brine- volume corrected TA?

We added a figure panel on the figure 2 showing the brine volume concentration in the ice cover during the whole experiment. We add some text in the section "sea ice and seawater physical conditions".

I'm not sure to understand what is the brine-volume corrected TA…

**Results**

Lines 166-171. The metrological and salinity data is not part of the results, is already presented by Rysgaard et al. (2014) and could be moved to the site description and methods as background data.

Agreed. However, we present these data in a section called "sea ice and seawater

physical conditions". This section is needed to make sure the reader knows how the physical conditions of both the ice cover and the underlying seawater are evolving during the experiment. This section leads directly on from the methods section at the very beginning of the results.

Line 191. SD = 8.75 should have only one decimal (due to the accuracy and precision of the measurements), please change to SD=8.8. Do you mean SD="standard deviation"? "variations of .... are quite small", do you mean that "they are almost within the uncertainty of the analytical methods"?

We changed that part and made an ANOVA test to confirm the TA and $TCO_2$ means of the 3 depths are not statistically different. The manuscript now reads: " We performed an ANOVA test over the 3 depths and the means are not statistically different (p<0.01). Therefore we will consider the average concentration."

Lines 191-195. I suggest to also write the TA and $TCO_2$ differences from start to end. That helps to understand the figures.

Both values of TA and $TCO_2$ at the beginning and at the end of the experiment are already given in the text (L 191-195 of the original manuscript) and in the table 1.

Line 208. Same as earlier, is this melted sea ice with or without ikaite? If this is in melted sea ice including ikaite crystals, you need to clarify the bulk sea ice as "melted (including ikaite)". Please explain and add to method.

In the methods section, we state that we are melting bulk sea ice samples (ikaite still included within the ice samples), we do not know of a method to remove the ikaite from the samples without melting them.

Line 208-223. I would like you to present the averaged salinity used for sea ice.

In this section we refrain from mentioning the averaged bulk ice salinity. Here we present TA and $TCO_2$ in bulk sea ice and the normalized TA and $TCO_2$ in sea ice (noted as nTA and $nTCO_2$), so mention of average bulk salinity would likely muddy the water for the reader. The only time we mention average sea ice salinity is when we introduce the calculation of the expected TA and $TCO_2$ based on sea ice salinity. The manuscript specifies that TA, $TCO_2$ and S are averaged throughout the ice cover where that information is pertinent.

Lines 225-226. Did you measure $CO_2$ ice-air exchange on top of the snow or did you remove the snow? This will give different flux results. Please explain.

We did not remove the snow cover from the ice to measure the air-ice $CO_2$ fluxes. The snow removal just at the location of the chamber will not make our estimation of the air-ice $CO_2$ fluxes representative for the whole ice covered pool. In addition, removing the snow will allow the ice to cool down quite rapidly (as illustrated in the figure 2, when we removed the snow on 23 January), promoting a release of $CO_2$ from the ice to the atmosphere. The exact role of the snow cover in term of air-ice $CO_2$ fluxes is not well known and is worth dedicated studies, but that is beyond the scope of this work.

We changes the manuscript as followed (to include the notion of snow over sea ice): "The $CO_2$ fluxes measured at the variably snow-covered sea ice surface (see Figure 2b), ranged from 0.29 to 4.43 mmol m$^{-2}$ d$^{-1}$ show that growing sea ice released $CO_2$ to the atmosphere (Fig. 5)."

Line 227. Add "from source" to get "switched from source to sink for...."

Thanks for the correction.

Line 225-228. It seems that from the measured $CO_2$-flux measurements, the sea ice acts as a net $CO_2$ source, and not a net $CO_2$ sink for atmospheric $CO_2$. This is contradictory to what is discussed about sea ice as a $CO_2$ sink. Please explain in discussion section.

In these lines we mean to indicate that we found that sea ice was a source of $CO_2$ to the atmosphere during growth and a sink for atmospheric $CO_2$ during melt. In the discussion, we attempted to do a $TCO_2$ budget. We mention the ice cover is, on average over the duration of the experiment, releasing 0.08 mol of $CO_2$ to the atmosphere. This is consistent with the reported measure of air-ice $CO_2$ fluxes from the lines 225-228.
The ability for the ice to act as a sink or source for atmospheric $CO_2$ is not only linked to the air-ice exchange of $CO_2$, but also to where this $CO_2$ is going. In this manuscript we also calculate how much $TCO_2$ is exported from the ice to the underlying seawater and we confirmed that sea ice primarily export $TCO_2$ to the water column.

Lines 229-230. The references mentioned confirm that the measured $CO_2$ fluxes are in the same order of magnitude. Please add numbers and direction of the $CO_2$ flux in their studies and perhaps discuss more in the discussion section.

We have added the necessary amendments to the new text, which now reads: "These ranges of air-ice $CO_2$ exchanges are of the same order of magnitude as fluxes reported on natural sea ice using the same chamber technique in the Arctic during the initial sea ice growth (from 4.2 to 9.9 mmol m$^{-2}$ d$^{-1}$ in Geilfus et al., 2013) and during the spring-summer transition (from -1.4 to -5.4 mmol m$^{-2}$ d$^{-1}$ in Geilfus et al., 2015). In Antarctica air-ice $CO_2$ fluxes were reported during the spring-summer transition from 1.9 to -5.2 mmol m$^{-2}$ d$^{-1}$ by Delille et al (2014), from 0.3 to -2.9 mmol m$^{-2}$ d$^{-1}$ (Geilfus et al., 2014) and from 0.5 to -4 mmol m$^{-2}$ d$^{-1}$ (Nomura et al., 2013)."

**Discussion**

Line 237. What do you mean with "very low"?

Very low was poor word choice, so we have revised the text to read: "During this experiment, neither organic matter nor biota were purposely introduced into the pool; the observed range of bulk ice microbial activity (5.7 x 10$^{-9}$ on 14 January to 7.5 x 10$^{-7}$ g C L$^{-1}$ h$^{-1}$ on 21 January) and algal Chl $a$ (0.008 on 14 January to 0.002 µg L$^{-1}$ on 21 January) were too low to support any biological activity (Rysgaard et al., 2014). Therefore biological activity is unlikely to have played a role."

Lines 236-238. The biological processes are assumed to have insignificant effect on the carbonate system. Did you check the bacterial activity in bulk sea ice both the start and the end of the experiment? I suggest that this is mentioned in the method description. It would be valuable to relate the estimated microbial activity (gCL$^{-1}$h$^{-1}$) and algal Chl a (µg L$^{-1}$) to the changes you measure in $TCO_2$ (in µmol kg$^{-1}$) to obtain a better idea of the biological impact of $TCO_2$. What is the biological activity and effect of $TCO_2$ in the underlying water, particularly the microbial activity could be significant? Did you measure microbial activity in the seawater before and after the experiment?
We added the following info into the method section:

"Bulk ice samples for biological measurements were collected between 14 and 21 January. Filtered (0.2 μm) SERF seawater (FSW) was added at a ratio of 3 parts FSW to 1 part ice and the samples were left to melt in the dark. Chlorophyll *a* was determined on three occasions by filtering two aliquots of the melted ice sample onto GF/F filters (Whatmann brand) and extracting pigments in 10 ml of 90% acetone for 24 h. Fluorescence was measured before and after the addition of 5% HCl (Turner Designs Fluorometer) and Chl *a* concentration was calculated following Parsons et al. (1984). Measurements of bacterial production were done four times during the biological sampling period by incubating 6-10 ml subsamples of the ice-FSW solution with $^3$H-leucine (final concentration of 10 nM) for 3h at 0°C in darkness (Kirchmann, 2001). Half of the samples were spiked with trichloroacetic acid (TCA, final concentration 5%) as controls prior to the incubation, while the remaining active subsamples were fixed with TCA (final concentration 5%) after incubation. Following the incubation, vials were placed in 80°C water for 15 minutes (Garneau et al., 2006) before filtration through 0.2 μm cellulose acetate membranes (Whatmann brand) and rinsing with 5% TCA and 95% ethanol. Filters were dried and dissolved in scintillation vials by adding 1 ml ethyl acetate, and radioactivity was measured on a liquid scintillation counter after an extraction period of 24 h. Bacterial production was calculated using the equations of Kirchman (1993) and a conversion factor of 1.5 kg C mol$^{-1}$ (Ducklow et al., 2003)."
As shown in the previous comments, the level on bacterial production or Chl *a* were too low to have any impact of the $T$CO$_2$ during the experiment.

Line 252. Same as earlier about "dissolution of ikaite in water column and sea ice".
  See previous responses regarding your comments on L 127-141-208.

Lines 260-261. This statement is not valid as is. Please change to: "Assuming no biological effect, ikaite precipitation/dissolution and gas exchange (TCO$_2$), TA and TCO$_2$ are considered conservative with salinity. Thus we can calculate..."
  Thanks for the correction.

Line 257. Repetition: a ratio 2:1.
  We want to keep that repetition as it is essential that the reader understand the concept of ratio TA:$T$CO$_2$ of 2:1 to understand how we estimated the precipitation/dissolution of ikaite within sea ice and the underlying seawater.

Line 269. Add "assumed to be only due to..." after "...this experiment are..".
  Thanks for the correction.

Lines 271-274. Please explain better what you mean with "lack of TA". What do you mean with either dissolved or exported out of the sample? What means "exported out of the seawater sample"?
  We agree that this was unclear so we have revised the text to read: "The difference between TA$_{(sample)}$* and the observed TA is only due to the precipitation or dissolution of ikaite crystals. In case of ikaite precipitation (*i.e.* TA$_{(sample)}$* > TA$_{(sample)}$), half of this positive difference corresponds to the amount of ikaite precipitated within the ice. This ikaite may either remain or may be exported out of the ice. A negative difference (*i.e.* TA$_{(sample)}$* < TA$_{(sample)}$), indicates ikaite dissolution."

Lines 277-278. "ikaite is precipitated and CO$_2$ released from the ice to the atmosphere ;

both processes reduce $TA_{(ice)}$ and $TCO_{2(ice)}$." This statement should be changed since $TA_{(ice)}$ is not reduced by $CO_2$ exchange.

This statement has been changed, now the manuscript reads: "The higher $TA_{(ice)}^*$ and $TCO_{2(ice)}^*$ compared to the averaged $TA_{(ice)}$ and $TCO_{2(ice)}$ (Fig. 7a, b) is expected as ikaite is precipitated (Rysgaard et al., 2014) and $CO_2$ released from the ice to the atmosphere (Fig. 5, 6); processes reducing $TA_{(ice)}$ and $TCO_{2(ice)}$."

Line 285-286: What is "relatively high sea ice temperatures"? Is this temperature high enough for ikaite dissolution, "likely promote ikaite dissolution"? Please explain. I would think that it is more likely that ikaite is rejected from the sea ice to the underlying water due to increased brine volume and dissolved later (storage, analysis?). It would be good to relate this temperature increase in the sea ice to brine volume values (e.g. >5%) when the brine channels connect to each other and promote solutes and gases to escape from the ice. Presenting the evolution of the brine volume fractions in a table or figure during the study would improve some of the understanding of the results, as was suggested earlier in this review.

According to our method used to measure $TA_{(ice)}$ and estimate $TA_{(ice)}^*$: the difference between the $TA_{(ice)}^*$ (TA expected from the salinity changes) and the observed TA is assumed to only be due to the precipitation/dissolution of ikaite crystals. In case of ikaite precipitation (*i.e.* $TA_{(ice)}^* > TA_{(ice)}$), half of this positive difference corresponds to the amount of ikaite precipitated within the ice. This ikaite may either remain or may be exported out of the ice. A negative difference (*i.e.* $TA_{(ice)}^* < TA_{(ice)}$), indicates ikaite dissolution.

Therefore, the warmer temperature observed in the ice and the negative difference between $TA_{(ice)}^*$ and $TA_{(ice)}$ indicates the ikaite dissolution. However, we are also considering the possibility for an export of ikaite from the ice to the underlying seawater to happen as the brine volume increased and the vertical permeability of the sea ice increased at that time of the experiment. This is already mentioned in the manuscript. We added the reference to the sea ice brine volume content in the ice.

Line 296. What do you mean with "good agreement"? Please specify.

This statement has been changed, now the manuscript read: "Both ikaite measurements are of the same order of magnitude however the average (22 µmol kg$^{-1}$) and maximum (100 µmol kg$^{-1}$) of direct observations presented by Rysgaard et al. (2014) were lower than our estimated average (40 µmol kg$^{-1}$) and maximum of up to 167 µmol kg$^{-1}$ over this whole experiment. Deviations are likely due to methodological differences. Here, sea ice samples were melted to subsample for TA and $TCO_2$, Ikaite crystals may have dissolved during melting, leading to an underestimation of the total amount of ikaite precipitated in the ice. However, the difference between $TA_{(ice)}^*$ and $TA_{(ice)}$ provides an estimation of how much ikaite is precipitated in the ice cover, including those crystals potentially already exported to the underlying seawater. The method used by Rysgaard et al., (2014) avoid the bias of ikaite dissolution during sea ice melt with the caveat that crystals need to be large enough to be optically detected. If no crystals were observed, Rysgaard et al., (2014) assumed that no crystals were precipitated in the ice, though ikaite crystals could have been formed and then exported into the underlying seawater prior to microscopic observation of the sample, which may explain the difference observed between both

methods during initial sea ice formation (15-18 January) when the ice was still very thin. In addition, the succession of upward percolation events could have facilitated the ikaite export from the ice cover to the underlying seawater. Estimations from both methods show similar concentrations when the ice (i) warmed due to snowfall (18-23 January) and (ii) cooled once the snow was removed (on 23 January). Once the ice started to melt (26 January), Rysgaard et al., (2014) reported a decrease in the ikaite precipitation while in this study we reported a negative difference between $TA_{(ice)}^*$ and $TA_{(ice)}$, possibly indicating that ikaite dissolved in the ice."

Lines 298-300. This sentence could perhaps be moved to the method description.
In the method section, we already have this sentence: "The plastic bag was sealed immediately and excess air was gently removed through the valve using a vacuum pump. The bagged sea ice samples were then melted in the dark at 4°C to minimize the dissolution of calcium carbonate precipitates (meltwater temperature never rose significantly above 0°C)." This implies what is stated on lines 298-300 of the original manuscript.

Lines 311-313. Please add "in this study" between "underlying seawater" and "is the dissolution...". Also add "export of ikaite from the ice" before "dissolution of..." so the sentence will be: " ....carbonate system in the underlying water in this study is the export of ikaite from the ice and dissolution of calcium carbonate". Please change the next sentence to: "While a few studies of ikaite precipitation....".
We changed the text accordingly.

Lines 315-318. Please add: "according to the study by Fransson et al. (2013)" after where the crystals are dissolved". This study needs to be mentioned since this is one of the first studies describing the carbonate chemistry (such as TA, $TCO_2$) evolution of the sea ice and underlying water (upper 10m) and the sea ice processes such as precipitation and dissolution of ikaite, affecting TA, $TCO_2$ and aragonite saturation from ice formation (in November) to ice melt (in June). They suggested that the high TA found in the upper 10 m under the sea ice was a result of solid ikaite rejected from the ice, dissolved in the water or in the sample before analysis.
We added the reference.

Line 319. Please explain how you obtained the 66 µmol kg$^{-1}$ maximum concentration.
The 66 $\mu$mol kg$^{-1}$ comes, as stated in the manuscript, from half the difference between $TA_{(sw)}^*$ and $TA_{(sw)}$. It increases from 0 at the first day (as $TA_{(sw)}^*$=$TA_{(sw)}$) to a maximum of 66 $\mu$mol kg$^{-1}$ the last day, as shown in the figure 8a and mentioned in the manuscript.

Line 320. Change to "17-days long".
We changed it into "During this experiment"

Lines 336-345. I am concerned about the 1-day delay of the measured $pCO_{2sw}$ compared to the $npCO_{2sw}$-normalized values in Figure 3d after turning on the heat. This is unclear to me since this temperature increase should be directly discerned in $pCO_{2sw}$ and it has to be explained or discussed. Why is there a delay? The sentence "process other than a the temperature change affected the $pCO_{2(sw)}$". Do you have any suggestions on what other processes affected $pCO_{2(sw)}$"?

We changed the text as followed:

"The $pCO_{2(sw)}$ is highly correlated with the seawater temperature (Fig. 2) with a rapid decrease of $pCO_{2(sw)}$ during the first days of the experiment (13 to 15 January) and a relative constant $pCO_{2(sw)}$ until 27 January. However, on 26 January, the heat was turned back on affecting the seawater temperature on the same day (Fig. 2) while the impact of increasing temperature on the $pCO_{2(sw)}$ appeared one day later (Fig. 3d). We normalized the $pCO_{2(sw)}$ to a temperature of -1°C (after Copin-Montegut (1988), noted as $npCO_{2(sw)}$, blue line on Fig. 3d). The $npCO_{2(sw)}$, does not show major variations during sea ice growth with values around 380 µatm. However, once the heat is turned on and the seawater temperature increased (on 26 January), $npCO_{2(sw)}$ decreased from 383 µatm to 365 µatm, while $pCO_{2(sw)}$ did not change in response to increased seawater temperatures until 27 January, suggesting that a process other than temperature change affected the $pCO_{2(sw)}$.

According to equation 1, the dissolution of calcium carbonate has the potential to reduce $pCO_{2(sw)}$. Therefore, during sea ice growth and the associated release of salt, TA, $TCO_2$ and ikaite crystals to the underlying seawater, ikaite dissolution within the seawater could be responsible for maintaining stable $pCO_{2(sw)}$ values while seawater salinity, $TA_{(sw)}$ and $TCO_{2(sw)}$ are increasing. Once the seawater temperature increased (26 January), sea ice melt likely released ikaite crystals to the underlying seawater (Fig. 2, 8a) along with brine and meltwater, a process that would continuously export ikaite from the sea ice as the volume interacting with the seawater via percolation or convection increased. The dissolution of these crystals likely contributed to keeping the $pCO_{2(sw)}$ low and counterbalancing the effect of increased temperature. We argued that once all the ikaite crystals are dissolved, the increase seawater temperature increased the $pCO_{2(sw)}$ simultaneously with the $npCO_{2(sw)}$ (27 January, Figure 3)."

Lines 355-357. Compare with brine volume fraction.

We added the figure panel of the brine volume content in the ice during the whole experiment to Figure 2. This figure is explained in the "sea ice and seawater physical conditions" conditions and show that the ice cover is mainly "permeable", according to the permeability threshold of 5% brine volume from Golden et al (2007). The only 2 occasions where the ice was "impermeable" was on 23 January, when the ice cooled down due to the snow removal from its surface and during the early sea ice growth. This suggests that brine and/or seawater could freely circulate within the ice cover, along with ikaite crystals. Therefore I don't think we need to add something in the text as we suggest 1) the ikaite rejection along with the brine and 2) the increase of the brine connectivity could facilitate the exchange sea ice-seawater.

Lines 358-367 and Figures 8a, b. The calculation procedure is difficult to follow and information on volumes of water and sea ice are missing. I am not convinced why the ikaite (mole) in seawater is so large. It is mentioned in the Figure 7c caption (almost same figure as Figure 8a) that "the ikaite is estimated from half of the difference between $TA_{(ice)}^*$ and $TA_{(ice)}$", but in the figures it seem that data is not presented as "half". Could you explain? How was "0 to 43% of ikaite crystals remain" calculated?

We add the data used for the calculation. The exact values are presented earlier in the manuscript (see next comment). Data presented in the fig 8a are half the difference between $TA_{(sw)}^*$ and $TA_{(sw)}$. Regarding the estimation of ikaite remaining in the ice. We

changed the section 5.3 as followed:

"We estimated the amount of ikaite precipitated and dissolved within sea ice and seawater based on the sea ice (and seawater) volume (in $m^3$), the sea ice and seawater density, the concentration of ikaite precipitated and dissolved within the ice cover (Fig. 7c), and the concentration of ikaite dissolved in the water column (Fig. 8a). Within the ice cover, the amount of ikaite precipitated-dissolved ranged from -0.7 to 1.97 mol (Fig 8b, Table 2), with a maximum just after the snow was cleared on 23 January. In the underlying seawater, the amount of ikaite dissolved in the pool increased from 0.47 mol on the first day of the experiment to 11.5 mol on 25 January when sea ice growth ceased. Once the ice started to melt the amount of dissolved ikaite increased up to 20.9 (28 Jan) and 26.7 mol (29 January, Table 2). The estimation of ikaite dissolution in the pool is significantly higher than the estimated amount of ikaite precipitated (and potentially exported) within the ice cover, especially during sea ice melt. Within the ice cover, the ikaite values presented here represent a snapshot of the ikaite content in the ice at the time of sampling. In the underlying seawater, ikaite dissolution increased $TA_{(sw)}$ cumulatively over time.

The difference between $TA_{(ice)}^*$ and $TA_{(ice)}$ provides an estimation of ikaite precipitated within the ice, including potential ikaite export to the underlying seawater, so it cannot be used to determine how much ikaite remained in the ice versus how much dissolved in the water column. However, Rysgaard et al., (2014) indicate ikaite precipitated within the ice based on direct observations. Using the ikaite concentration reported in Rysgaard et al (2014) (and shown in Fig. 7c), the sea ice volume (in $m^3$) and density, we calculate that 0 to 3.05 mol of ikaite precipitated within the ice cover during sea ice growth (Fig. 8b and Table 2). This amount decreased to 0.46 and 0.55 mol during the sea ice melt (28 and 29 January, respectively). Increased ikaite dissolution in the water column when the ice began to melt (from 11.5 to 20.9 mol) indicates that 9.4 mol of ikaite were stored in the ice and rejected upon the sea ice melt. This amount is about three times the amount of ikaite precipitated in the ice estimated by Rysgaard et al., (2014) at the end of the growth phase (3.05 mol, Table 2), suggesting more work is needed best estimate ikaite precipitation within sea ice.

Once the ice started to melt, the increased ikaite dissolution from 11.5 mol to 20.9 mol (28 January) and to 26.7 mol (29 January) suggests that about the same amount of ikaite is dissolved during the sea ice growth as during the first two days of the sea ice melt. The amount of ikaite dissolved in the water column after melt commenced continued to increase cumulatively, suggesting that ikaite is continuously exported to the underlying seawater as increased sea ice temperatures permit more of the volume to communicate with the underlying seawater. Therefore, we can assume than more than half of the amount of ikaite precipitated within the ice remained in the ice cover before ice melt began."

Lines 376-377. Please provide numbers of your parameters such as volume, density, and pool dimensions used in the calculations.

The size and volume of the pool is given in the section "site description", the volume of seawater-sea ice is fixed and started with a seawater depth of 2.6 m (as described in the site description section). Once the ice started to grow, the seawater depth decreases by the volume of sea ice growth. Sea ice and seawater density are calculated based on temperature and salinity using long-standing equations found in the literature.

Line 380. Was the $CO_2$ fluxes measured on snow and on ice from removed snow?
  See previous respond regarding the same question.

Line 388. Add "(up to 99% as brine)"...
  We changed the text accordingly.

Line 396-398. What was the wind speed during the study? It would be interesting to know since $CO_2$ fluxes are highly dependent on wind speed.
  See previous respond regarding the same question.

Line 430-431. This statement is not right. The effect of processes in sea ice such as ikaite precipitation and dissolution affecting the carbonate chemistry and aragonite saturation state (ocean acidification) in the under-ice water has been address in the seasonal study by Fransson et al. (2013). This study should be mentioned. However, in natural sea ice, there is also advection and other processes acting on the under-ice water, which makes the artificial mesocosm experiment a suitable environment to study effects in a more confined and controlled way.
  The manuscript now reads: "However, any understanding of the effect of ikaite precipitation in sea ice on ocean acidification is still in its infancy (e.g. Fransson et al., 2013)."

Line 436. "sea ice decreases pH and increases $\Omega_{aragonite}$". Could you please explain why they change in opposite directions?
  We made a mistake and change the manuscript as followed: "During ice growth, sea ice brine rejection appears to increase both pH (from 8.00 to 8.06) and $\Omega_{aragonite}$ (from 1.28 to 1.65) of the underlying seawater, offsetting the effect of decreased temperature. A slight increase of $\Omega_{aragonite}$ was predicted due to increased salinity and a proportional increase of TA and $T\mathrm{CO}_2$ as depicted in $\Omega_{aragonite}{}^{*}$. However, the effect of ikaite rejection and subsequent changes in TA strongly enhance the increase of $\Omega_{aragonite}$."

Lines 438-446. There are few studies such as Chierici el al. (2011) that I suggest should be mentioned in this discussion since this is the first study of the changes of the carbonate chemistry and aragonite saturation state in the underlying water (mixed layer) during a full annual cycle in the Arctic, covering all seasons (autumn, winter, spring and summer). They found relatively low $\Omega_{aragonite}$ in winter under the ice, explained mainly by remineralisation and brine rejection. In spring, $\Omega_{aragonite}$ increased mainly as a result of primary production. Fransson et al. (2013) also studied the carbonate chemistry and $\Omega_{aragonite}$ in underlying water but focused on the upper 10m, showing more of the impacts of sea ice processes.
  We added the reference Chierici et al., (2011) and the change the text as followed: "Since the discovery of ikaite precipitation in sea ice (Dieckmann et al., 2008), research on its impact on the carbonate system of the underlying seawater has been ongoing. Depending on the timing and location of this precipitation within sea ice, the impact for the atmosphere and the water column in terms of $CO_2$ transport can be significantly different (Delille et al., 2014). Dissolution of ikaite within melting sea ice in the spring and export of this related high TA:$T\mathrm{CO}_2$ ratio meltwater from the ice to the water column will decrease the $p\mathrm{CO}_2$, increase pH and $\Omega_{aragonite}$ of the surface layer

seawater. Accordingly, during sea ice melt, an increase of $\Omega_{aragonite}$ in the surface water in the Arctic was observed (Chierici et al., 2011, Fransson et al., 2013, Bates et al., 2014). However, it was difficult to ascribe this increase to the legacy of excess TA in sea ice, ikaite dissolution or primary production."

**Conclusion**

Line 448. "17-day".
We changed the first sentence of the conclusions. Now it reads: "We quantified the evolution of inorganic carbon dynamics from initial sea ice formation from open water to its melt in a sea ice-seawater mesocosm pool from 11 to 29 January 2013."

Line 451. Change the sentence to "....while export of ikaite from the ice and dissolution of ikaite was the main ...."
We made the requested change.

**Tables**

Table 1. should include more information such as all sampling occasions, not only start conditions.
We added the conditions at the end of the sea ice growth and at the end of the experiment. The rest of the dataset is shown in the figures in the manuscript. We prefer these data are contained in figures because they provide more immediately meaningful information than tables can provide.

Table 2. This table could perhaps also include "ikaite (mol) seawater". In the header it should be added "sea ice" in the ikaite (mol) column.
We added the variables asked and changed the header.

**Figures and figure captions:**

Figure 2. This figure is not the result of this manuscript and has been presented in Rysgaard et al. (2014). I suggest moving it to background information for the site description. Figure 2d is very unclear and it is impossible to discern the different parameters, and should be changed. Figure 2d caption is unclear of what is what with the different colors and depths shown in the figure. I suggest separating salinity and temperature in two different figures for clarity.
We would like to keep the figure 2 as it is even if yes a small part of the data presented in this figure were presented in the manuscript of Rysgaard et al (2014). We made some changes in the plot 2d to increase the visibility and made the plot clearer.

It is difficult to see if the salinity is higher at the end of the experiment or not. This has to be more evident in the method and discussed, if the salinity never returns to start salinity.
With the changes made in the figure 2D, we can easily see the S at the beginning and the end of the experiment. We also added these data in the table 1, as asked by the reviewer 1. The salinity does not return to the start salinity, simply because we stopped the experiment while we still had ice in the pool. Therefore a significant amount of freshwater was still "unavailable" to dilute the pool back to the start salinity.

Please decide if you use big (A) or small letter (a) in caption and figure, be consistent.

We did, thanks.

Figure 3. $TA^*$ and $TCO_2^*$ are defined when this figure is referred to. The parameters should be defined in the result section (when the figure is firstly mentioned) to understand the results shown in Figure 3.

Fig. 3 is first mentioned when we are presenting the $TA_{(sw)}$, $TCO_{2(sw)}$ and $pCO_2$ data in the results section. Then, in the discussion, when we are introducing $TA_{(sw)}^*$ and $TCO_{2(sw)}^*$ and discussing it, we are referring again to the figure 3 as both are shown there. It doesn't make sense to introduce $TA_{(sw)}^*$ and $TCO_{2(sw)}^*$ in the results section as the reader won't understand at that stage why we are introducing these variables. Therefore we will keep the figure 3 and the text as it is now.

Figure 3d has also very unclear colors. The "blue" line should be defined in the figure caption. In addition, add and define $TA^*$ and $TCO_2^*$ in the caption (a, b) as well as add the color "black" (a,b) and "red" and "black" and "green" (c) for more consistent presentations of the data.

The black line on each panel represent the concentration measured during the experiment ($TA$, $TCO_2$, $pCO_2$) and the figure caption reads: "the seawater $pCO_2$ ($\mu$atm) measured in situ (black) and corrected to a constant temperature of -1°C (blue)." And "In panels (**a**) and (**b**) the black line is the average over the three depths while the dotted red line is the expected concentrations according to the variation of salinity observed and calculated from the mean values of the three depths." Therefore, the black line is defined in both the figure caption and the figure itself.

As shown here, the blue line is defined, no change needed.

$TA^*$ and $TCO_2^*$ are also defined as the figure caption reads: "the dotted red line is the expected concentrations according to the variation of salinity observed and calculated from the mean values of the three depths." However, to please the reviewer, we added some information to the end of the sentence that now reads: "In panels (**a**) and (**b**) the black line is the average over the three depths while the dotted red line is the expected concentrations according to the variation of salinity observed and calculated from the mean values of the three depths ($TA_{(sw)}^*$ and $TCO_{2(sw)}^*$, respectively)."

I am concerned about the 1-day delay of the measured $pCO_{2sw}$ compared to the $npCO_{2sw}$-normalized values in Figure 3d after turning on the heat. This is unclear to me since this temperature increase should be directly discerned in $pCO_{2sw}$ and it has to be explained or discussed. Why is there a delay?

See our responses earlier in the review on your comment on L336.

Figure 5. Add "positive air-ice $CO_2$ flux means outgassing from the ice and negative $CO_2$ flux means uptake of atmospheric $CO_2$.

We add the text as suggested.

Figure 6. Define the green dotted line in caption.

We added the following text to the figure caption: A linear regression is shown in green for the ice samples (a) and blue for the seawater samples (b).

Figures 7. The figure 7b of changes in $TCO_2$ includes $CO_2$ flux but it does not say in the text.

The figure 7b is the evolution of $TCO_{2(ice)}$ and the expected $TCO_{2(ice)}$ according to the salinity changes. Changes of $TCO_2$ in sea ice are not only linked to the $CO_2$ (gas) exchanges, but also ikaite precipitation-dissolution and exchange of $TCO_2$ between sea ice and the underlying seawater.

Figure 7c does not show "half the TA" as I can see. Please explain or I missed something. We doubled check the calculation and yes, the fig 7c does show half the difference between $TA_{(ice)}^{*}$ and $TA_{(ice)}$.

Figure 9. What explains the large difference on the 24-25 January between ice-water exchange of $CO_2$ and total $TCO_2$ loss from sea ice? The maximum amount of ikaite precipitated in the ice happen just after the snow clearing (after 23 January) so, the big difference on the 24-25 January are due to more $TCO_2$ trapped under the form on ikaite than before.

**Added references:**

Anderson, L.G., E. Falck., E. P. Jones., S. Jutterström and J. H. Swift. 2014 Enhanced uptake of atmospheric $CO_2$ during freezing of seawater: A field study in Storfjorden, Svalbard. JGR Vol. 109, C06004, doi:10.1029/2003JC002120, 2004

Brown et al. (2016)

Chierici, M., Fransson, A., Lansard, B., Miller, L.A., A. Mucci., E. Shadwick., H. Thomas, J E. Tremblay., T. Papakyriakou. 2011. *The impact of biogeochemical processes and environmental factors on the calcium carbonate saturation state in the Circumpolar Flaw Lead in the Amundsen Gulf, Arctic Ocean*. JGR-Oceans. 116, C00G09, doi:10.1029/2011JC007184.

Fransson, A., Chierici, M., Miller, L.A., Carnat, G. Papakyriakou T, et al., 2013. Impact of sea-ice processes on the carbonate system and ocean acidification at the ice-water interface in the Arctic Ocean. 2013. Journal of Geophysical Research-Oceans, 118, 1–23, doi:10.1002/2013JC009164.

Hu, Y., D.A..Wolf-Gladrow, G.S. Dieckmann, C. Völker, G. Nehrke 2014. A laboratory study of ikaite ($CaCO_3\cdot6H_2O$) precipitation as a function of pH, salinity, temperature and phosphate concentration, Marine Chemistry 162 (2014) 10–18, http://dx.doi.org/10.1016/j.marchem.2014.02.003

Nomura D, Assmy P, Nehrke G, Granskog MA, Fischer M, Dieckmann GS, Fransson A, Hu Y, Schnetger B, 2013. Characterization of ikaite ($CaCO_3 2\bullet6H_2O$) crystals in first-year Arctic sea ice north of Svalbard. Annals of Glaciology,

54(63)doi:10.3189/2013AoJ62A034

---

## Author Response (AR2)

Referee # 3:

I suggest adding a comment on that salinity and TA in seawater did not have the same concentrations before the experiment (t=0) and after (=end) the experiment due to remaining sea ice in the pool. That is what I mean with "closing the TA budget" (perhaps wrong wording). TA should be the same in the seawater before and after finishing the experiment, if all ice melted (and taking into account the sampled ice TA). For clarification, please just add a comment in the method section or when introducing Table 1 or Figure 2.

→We have changed the caption for table 1 to:
   "Note that seawater salinity and TA$_{(sw)}$ do not reach the initial seawater values as sea ice was still present at the end of the experiment."
   We also added a comment on line 229 in the introduction of the TA and TCO$_2$ measurements in the pool that reads "As the experiment stopped before the ice was completely melted in the tank, both the seawater salinity and TA$_{(sw)}$ do not reach their initial values at the end of the experiment (Table 1, Fig 2 and 3)."

Lines 510-513, after comment on the study by Fransson et al 2013 "....release of brine decreased Ωaragonite by 0.8 ......related to CO2 enrichment of brine". I suggest adding (for consistency): "Conversely, during ice melt, Ωaragonite increased by 1.4 between March and May, likely due to both CaCO3 dissolution and primary production. This contrasts...."

→We followed the suggestion. Thanks.

Line 517, after .....DOE (1994). "This shows the complexity of ikaite and its impact on the carbonate system and Ω in the underlying water."

→We followed the suggestion. Thanks.

Line 528. Add "may" before "therefore" so that the sentence reads ".....impede the effect of ikaite rejection and may therefore lower Ωaragonite.

→We followed the suggestion. Thanks.

Table 2. Please check the significant numbers of decimals and values for all the parameters. Are two decimals significant regarding the accuracy in the measurements and calculations? I think that one or no decimal is more accurate. This is valid also for Figure 6 linear regression equations and R2.
Table 2. What parameter is in the first column (and unit?)? Date/time in decimals?

→The rule of significant numbers specifies that we should use the number of significant figures in the factor with the least significant figures. In our case, the least amount of significant figures is 4 (sea ice and seawater volume). Therefore, all data presented in table 2 has a maximum of 4 significant numbers. Data from table 2 are also rounded to the number of significant figures in the factor with the least significant figures (here again, 4). Which is why, for small values, we keep 2 decimals. The first column is the Day-of-year (DOY). We've added this information in the table.

Suggested additional reference Lyakhin (1970): show evidence that sea ice can release solid $CaCO_3$ to the underlying water
Reference:
Lyakhin, Y. I. (1970), Saturation of water of the Sea of Okhotsk with calcium carbonate, Oceanology, 10, 789–795.
→ Thanks for the reference, however I don't have the reference and I have a hard time finding it.

Referee # 4:
The paper analyses a three week mesocosm study performed at the outdoor sea ice research facility SERF in Winnipeg with respect to the transfer of ikaite crystals to the underlying water during sea ice growth and melt, and it;'s effect on the water and ice carbon budget. The facility allows for a constrained environment which allows to analyze the full "ocean" budget, with out the uncertainty of horizontal and vertical processes in a deep ocean.

The study seems a reasonable set up and analyses performed adequately. The results provide some important additions to our understanding of the role of ikaite precipitation and dissolution during sea ice growth and melt processes.

I recommend publication with minor revisions.

Requested revisions are mostly minor clarifications with a few concerns with respect to the suitability of the set up for the analysis of specific processes. My main question is with respect to the melting procedure, i.e. does melting initiated from an underwater source provide the adequate representation for sea ice melting usually initiated by atmospheric warming and incident shortwave. This is particularly relevant when discussing release of ikaite crystals through melted channels at the bottom of the sea ice versus exchange of CO2 at the surface.

→SERF experiments are of course inherently constrained by the facility's infrastructure; it is not possible to simulate sea ice melt during January from the top as might occur in the Arctic. However, the bottom sea ice melt prescribed in this experiment was as representative as possible of the natural environment. For example, (i) the case of incomplete sea ice cover with substantial spring/summer incident radiation heating the surface ocean (e.g. Babb et al., 2016), or (ii) complete sea ice cover in the presence of a substantial ocean heat flux (REF). The heat was turned on in the pool while the atmospheric temperature was increasing (see figure 2). Changes observed in the ice cover (temperature, salinity, opening of the brine channels) occur in the same way as in polar regions under similar forcing.
Specifically addressing the reviewer's comment above regarding "discussing release of ikaite crystals through melted channels at the bottom of sea ice versus exchange of $CO_2$ at the surface", observation of spring/summer sea ice temperature profiles as the ice warms have long indicated that the middle the sea ice volume stays the coldest the longest; this cold middle area inhibits connectivity between the warming surface ice layer and the relatively warm bottom ice layer bathed in seawater warmed by incident solar radiation. For example, Babb et al., (2016) show 198 cm of bottom melt vs. 135 cm of surface melt as a multiyear sea ice floe (523cm winter thickness) warmed with a continuously c-shaped temperature profile.
Babb, D.G. and others, 2016, Physical processes contributing to an ice free Beaufort Sea during September 2012, Journal of Geophysical Research Oceans, 121, 267-283, doi: 10.1002/2015JC010756.

Minor comments:

Abstract

The first 1-3 sentences seem a bit disconnected and I am feeling I am missing a sentence on what ikaite is about.
→ The first 3 sentences were already changed in the most recent version. The reviewer has read and reviewed a previous version of the manuscript.

sentence 1 could be
→I don't understand what the reviewer means by this comment.
L14 rm "from open water"
→This sentence has changed since the version on which the reviewer commented.

L19 up to => therefrom , then rm :of the ikaite...within sea ice"
→ This sentence has changed since the version on which the reviewer commented.

Main text:
L29 released to the atmosphere - from where?
→The sentence refers to anthropogenic carbon, the release of carbon due to human activities.

L50 rejected from sea ice - clarify direction (where to)
→ We've added "...to the underlying seawater"

L67/68 fluxes are....incorporated into water masses ....or below sea ice – something is wrong here, please clarify sentence
→The sentence was already edited as a result of the previous round of review.

L70 Are you referring to the shrinking of the brine channels due to freezing ? If so please clarify
→We prefer to leave this sentence as it is.

L72-74 would move the "internal ice melt" forward , since that is what increases the ibrine volume, correct?
→ The sentence to which the reviewer is referring currently states that internal melting dilutes brine resulting in decreased brine salinity.  This of course would increase the brine volume, but the important point here is decreased brine salinity as it relates to decreased TA and $TCO_2$ leading to lower $pCO_2$. So, we prefer to leave the sentence as it stands.

L75 Eq 3 – should this be Equ 1?
→This was corrected in the previous round of revisions.

L86 of the different processes
→We've made the suggested change.

L90 the addition "from open water" makes the sentence a bit convoluted. I think it would be sufficient to clarify once at the beginning that the experiment starts with open water, then only write " from initial sea ice formation to melt" ( This occurs several times in the manuscript)

→We followed the suggestion, thanks.

L95 suggest to rm "major changes in"

→We deleted "major" from the sentence.

L104 formulated => formed ???

→This text was changed to "made by" in the previous round of reviews.

L106ff How realistic is it to have an under water heat source versus an atmospheric heat source, particularly in terms of melting the ice structure and opening of brine channels from top or bottom, allowing fluxes out of the ice ???

→We have endeavored to address this comment on the first page of this document when the question of heating from below was first raised by the reviewer.

L113 How well mixed is the underlying water – is 1.3m a representative depth?

→The 1.3m was chosen because it is half of the seawater depth in the pool (cf. line 109 in the newer version). A discussion on how well mixed the pool can be found lines 219-222 of the most recent version of the manuscript.

L244 rm Therefore (previous sentence is not the cause of this)

→The sentence was changed in the previous round of revision of the manuscript.

L 260 expected : expected based on what? (does that refer to no ikaite interaction?)

→Expected based on the fact that TA and $TCO_2$ are conservative with salinity (cf the beginning of the sentence.) The sentence was changed in the most recent round of revisions prior to this review (line 299).

L267 throughout the ice

→ The sentence was changed in the most recent round of revisions prior to this review.

section 5.2.1 Suggest to define some dTA for the difference between TA(ice)* and TA(ice), this might make the text clearer.

→We prefer to keep the text as it is now.

L279 black dots =diamonds

→We made the suggested change.

L280 .. highly variable ( isn't that something that would be expected, ie. Concentration needs to become high enough for crystallization, but once it is precipitated the concentration needs to be build up again before more crystallization can happen??? - not sure, I am not a chemist...)

→ It is expected as this is something we observed by direct observation in the manuscript of Rysgaard et al (2014) (during the same SERF experiment). By variable we mainly suggest that ikaite can precipitate-dissolve during the same experiment, based on the ice temperature, as suggested by Rysgaard et al (2014).

L282 I can't see this in the Figure (7a??) please clarify. If differences are minor a zoom in and clarification might help.
→These differences are displayed on Figure 7c, as mentioned in the text (in both old and new version of the manuscript).

L290-309 This section would profit from some streamlining, as is the message is confusing
e.g. L299 you talk about underestimation while the figure shows mostly overestimation till Jan 17. While this is addressed later it doesn't make sense where it is stated.
L301-303 could ikaite also be redissolved with sea water infiltration?
L307-308 clarify that the message is precip is not only reduced but actual dissolution occurs
L309 can you discern this from the water column? If so state this and refer to section where this is discussed
→This whole section (5.2.1. Sea ice) has been changed as a result of the previous round of revision; it's now lines 316-358 in the newest version.

Section 5.22 could the export towards the underlying sea water be intensified by the melting from the bottom? (Since it already starts before the melting is turned on, it might not be the case, but it might be good to discuss.
→Here as well, we substantially changed section 5.3 "Ikaite export from the ice to the water column" as a result of the previous round of revision. The reviewers' concern above is addressed in the newest version of the paper.

L320 one-month long – Isn't the experiment 17 days only or is this another experiment?
→The text has been changed according to previous reviewer asking the same thing.

L 326 started => starts
→ Thank for the correction.

L 320-349 again, this starts to become confusing, any streamlining would help
→ We substantially changed this text as a result of the previous round of revision. The reviewers' concern above is addressed in the newest version of the paper.
L344 delay of reaction ??
→ 1 day delay. The text changed in the newer version of the manuscript.

L358 Please clarify if you were actually measuring ikaite or derived it.
→ We substantially changed this text as a result of the previous round of revision. The reviewers' concern above is addressed in the newest version of the paper.

L367 rm contain

→ The sentence was changed in the most recent round of revisions prior to this review.

5.4 This title is strange, I suggest to put the " attempt of a CO2 budget in a separate section (from line 400 and maybe frame it as an error or uncertainty analysis...)
→ We have changed the title of this section; it now reads "Air-ice-seawater exchange of inorganic carbon".

L399 Are you generating this kind of turbulence in the tank?
→4 pumps were installed at the bottom of the pool to inducing a consistent current. This is explained in the new version of the manuscript, line 121, in the method section.

L424 Suggest: As a result several area => Several areas
→Already corrected, thanks.

L427 the annual
→Already corrected, thanks.

L428 saturation state
→Thanks for the correction.

L440 suggest using "alleviate" instead of "hamper"
→Thanks for the correction.

L448 month-long – 17 day?
→Already corrected, thanks.

L449 rm "from open water"
→We've removed "open water".

L450 suggest "direct CO2 exchange" to clarify this is released from the ice to the atmosphere
→During this experiment, we measured released of $CO_2$ from the ice to the atmosphere and uptake of atmospheric $CO_2$. Therefore we will keep the sentence at it is now.

L452 estimation => estimates; rm here
→The text changed in the new version of the manuscript, thanks.

L455 were missing ???? had been removed?
→The text changed in the new version of the manuscript, thanks.

L470 responsible formation
→The text changed in the new version of the manuscript, thanks.

Figure 2 use consistent ABCD, abcd in caption and figure
→This has been corrected in the new version, thanks.

Panel d indicate blue and green, What is the black line in panel d, also the different symbols are not discernible
→This has been corrected in the new version, thanks.

caption last line: profile => profiles
→Thanks for the correction.

Fig 3 caption clarify what is indicated with * (E.g. add in brackets after "variation of salinity observed" (**).
→This has been corrected in the new version, thanks.

Fig. 6 what are the green and blue lines
→This has been corrected in the new version, thanks.

[revised manuscript text omitted]

Nicolas-Xavier Geilfus 13/7/2016 10:48

Nicolas-Xavier Geilfus 13/7/2016 10:48

Nicolas-Xavier Geilfus 13/7/2016 10:48

Nicolas-Xavier Geilfus 13/7/2016 10:48

Nicolas-Xavier Geilfus 13/7/2016 10:48

Nicolas-Xavier Geilfus 13/7/2016 10:48

Nicolas-Xavier Geilfus 13/7/2016 10:48

Nicolas-Xavier Geilfus 13/7/2016 10:49

Nicolas-Xavier Geilfus 13/7/2016 10:49

Nicolas-Xavier Geilfus 13/7/2016 10:49

Nicolas-Xavier Geilfus 13/7/2016 10:49

Nicolas-Xavier Geilfus 13/7/2016 10:49

Nicolas-Xavier Geilfus 13/7/2016 10:49

Nicolas-Xavier Geilfus 13/7/2016 10:49

Nicolas-Xavier Geilfus 13/7/2016 10:49

Nicolas-Xavier Geilfus 13/7/2016 10:49

Nicolas-Xavier Geilfus 13/7/2016 10:49

Nicolas-Xavier Geilfus 13/7/2016 10:49

Nicolas-Xavier Geilfus 13/7/2016 10:49

Nicolas-Xavier Geilfus 13/7/2016 10:49

Nicolas-Xavier Geilfus 13/7/2016 10:49

Nicolas-Xavier Geilfus 13/7/2016 10:49

Nicolas-Xavier Geilfus 13/7/2016 10:49

Nicolas-Xavier Geilfus 13/7/2016 10:50

Nicolas-Xavier Geilfus 13/7/2016 10:50

Nicolas-Xavier Geilfus 13/7/2016 10:50

Nicolas-Xavier Geilfus 13/7/2016 10:50

Nicolas-Xavier Geilfus 13/7/2016 10:50

---

## Author Response (AR3)

Editor Decision: Publish subject to minor revisions (Editor review) (11 Aug 2016) by Prof. Dr. Lars Kaleschke

Comments to the Author:

Dear authors,

After reading again the manuscript and your answers to the reviewers I have only three remaining minor questions before I can accept the paper:

1) Introduction:

The Arctic Ocean are taking up -66 to -199 Tg C year$^{-1}$, contributing 5-14% to the global ocean CO2 uptake (Bates and Mathis, 2009)

Why given with a negative sign? Should be clear that it means uptake.

→It is generally admit and recognize in the literature that a negative flux corresponds to an uptake while a positive flux corresponds to a release. We add this precision in the text.

I am not sure where the numbers (24-100 Tg C year$^{-1}$) in IPCC FAQ 6.1 Figure 1 (page 66) come from but they are considerably smaller.
http://www.ipcc.ch/pdf/assessment-report/ar5/wg1/WG1AR5_Chapter06_FINAL.pdf

→Thanks for this question. The estimation from the IPCC (24-100 Tg of C yr$^{-1}$) comes from McGuire et al (2009), Sensitivity of the carbon cycle in the Arctic to climate change, Ecological Monographs, 79(4), pp523-555.

In this work, they used the estimation from Bates 2006 (31-45 Tg C yr$^{-1}$) and from Anderson et al (1998b) (about 24 Tg C yr$^{-1}$). From there they estimate that the Arctic Ocean is taking up from 24 to 100 Tg C yr$^{-1}$. How they reach this number is not mentioned in the manuscript. The only precision is the followed (p533)

"Given the estimates of Anderson et al. (1998b) and Bates (2006), we infer that the mean annual sink for atmospheric $CO_2$ of the Arctic Ocean and its associated shelf seas lies between 24 and 100 Tg C/yr."

Which is not helping... In the discussion of the figure 3 (p540) from McGuire et al (2009) where the Arctic Ocean uptake is mentioned (24 to 100 Tg C yr$^{-1}$), they also wrote the following:

"Our review indicates that the Arctic plays an important role in the global dynamics of both $CO_2$ and CH4. Top-down atmospheric analyses indicate that the Arctic is a sink for atmospheric $CO_2$ of between 0 and 0.8 Pg C/yr (Fig. 3), which is between 0% and 25% of the net land/ocean flux of 3.2 Pg C/yr estimated for the 1990s by the IPCC's Fourth Assessment Report (AR4; Denman et al. 2007)."

They don't specify how they reach the estimation 0-0.8 Pg of C/yr... While this number is mentioned in both the abstract and the discussion.

Therefore, I can't really answer your question as I have contradictory information coming from the referred manuscript. I could only suggest to use the most recent estimation of the atmospheric $CO_2$ sink for the Arctic Ocean realized by Bates and Mathis (2009), as McGuire et al (2009) is using their old estimation.

2) Figure 6 release/uptake? Not sure if I understand the direction of the fluxes. Perhaps it is reverted?

→I'm not sure to understand. The figure related to the fluxes is the figure 5, not 6. I hope the clarification of the flux unit from the previous comment will help understand the presentation of the flux data.
The manuscript read:
"The $CO_2$ fluxes measured at the variably snow-covered sea ice surface (Fig. 2b), ranged from 0.29 to 4.43 mmol m$^{-2}$ d$^{-1}$ show that growing sea ice released $CO_2$ to the atmosphere (Fig. 5). However, as soon as the ice started to warm up and then melt, the sea ice switched from source to sink for atmospheric $CO_2$ with downward fluxes from -1.3 to -2.8 mmol m$^{-2}$ d$^{-1}$."
The show that a positive flux corresponds to a release of $CO_2$ to the atmosphere and a negative flux an uptake of atmospheric $CO_2$, as explained now in the introduction.

3) The emission of salts to the atmosphere can very likely be neglected in your overall budget. However, calcium carbonate has been found in firn of Talos Dome, Antarctica and thought to originate from sea ice. This is important for atmospheric chemistry and climate reconstructions from ice cores. Perhaps worthwhile to mention?

→The report of calcium carbonate in continental ice in Antarctica was, at some point, suggested to come from sea ice and based on the assumption that calcium carbonate could precipitate within sea ice. At the time, ikaite precipitation within sea ice was not formally known and/or reported in the literature.
Now the missing link will be the atmospheric export of ikaite from the sea ice cover. This is far away from my field of expertise and there is nothing in our present manuscript that could bring anything to answer that question.

Line 435 Typo: Therefore, we can assume thaT more than

→ Thx for the correction.

Best regards

Lars Kaleschke